# In situ cell-surface conformation of the TCR-CD3 signaling complex

Aswin Natarajan[1,2], Yogambigai Velmurugu [ID][1,2], Manuel Becerra Flores[3], Fatoumatta Dibba[1,2], Saikiran Beesam [ID][1,2], Sally Kikvadze [ID][1,2], Xiaotian Wang [ID][1,2], Wenjuan Wang[1,2], Tianqi Li[1,2], Hye Won Shin[1,2], Timothy Cardozo[3] & Michelle Krogsgaard [ID][1,2][✉]

## Abstract

The extracellular molecular organization of the individual CD3 subunits around the αβ T cell receptor (TCR) is critical for initiating T cell signaling. In this study, we incorporate photo-crosslinkers at specific sites within the TCRα, TCRβ, CD3δ, and CD3γ subunits. Through crosslinking and docking, we identify a CD3ε′-CD3γ-CD3ε-CD3δ arrangement situated around the αβTCR in situ within the cell surface environment. We demonstrate the importance of cholesterol in maintaining the stability of the complex and that the 'in situ' complex structure mirrors the structure from 'detergent-purified' complexes. In addition, mutations aimed at stabilizing extracellular TCR-CD3 interfaces lead to poor signaling, suggesting that subunit fluidity is indispensable for signaling. Finally, employing photo-crosslinking and CD3 tetramer assays, we show that the TCR-CD3 complex undergoes minimal subunit movements or reorientations upon interaction with activating antibodies and pMHC tetramers. This suggests an absence of 'inactive-active' conformational states in the TCR constant regions and the extracellular CD3 subunits, unlike the transmembrane regions of the complex. This study contributes a nuanced understanding of TCR signaling, which may inform the development of therapeutics for immune-related disorders.

Keywords T Cell Receptor; TCR-CD3 Complex; Photo-crosslinking; Computational Docking; TCR Signaling
Subject Categories Immunology; Signal Transduction; Structural Biology

## Introduction

T cell receptors (TCRs) play a crucial role in initiating T cell immune responses by recognizing antigenic peptides presented by major histocompatibility complexes (MHC) expressed on antigen-presenting cells (APCs) and signal through associated CD3 subunits (Krogsgaard and Davis, 2005). The αβTCR is a heterodimeric molecule consisting of variable (Vα, Vβ) and constant (Cα, Cβ) domains in each subunit. Antigen recognition is facilitated by complementarity-determining regions (CDRs) within the variable domains, while the constant domains interact with CD3 subunits (Davis and Bjorkman, 1988; Natarajan et al, 2016).

The αβTCR-CD3 complex is composed of an αβTCR heterodimer, a CD3γε heterodimer, a CD3δε heterodimer, and a CD3ζζ homodimer. Notably, αβTCR possesses C-terminal regions lacking any intracellular signaling domains. Each CD3 subunit's cytoplasmic tail contain either 1 or 3 immunoreceptor tyrosine-based activation motifs (ITAMs), which can be phosphorylated to propagate signals within the cell (Kane et al, 2000). This arrangement implies that communication of cognate pMHC-TCR interactions to T cell cytoplasm must occur through the CD3 subunits. This underscores the pivotal role of CD3 subunits in transducing signals from TCR engagement to initiate downstream T cell activation.

The anchoring core of the TCR complex is formed by the bundle of transmembrane helices (TMs) of the TCR and the CD3 chains. Highly conserved, charged residues in the TMs, along with a membrane-proximal tetracysteine motif, are required for the clustering all TCR-CD3 complex components (Borroto et al, 1998; Call et al, 2002; Call and Wucherpfennig, 2005; Xu et al, 2006) and important for signal transmission across the TM regions (Wang et al, 2009). Interactions between extracellular regions are essential for the formation of the bioactive TCR-CD3γε/δε complex (Fernandes et al, 2012; He et al, 2015) and subsequent T-cell signaling (Natarajan et al, 2016).

NMR chemical shift perturbation (CSP) studies of the extracellular components of αβTCR and CD3 subunits, while providing residue-specific information, suggested different binding modes (single-sided (He et al, 2015) and double-sided (Natarajan et al, 2016)) of CD3γε/δε to the αβTCR. Indeed, the peptide linking segment between the CD3 extracellular folded domains and their corresponding TMs is sufficiently long to accommodate either a one-sided or two-sided conformation. Previous studies offered insights into the composition and orientation of the TCR-CD3 complex, but lacked the native TCR-CD3 TMs (Arechaga et al, 2010; Birnbaum et al, 2014; He et al, 2015; Natarajan et al, 2016).

[1]Laura and Isaac Perlmutter Cancer Center, NYU Grossman School of Medicine, New York, NY 10016, USA. [2]Department of Pathology, NYU Grossman School of Medicine, New York, NY 10016, USA. [3]Department of Biochemistry and Molecular Pharmacology, NYU Grossman School of Medicine, New York, NY 10016, USA. [✉]E-mail: Michelle.Krogsgaard@nyulangone.org

Recent non-crystalline, cryo-EM structures of the human TCR-CD3 complex included the connecting peptide linker segments and the 8-TM helix bundle (without intracellular cytosolic regions), and revealed the orientation of CD3 subunits and specific contact details between individual subunits (Chen et al, 2022; Dong et al, 2019; Saotome et al, 2023; Susac et al, 2022). Notably, these structures showed both CD3γε and CD3δε binding to the TCR from the same side in non-MHC-ligated and pMHC-ligated states. While the use of detergents raises concerns about the physiological relevance of these observations, recent cryo-EM studies indicated that pMHC induced minimal to no conformational changes in the TCR constant domains or CD3 subunits in the TCR-CD3 complex (Saotome et al, 2023; Susac et al, 2022). However, a previous report suggests that engagement of multiple TCR-CD3 complexes by dimeric or tetrameric pMHC is required to induce CD3ε cytosolic conformational changes (Minguet et al, 2007). Moreover, Lanz et al, identified that pMHC tetramer binding induces loosening of αβTCR association with CD3ζ although monomeric pMHC binding can still cause allosteric changes in TCR-CD3 complex that led to intracellular signaling (Lanz et al, 2021). In this study, we employ a panel of pMHC tetramers, known to elicit varying functional responses (Krogsgaard et al, 2003), to probe specific crosslink changes and, consequently, conformational changes in the TCR-CD3 complex in the native membrane surface.

Photo-crosslinking of incorporated unnatural amino acids (UAA) is a powerful tool for investigating complex protein-protein interactions, molecular mechanisms, and spatiotemporal conformational states (Coin, 2018; Coin et al, 2013). This technique has been successfully employed to map ligand-binding sites for diverse proteins, including G protein-coupled receptors, neurokinin-1 receptor, and a human serotonin transporter (Gagnon et al, 2019; Grunbeck et al, 2011; Rannversson et al, 2016; Valentin-Hansen et al, 2014). Noteworthy applications of photo-crosslinking also include the analysis of histone-histone interactions leading to chromatin condensations (Wilkins et al, 2014) and identification of RNA-binding sites in riboprotein complexes (Kramer et al, 2014). Cell-based photo-crosslinking techniques were used to identify transient complex formations in multiple protein systems involving transcription activators VP16 and Gal4 with Swi/Snf chromatin remodeling complex (Krishnamurthy et al, 2011), during nuclear transport involving nucleoporins (Yu et al, 2012), chaperone-assisted protein folding (Zhang et al, 2011) among others. Photo-crosslinking, distinguished by its precision and specificity in capturing dynamic interactions, stands out as a superior method compared to other non-specific crosslinking approaches, by allowing for targeted probing of molecular associations, providing a nuanced understanding of complex structures. Despite its effectiveness, this technique has yet to be extensively applied to the study structures of immune receptors.

In this study, we present a model depicting the in situ cell-surface conformation of the TCR-CD3 signaling complex in mice, using constraints derived from site-specific photo-crosslinkers. This model reveals a one-sided CD3ε′-CD3γ-CD3ε-CD3δ subunit arrangement around the αβTCR. A detailed comparison with the previously solved human TCR-CD3 cryo-EM structures (Chen et al, 2022; Dong et al, 2019; Saotome et al, 2023; Susac et al, 2022) highlights a similar overall arrangement with specific differences, particularly in the TCR-CD3 interface residues. This approach underscores the considerable utility of combining photo-crosslinking with computational molecular docking for the in-depth study of protein-protein interactions in multi-subunit receptors. Furthermore, photo-crosslinking enabled the determination of the essential role of cholesterol in the stability of the TCR-CD3 complex assembly and demonstrated the overall similarity between cell-surface and detergent-purified TCR-CD3 complex structures. Introducing mutations based on cross-link and cryo-EM structures to stabilize the TCR-CD3 interface resulted in dampened T cell signaling, suggesting that subunit flexibility is crucial for signal transduction. Finally, binding of antibodies and pMHC tetramers to the TCR-CD3 complex did not induce noticeable crosslink changes, indicating an absence of large stable conformational changes or subunit reorientations during signal propagation in the TCR-CD3 complex. This study contributes valuable insights into the structural dynamics and functional aspects of the TCR-CD3 complex, emphasizing the broader relevance of photo-crosslinking methodologies in unraveling immune receptor mechanisms.

## Results

### The TCR-CD3 complex is amenable to UAA incorporation and photo-crosslinking

To incorporate photo-crosslinkable UAAs at specific codons, such as the amber stop codon, facilitating crosslinking of TCR and CD3 subunits, we designed orthogonal tRNA/aminoacyl-tRNA synthetase (tRNA-aaRS) pairs. These pairs incorporate UAA present in cell culture media into the nascent protein at designated sites (Fig. 1A). In previous work, we successfully co-transfected plasmids encoding tRNA-aaRS, TCR, and CD3 subunits into human embryonic kidney (HEK) 293T cells and incorporated UAA photo-crosslinker, such as p-azido-phenylalanine (pAzpa) and H-p-Bz-Phe-OH (pBpa), site-specifically into the TCR (Wang et al, 2014). The efficacy of this approach was demonstrated by probing interactions between TCR subunits by photo-crosslinking (Wang et al, 2014).

Building upon this work, the present study aimed to map extracellular TCR-CD3 interactions in a native cell environment by incorporating the unnatural amino acid pAzpa into previously identified TCR-CD3 interaction sites (Beddoe et al, 2009; Dong et al, 2019; Kim et al, 2009; Kuhns and Davis, 2007; Natarajan et al, 2016). For larger regions such as Cβ FG loop, Cβ G strand, and Cα AB loop we chose alternating residues for introducing crosslinkers in these regions. We incorporated pAzpa, which has been proven superior to pBpa for incorporation into the TCR (Wang et al, 2014), in the mouse 2B4 TCR constant regions. Subsequently, we crosslinked it to neighboring mouse CD3 subunits, and vice versa, through UV (360 nm) irradiation (Fig. 1B).

For expression on mammalian cells, the TCR subunits (α- and β-) and CD3 subunits (γ-, δ-, ε- and ζ-) were connected by self-cleavable 2A peptides to ensure stoichiometric expression of the different subunits in the TCR-CD3 complex (Fig. 1C). To facilitate detection of crosslinked subunits by Western blot, distinct protein tags were added to the C-terminal ends of various subunits (TCRα: c-Myc, TCRβ: V5, CD3γ: VSV-G, CD3δ: FLAG and CD3ε: HA) (Fig. 1C). To verify pAzpa incorporation and assess photo-crosslinking, we examined the ability of TCRα with S41 and K65

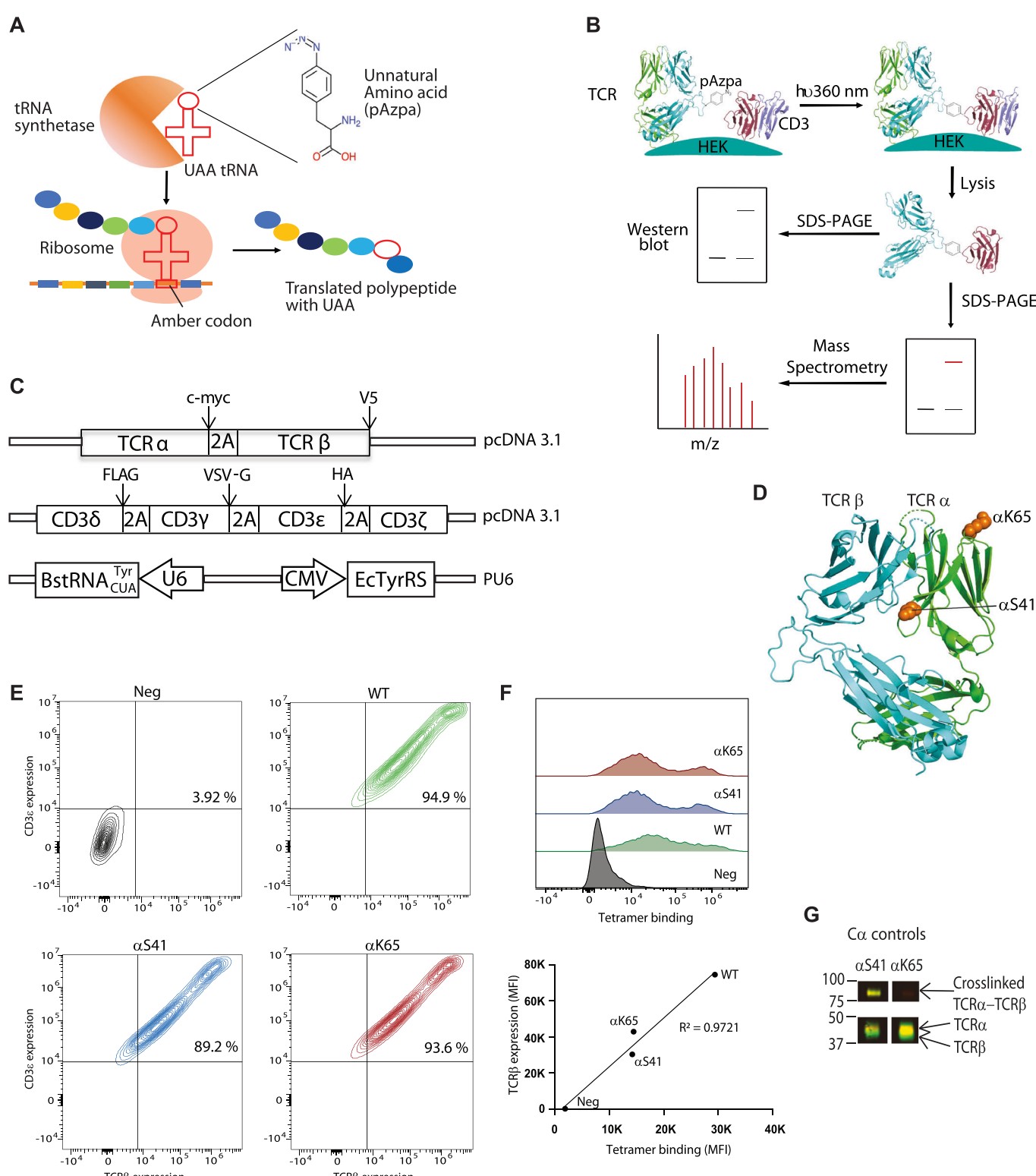

mutations to crosslink TCRβ (Wang et al, 2014) (Fig. 1D). The TCRα S41 mutant serving as a positive control due to its proximity to the TCRβ subunit, and the K65 TCRα mutant, acting as a negative control being distal to the TCRβ subunit in the CDR region, were transfected into 293T cells. Both wild type 2B4 and

mutant TCRα cell lines stained positive for both TCRβ and CD3ε (Fig. 1E). After UV excitation, we assessed the viability of assembled WT and mutant TCR-CD3 complexes for pMHC (I-E$^k$/MCC)(Krogsgaard et al, 2003) tetramer binding and confirmed that tetramer binding correlated with TCR surface expression,

◀ **Figure 1. The membrane-associated TCR-CD3 complex is amenable to UAA incorporation and photo-crosslinking.**

(A) Schematic overview of UAA (pAzpa) incorporation into translated protein by orthogonal tRNA/tRNA synthetase pair. (B) General outline of the steps involved in the crosslinking assay in 293T cells. (C) Illustrations of the 2B4 TCR, CD3, and tRNA-aaRS expression plasmids and locations of peptide tags utilized for Western blot identification. (D) Locations of αS41, αK65 indicated as spheres in the 2B4 TCR crystal structure (PDB: 3QJF). (E) TCRβ and CD3ε expression profiles of wild-type 2B4 TCR (green), αS41 (blue), αK65 (red) by flow cytometry (stained with APC-conjugated H57-597 antibody—TCRβ and PE-conjugated 145-2C11 antibody—CD3ε) shows successful surface and comparable expression after UAA incorporation. The percentage of cells positive for both TCRβ and CD3ε staining is indicated. (F) Top, Histograms of IE^k/MCC-APC tetramer binding to 293T cells transfected with WT, αS41, and αK65 TCR-CD3 complexes. Bottom, Correlation between surface expression of TCRβ (MFI, stained with APC-conjugated H57-597 antibody) and IE^k/MCC tetramer staining (MFI, IE^k/MCC-APC tetramer). (G) TCR mutant αS41 (positive control) crosslinks with TCRβ with the crosslinking band migrating between 75 to 100 kDa illustrating the feasibility of the technique to crosslink nearby subunits. The blot was stained with anti-TCRα (cMyc) antibody and anti-TCRβ (V5). Anti-rabbit IRDye 680LT- and anti-mouse IRDye 800CW were used as secondary antibodies for detection. Source data are available online for this figure.

verifying a fully assembled and functional TCR-CD3 complex (Fig. 1F). Western blot analysis revealed a crosslinked TCRα-TCRβ band for the αS41 mutant, between 75 and 100 kDa, corresponding to the size of TCRα + TCRβ (Fig. 1G). No such band was observed for the negative control K65 TCRα mutant, while non-crosslinked TCRα and TCRβ subunits appeared at bands between 37 and 50 kDa (Fig. 1G). In summary, these results demonstrate our capability to efficiently express correctly assembled TCR-CD3 complexes on the 293T cell surface, validated by tetramer binding. Moreover, we successfully incorporated pAzpa into a specific location in the TCR α-subunit and crosslinked TCR α-subunit to the adjacent TCR β-subunit.

## CD3 subunits crosslinks with specific TCR regions, indicating a one-sided CD3 subunits arrangement around the TCR

Earlier investigations, employing mutagenesis (Kuhns and Davis, 2007), docking (Sun et al, 2004), molecular dynamics (Martinez-Martin et al, 2009), NMR (He et al, 2015; Natarajan et al, 2016), cryo-EM (Dong et al, 2019) and inference from crystal structures (Arnett et al, 2004; Kjer-Nielsen et al, 2004), have identified multiple interaction sites on the TCR for CD3. The proposed interaction sites include the AB loop and DE loop of TCR Cα, and the CC′ loop, FG loop, G strand, helix 3 and helix 4-F strand of TCR Cβ (Fig. 2A; Appendix Table S1). In our study, we utilized UAA (pAzpa) incorporation and crosslinking to probe these interaction sites, aiming to construct a detailed model of TCR-CD3 complex assembly in the native membranal environment on mammalian cells.

Mutagenesis studies have demonstrated that the TCR Cβ CC′ loop interacts with CD3εγ subunits, and the Cα DE loop interacts with CD3εδ (Kuhns and Davis, 2007). Correspondingly, cryo-EM structures show that the Cα DE loop contacts CD3δ, and the Cβ CC′ loop contacts CD3γ (Chen et al, 2022; Dong et al, 2019; Saotome et al, 2023; Susac et al, 2022). To validate these interaction sites, we transfected TCRs with specific mutations into 293T cells, including A172 and D174 in the Cα DE loop and N164, K166, V168, S170 and G171 in the Cβ CC′ loop (Fig. 2A). Post-transfection, 75-96% of 293T cells expressed TCRβ and CD3ε for the Cα DE loop mutants (A172 and D174) and Cβ CC′ loop mutants (βK166, βV168, βS170 and βG171 (Appendix Figs. S1A and S2A). However, the βN164 mutation in the CC′ loop resulted in poor surface expression of the complex, indicating a potential role for βN164 in complex stability (Appendix Fig. S2A). Notably, βN164 is closely associated with CD3γ in cryo-EM structures (Chen et al, 2022; Dong et al, 2019;

Saotome et al, 2023; Susac et al, 2022). IE^k/MCC tetramer binding assays confirmed a correlation between tetramer binding and TCR surface expression for both Cα DE loop and Cβ CC′ loop mutants (Appendix Figs. S1A and S2A). Crosslinking studies demonstrated that A172 and D174 (Cα DE loop) and S170 and G171 (Cβ CC′ loop) crosslinked with the CD3δ subunit of the CD3δε heterodimer (Fig. 2B,C; Appendix Figs. S1B and S2B). Bands corresponding to crosslinked TCRα-CD3δ were observed below the 75 KDa marker (Fig. 2B,C), consistent with the size of the combined crosslinked subunits. In the cryo-EM structure, the residue corresponding to A172 in the Cα DE loop ($S_{186}$ in the human TCR Cα) contacts CD3δ. (Note: the alternate residue numbering in human cryo-EM structure is shown in subscript). In addition, while there is no contact between the Cβ CC′ loop and CD3δ, the other end of the CC′ loop ($G_{182}$) interacts with CD3γ in the cryo-EM structure (Dong et al, 2019). Overall, the relative positioning of the Cα DE loop and Cβ CC′ loop to CD3δ, as determined from our crosslinking data, suggests a consistent alignment of the TCR-CD3 complex in the native cellular environment with the cryo-EM structure (Dong et al, 2019).

High-resolution X-ray structural studies and fluorescence-based experiments have provided insights into conformational changes within the AB loop of the TCR Cα domain upon agonist binding, leading to T cell activation (Beddoe et al, 2009). Notably, the deletion of the AB loop resulted in impaired T cell activation, as evidenced by low CD69 upregulation (Beddoe et al, 2009). Allosteric changes following antigen binding were also identified in the AB loop through an NMR chemical shift perturbation (CSP) study (Rangarajan et al, 2018). These observations prompted the hypothesis that the AB loop might serve as a possible CD3 interaction site. To explore this hypothesis, we transfected cells with TCR constructs containing specific mutations in the Cα AB loop and subsequently subjected them to UV-crosslinking. We then analyzed IE^k/MCC tetramer binding for the following Cα AB loop mutants: D132, R134, Q136, and S138 (Fig. 2D; Appendix Fig. S3A,B). Interestingly, we found that S138 crosslinked to TCRβ, and no crosslinks were observed between the Cα AB loop and any of the CD3 subunits (Fig. 2D; Appendix Fig. S3B). This suggests that the CD3 subunits are not in close proximity to the Cα AB loop. This conclusion is consistent with cryo-EM structures (Chen et al, 2022; Dong et al, 2019; Saotome et al, 2023; Susac et al, 2022), which also indicated the absence of interactions between the Cα AB loop and CD3 subunits.

Upon antigen ligation, the TCR functions as an anisotropic mechanosensor, with the Cβ FG loop serving as a lever to interact with neighboring CD3γε (Kim et al, 2009; Kim et al, 2010). Other

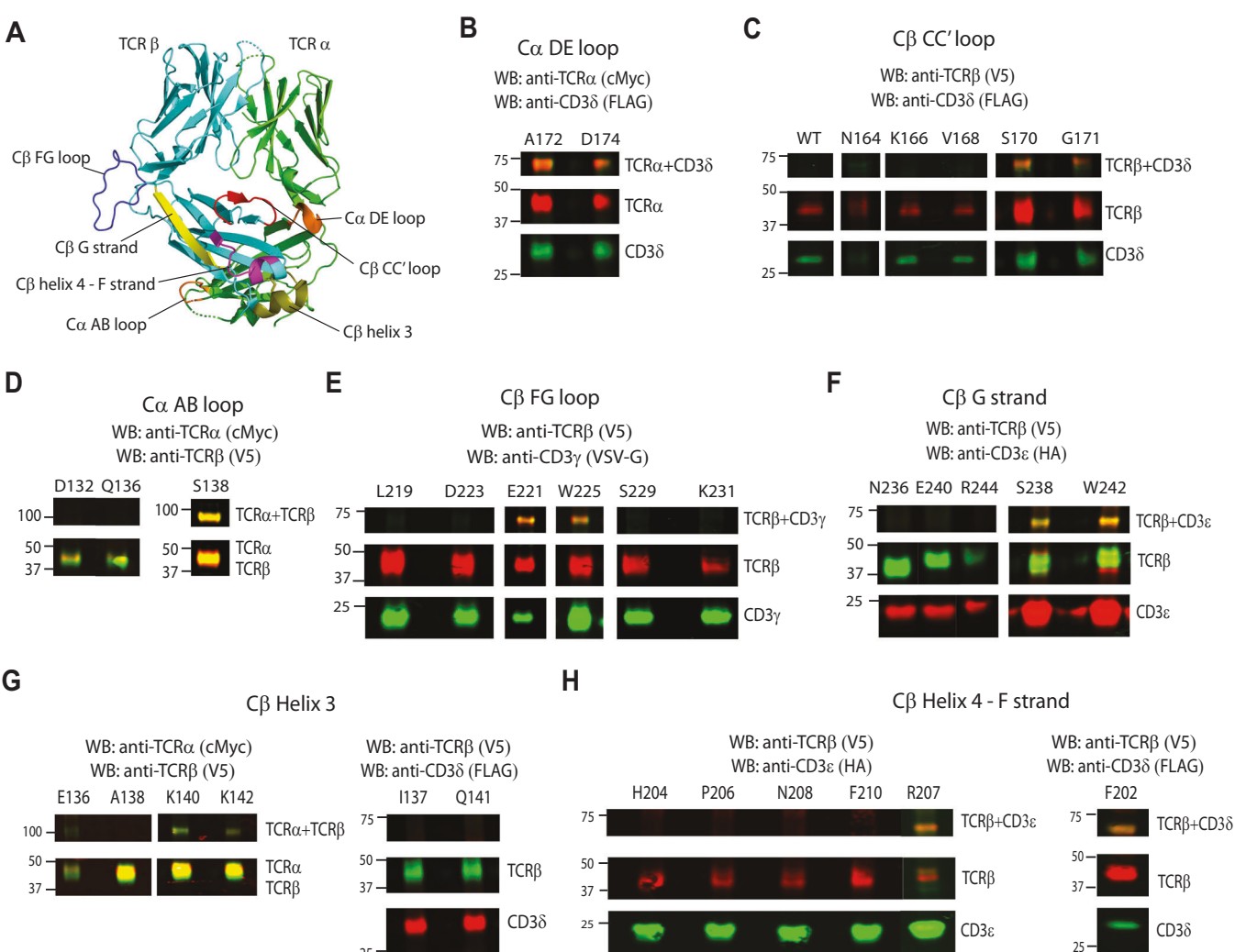

**Figure 2. CD3 subunits crosslinks with specific TCR regions indicating one-sided CD3 subunits arrangement around the TCR.**

(A) Location of Cα DE loop (orange), Cβ CC′ loop (red), Cβ FG loop (blue), Cβ G strand (yellow), Cβ helix 3 (olive), Cβ helix 4-F strand (magenta), and Cα AB loop (orange) on the 2B4 TCR crystal structure (PDB: 3QJF). (B) Western blot analysis of Cα DE loop mutants—A172 and D174. TCRα + CD3δ crosslinking band for both mutants are present below 75 kDa. The blot was stained with anti-TCRα (cMyc) antibody and anti-CD3δ (FLAG). (C) Western blot analysis of Cβ CC′ loop mutants—WT, βN164, βK166, βV168, βS170, and βG171. TCRβ + CD3δ crosslinked bands for βS170 and βG171 are present around 75 kDa. The blot was stained with anti-TCRβ (V5) antibody and anti-CD3δ (FLAG). (D) Western blot analysis of Cα AB loop mutants—αD132, αQ136, and αS138. TCRα + TCRβ crosslinking band for αS138 is present below 100 kDa. The blots were stained with anti-TCRα (cMyc) antibody and anti-TCRβ (V5). (E) Western blot analysis of Cβ FG loop mutants—βL219, βE221, βD223, βW225, βS229, and βK231. TCRβ + CD3γ crosslinked bands for βE221 and βW225 are present below 75 kDa. The blot was stained with anti-TCRβ (V5) antibody and anti-CD3γ (VSVG). (F) Western blot analysis of Cβ G strand mutants—βN236, βS238, βE240, βW242, and βR244. CD3ε crosslinked bands seen for βS238 and βW242 are present below 75 kDa. The blots were stained with anti-TCRβ (V5) antibody and anti-CD3ε (HA). (G) Western blot analysis of Cβ helix 3 mutants—βE136, βI137, βA138, βK140, βQ141, and βK142. TCRα + TCRβ crosslinked bands for βK140 and βK142 are present between 75 and 100 kDa. βE136, βA138, βK140, and βK142 blots were stained with anti-TCRα (cMyc) antibody and anti-TCRβ (V5). βI137 and βQ141 blots were stained with anti-TCRβ (V5) antibody and anti-CD3δ (FLAG). (H) Western blot analysis of Cβ helix 4-F strand mutants—βF202, βH204, βP206, βR207, βN208, and βF210. TCRβ + CD3δ and TCRβ + CD3ε crosslinked bands for βF202 and βR207, respectively, are present below 75 kDa. βH204, βP206, βR207, βN208, and βF210 blots were stained with anti-TCRβ (V5) antibody and anti-CD3ε (HA). βF202 blot was stained with anti-TCRβ (V5) antibody and anti-CD3δ (FLAG). The crosslinking bands in each case are present below 75 kDa. Data information: Anti-rabbit IRDye 680LT- and anti-mouse IRDye 800CW were used for all blots as secondary antibodies for detection. Source data are available online for this figure.

studies using single-molecule analyses, NMR, and molecular dynamics have unveiled the allosteric role of the FG loop in controlling TCR CDR catch-bond formation, peptide discrimination, Vβ-Cβ interface stabilization and CD3 communication (Chang-Gonzalez et al, 2024; Das et al, 2015; Rangarajan et al, 2018). In addition, the FG loop is identified to play a crucial role in T cell development particularly negative selection and proper

complex assembly (Sasada et al, 2002; Touma et al, 2006). Given the recognized significance of the FG loop in T cell function, we conducted transfections, UV crosslinking and IEᵏ/MCC tetramer binding analyses for specific TCR Cβ FG loop mutants, including L219, E221, D223, W225, S229 and K231 (Fig. 2A; Appendix Fig. S4A). Among these mutants, we observed that Cβ FG loop residues E221 and W225 crosslinked with CD3γ of the TCR-CD3

complex, as indicated by the presence of a band below the 75 kDa molecular marker representing crosslinked TCRβ-CD3γ in Western blot analysis (Fig. 2E; Appendix Fig. S4B). Notably, Cβ E221 and W225 did not exhibit crosslinking with CD3δ and CD3ε (Appendix Fig. S4B). However, it bears mentioning that in the cryo-EM structures, CD3ε′ (CD3γε heterodimer) is depicted in closer proximity to the Cβ FG loop (Chen et al, 2022; Dong et al, 2019; Saotome et al, 2023; Susac et al, 2022). Despite this, the crosslinking assay strongly suggests that CD3γ is closer to the Cβ FG loop than CD3ε in the mouse TCR-CD3 complex.

To further validate the proximity of the CD3γ subunit to the Cβ FG loop compared to CD3ε, we conducted mass spectrometry analysis on the excised crosslinked band below 75 kDa from the SDS-PAGE gel of the E221 TCR Cβ FG loop mutant. The analysis revealed a higher number of unique peptide fragments attributed to CD3γ and TCRβ in the cross-linked band (15 and 11, respectively) compared to other TCR-CD3 complex subunits such as CD3δ (6), CD3ζ (7), TCRα (4), and CD3δ (1) (Dataset EV1). This additional evidence strengthens the conclusion that CD3γ is indeed in closer proximity to the Cβ FG loop in the cell surface conformation.

Next, we examined residues within the Cβ G strand region known to interact with various CD3 subunits, particularly being implicated in CD3ε binding according to cryo-EM structures (Chen et al, 2022; Dong et al, 2019; Saotome et al, 2023; Susac et al, 2022). Through transfection, UV-crosslinking, and analysis of IE$^k$/MCC tetramer binding, we investigated Cβ G strand mutants, including N236, S238, E240, W242, and R244 (Fig. 2A; Appendix Fig. S5A). We found that residues S238 and W242 successfully cross-linked with CD3ε, evidenced by the appearance of a band below the 75 kDa molecular marker in the Western blot analysis (Fig. 2F; Appendix Fig. S5B). Importantly, Cβ S238 and W242 did not exhibit cross-linking with CD3γ and CD3δ (Appendix Fig. S5B). Notably, the residue corresponding to W242 in the cryo-EM structure, $W_{259}$, was identified to interact with CD3ε (Dong et al, 2019), indicating a consistent outcome between our crosslinking experiments and the cryo-EM data (Chen et al, 2022; Dong et al, 2019; Saotome et al, 2023; Susac et al, 2022). Mass spectrometry analysis of the excised crosslinked band for W242 mutant revealed PSM (peptide-spectrum match representing the total number of identified peptide sequences for the protein) values of 15 and 26 for TCRβ and CD3ε, respectively (Dataset EV2), while other subunits such as TCRα, CD3γ, CD3δ, and CD3ζ recorded PSMs of 5, 11, 3, and 3, respectively. Overall, these crosslinking experiments provide evidence that CD3γε is in closer proximity to the TCR region encompassing the Cβ FG loop and the Cβ G strand. Specifically, CD3γ is in closer proximity to the FG loop, while CD3ε is in closer proximity to the G strand.

Our earlier NMR analysis of TCR-CD3 ectodomains revealed interactions between TCR Cβ helix 3, helix 4-F strand regions, and CD3γε (Natarajan et al, 2016). Supporting this, other NMR CSP studies indicated helix 3 and helix 4 regions as CD3 binding regions (He et al, 2015) and that these regions undergo allosteric changes upon antigen ligation (Natarajan et al, 2017; Rangarajan et al, 2018). Notably, amino acid changes in the helix 3 region were shown to improve TCR expression and CD3 pairing (Sommermeyer and Uckert, 2010). Based on these reports, we transfected, UV-crosslinked, and analyzed IE$^k$/MCC tetramer binding of the following mutants in 293 T cells: In Cβ helix 3: E136, I137, A138, K140, Q141, and K142 (Fig. 2A; Appendix Fig. S6A); in Cβ helix 4-

F strand region: F202, H204, P206, R207, N208, and F210 (Fig. 2A; Appendix Fig. S7A). No crosslinks were observed between TCR Cβ helix 3 residues and any CD3 subunits. Instead, helix 3 residues K140 and K142 crosslinked with the neighboring TCR α-subunit (Fig. 2G; Appendix Fig. S6B). This observation aligns with cryo-EM structures, which did not show interactions between helix 3 residues and CD3 subunits. For the TCR Cβ helix 4-F strand mutants, crosslinking bands were evident for F202 and R207 below the 75 kDa molecular marker in the Western blot, corresponding to TCRβ + CD3δ and TCRβ + CD3ε, respectively (Fig. 2H; Appendix Fig. S7B). Cβ F202 did not crosslink with CD3γ and CD3ε, and Cβ R207 did not crosslink with CD3γ and CD3δ (Appendix Fig. S7B). This is in contrast to the cryo-EM structure (Dong et al, 2019), in which F strand residue $H_{226}$, which is near R207, stacks against CD3γ in the cryo-EM structure (Dong et al, 2019). Overall, our crosslinking analysis suggests that residues F202 and R207 in the TCR Cβ helix 4-F strand region are closer to CD3δ and CD3ε, respectively, while residues in Cβ helix 3 are not directly involved in CD3 interactions.

In summary, incorporation of UAA into different αβTCR sites revealed distinct spatial relationships within the TCR-CD3 complex. Specifically, CD3δ was found to be in proximity to the Cα DE loop and Cβ CC′ loop. CD3δε was situated closer to the Cβ helix 4-F strand regions. CD3γ exhibited a closer association with the Cβ FG loop, while CD3ε was found to be closer to the Cβ G strand. This mapping provides valuable insights into the structural organization of the mouse αβTCR-CD3 complex and reveals nuanced interactions between its components.

## Crosslinking reveals the CD3 subunit interface interacting with the TCR in the TCR-CD3 complex

After identifying specific TCR residues in close proximity to CD3 subunits, we performed reciprocal experiments to identify CD3 residues in closer proximity to TCR residues, utilizing the same UAA incorporation and photo-crosslinking approach employed for the TCR. The selection of CD3γ and CD3δ residues for UAA incorporation was based on their locations in the interface, near the interface, or away from the interface in the human TCR-CD3 cryo-EM complex structure (Dong et al, 2019). No mutations were introduced into the CD3ε subunit, as distinguishing between CD3ε subunit belonging to CD3δε or CD3γε would be challenging. According to the cryo-EM structure, we transfected, UV-crosslinked, and analyzed IE$^k$/MCC tetramer binding of the following CD3δ mutants in 293T cells: T5 in strand A; E8, D9 in AB loop; T17 in BC loop; V26 in CD loop; T35 in E strand; and K40 in EF loop (Fig. 3A; Appendix Fig. S8A, Appendix Table S2). Among these mutants, T5 was observed to crosslink with TCRβ (Fig. 3B; Appendix Fig. S8B) and T35 and K40 crosslinked with TCRα (Fig. 3B; Appendix Fig. S8B). T5 did not crosslink with TCRα, while T35 and K40 did not crosslink with TCRβ (Appendix Fig. S8B). The conserved residue K40 ($K_{62}$ in the cryo-EM structure) that crosslinks with TCRα is involved in H-bond interaction with TCRα, connecting peptide residue $K_{234}$ in the cryo-EM structure (Dong et al, 2019). Residues T5 and T35 (corresponding to $E_{27}$ and $R_{57}$ in cryo-EM structure) that crosslink with TCRβ and TCRα, respectively, are implicated in polar interactions with TCRα residue $R_{185}$ in the cryo-EM structure. Interestingly, we did not observe crosslinks for the conserved AB

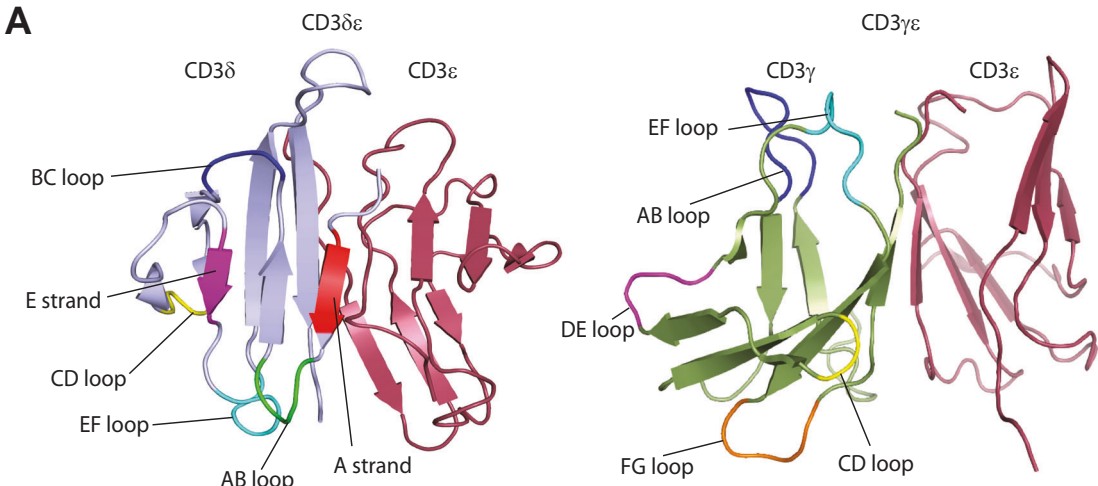

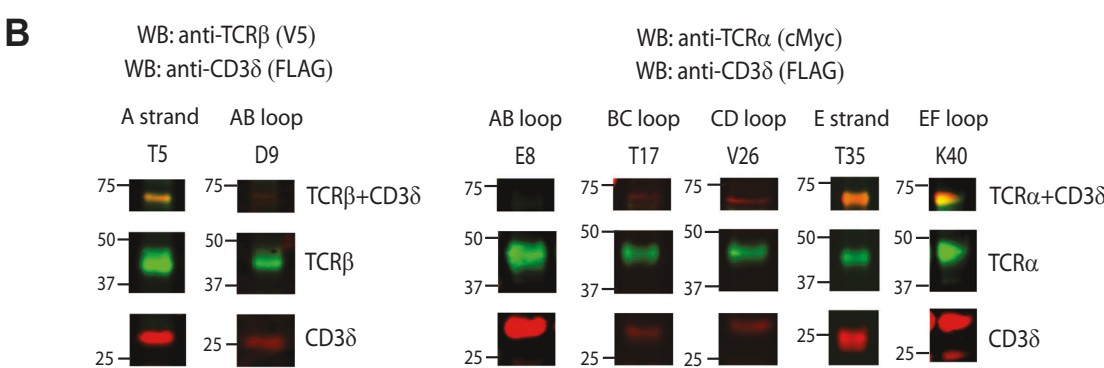

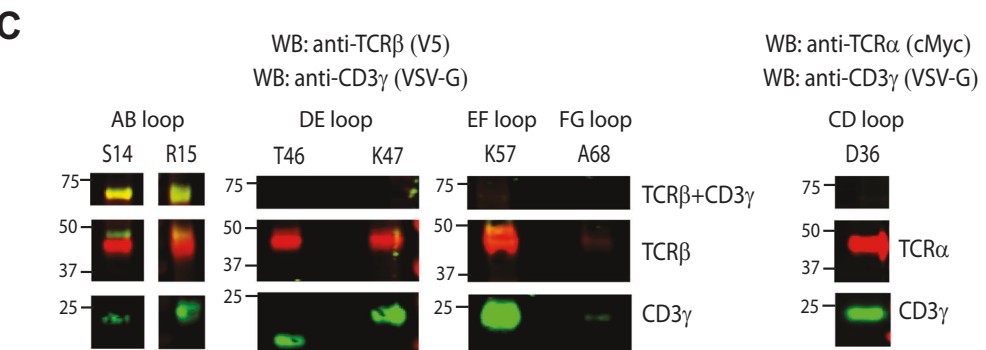

**Figure 3. Crosslinking reveals the CD3 interface subunits facing the TCR in the TCR-CD3 complex.**

(A) Left, location of CD3δ A strand (red), AB loop (green), BC loop (blue), CD loop (yellow), E strand (magenta), and EF loop (cyan) on the CD3δε mouse structure. Right, location of CD3γ AB loop (blue), CD loop (yellow), DE loop (magenta), EF loop (cyan), and FG loop (orange). (B) Western blot analysis of CD3δ mutants—A strand: T5; AB loop: E8, D9; BC loop: T17; CD loop: V26; E strand: T35; and EF loop: K40. TCRβ + CD3δ crosslinked band for T5 is present below 75 kDa. T5 and D9 blots were stained with anti-CD3δ (FLAG) and anti-TCRβ (V5). TCRα + CD3δ crosslinked bands for T35 and K40 are present below 75 kDa. E8, T17, V26, T35, and K40 blots were stained with anti-CD3δ (FLAG) and anti-TCRα (cMyc). (C) Western blot analysis of CD3δ mutants—AB loop: S14, R15; CD loop: D36; DE loop: T46, K47; EF loop: K57 and FG loop: A68. TCRβ + CD3γ crosslinked bands for S14 and R15 are present below 75 kDa. S14, R15, T46, K47, K57 and A68 blots were stained with anti-CD3γ (VSV-G) and anti-TCRβ (V5). D36 blot was stained with anti-CD3γ (VSV-G) and anti-TCRα (cMyc). Data information: Anti-rabbit IRDye 680LT- and anti-mouse IRDye 800CW were used for all blots as secondary antibodies for detection. Source data are available online for this figure.

loop residues E8 and D9, which interact with TCRα in the cryo-EM structure (Dong et al, 2019).

For CD3γ, we conducted transfection, UV-crosslinking, and analysis of IE$^k$/MCC tetramer binding with the following mutants in 293 T cells: S14, R15 in AB loop; D36 in CD loop; T46, K47 in DE loop; K57 in EF loop, and A68 in FG loop (Fig. 3A; Appendix Fig. S9A, Appendix Table S2). Among these mutants, we observed that AB loop residues S14 and R15 crosslink with TCRβ (Fig. 3C; Appendix Fig. S9B). The corresponding residues in the cryo-EM structure, Y$_{36}$ and Q$_{37}$, contact H$_{226}$ and G$_{182}$, respectively (Dong et al, 2019). The other mutants analyzed were in regions distant from the AB loop, suggesting that the region around the AB loop is the one facing and closer to TCRβ. Mass spectrometry analysis of the excised crosslinked band for the S14 mutant indicated PSM values of 23 and 24 for TCRβ and CD3γ, respectively (Dataset EV3). Other subunits, such as TCRα, CD3ε, and CD3ζ, recorded PSMs of 4, 8, and 3, respectively.

To identify differences between the in situ 'cell-surface' conformation and 'detergent-purified' cryo-EM structures, we implemented a modified protocol for our crosslinking experiment. In this approach, we harvested 293T cells expressing mutant TCR-CD3 complexes (Dong et al, 2019), immunoprecipitated the complexes using biotinylated anti-CD3ε antibody (2C11), and then subjected them to UV irradiation to activate crosslinking (Fig. EV1). Crosslinks that were initially identified when mutant TCR-CD3 complexes were UV-irradiated on cells were also present when mutant TCR-CD3 complexes were extracted using DDM buffer and UV-irradiated after immunoprecipitation in the presence of cholesterol (Fig. EV1). This suggests that the cell surface conformation of the TCR-CD3 complex closely mirrors the 'detergent-purified' conformation obtained from cryo-EM structures (Chen et al, 2022; Dong et al, 2019; Saotome et al, 2023; Susac et al, 2022). Interestingly, when using a N-Dodecyl-β-Maltoside (DDM) detergent buffer without cholesteryl hemisuccinate, which is typically used for TCR-CD3 complex extraction, although not completely abolished weaker crosslinking was observed for some of the tested mutants—Cβ E221, CD3δ K40, and Cα A172. This underscores the importance of cholesterol in TCR-CD3 assembly concerning these interactions (Appendix Fig. S10).

## Computational docking generates a CD3ε'-CD3γ-CD3ε-CD3δ model for CD3 binding

To generate a 3D model of the TCR-CD3 complex using crosslinking constraints (Fig. 4A), we employed computational molecular protein-protein docking (Fernandez-Recio et al, 2003). We generated all reasonable, unclashed, compact conformations for the mouse CD3γε and CD3δε domains docked to mouse 2B4 TCR domains, using available structures of human and mouse proteins as templates (Appendix Table S3). Prior to this, we modeled the pAzpa-mutations (both crosslinked and non-crosslinked amino acids) in silico within the mouse TCR, CD3γε, and CD3δε 3D structures. This allowed us to quantify and visualize the resulting perturbation of the local structure and electrostatic surface in the vicinity of the pAzpa mutant side chain, including calculation of the changes in van der Waals, electrostatic and solvation energy. This analysis does not reveal a correlation between energetic perturbation of pAzpa substitutions and the ability to crosslink. Appendix Table S4 shows that mutations that resulted in worse

energy are in the minority and are distributed randomly between crosslinks that were made (blue shaded) and those that did not occur.

Subsequently, we ranked thousands of unclashed, compact TCR-CD3γε-CD3δε extracellular conformations/orientations based on a hierarchy of constraints: (1) geometric and spatial compatibility of the C-termini of the CD3γε and CD3δε extracellular folded domains with the N-termini of their corresponding TM helices in the cryo-EM TM bundle, (2) acceptable distances for crosslinked amino acid pairs between CD3 subunit and TCR chains, (3) acceptable distances for crosslinked amino acid pairs between TCR and CD3 subunits, (4) calculated overall biophysical energy of the complex (van der Waals, solvation electrostatics, hydrogen bonding), (5) absence of encroachment with the plasma membrane, (6) absence of clashes with 2C11 antibody structure bound CD3ε (Leo et al, 1987), and (7) absence of encroachment with the pMHC binding site.

Based on the defined criteria, the most favorable TCR-CD3 conformation satisfying all mentioned constraints (Appendix Table S5) was identified and resembled the reported cryo-EM structure (Fig. 4B, left, right)(Dong et al, 2019). The TMs of all subunits from the cryo-EM structure (Dong et al, 2019) and connecting linkers were incorporated into the crosslink-guided model and found to be independently consistent with the model (e.g., the CD3 domains were docked without constraints and the crosslinks were recorded without bias from the cryo-EM 3D structure: if the TM helices clashed with the CD3 domains, this would be an indication of either an invalid docking method or artificial crosslinks, as would the inability of the linkers to connect the CD3 C-termini to the TM N-termini). This final structural model exhibited an overall contact area of 491.2 Å$^2$ between crosslinked TCR residues and CD3γε and a contact area of 342.4 Å$^2$ between crosslinked TCR residues and CD3δε, with predicted energies (ΔG) of −23.6 Kcal and −14.5 Kcal, respectively.

In this model, the CD3ε' chain of the CD3γε' heterodimer was positioned behind the Cβ FG loop (Fig. 4B, left), diverging from its direct alignment below the FG loop in the cryo-EM structure (Fig. 4B, right). The crosslink-guided model revealed the proximity of the CD3γ subunit to the Cβ FG loop, supported by observed crosslinking between the Cβ FG loop residues and CD3γ (Fig. 4B, center). Our analysis revealed variations in the electrostatic surfaces participating in binding interfaces in the individual components of the human TCR-CD3 complex compared to their modeled mouse counterparts, particularly on the CD3 subunits (Appendix Fig. S11). These distinctions underscored the differences between the human and mouse TCR-CD3 complexes in our model, despite an overall structural resemblance. The positioning of CD3δε on the crosslink-guided TCR-CD3 structure was nearly identical as the equivalent regions (A172 in crosslinking and S$_{186}$ in cryo-EM, both belonging to Cα DE loop) in the cryo-EM structure (Dong et al, 2019), although the relative orientation of CD3δε differed (Appendix Fig. S12). Our structure proposed a CD3ε'-CD3γ-CD3ε-CD3δ model for CD3 binding, emphasizing contacts between CD3γ (belonging to CD3γε) and CD3ε (belonging to CD3δε) (Fig. 4B; Appendix Fig. S12). Overall, the combination of crosslinking analysis and computational docking identified a cell-surface native conformation of the mouse αβTCR-CD3 complex similar to the human cryo-EM structure, albeit with minor distinctions in interface contacts.

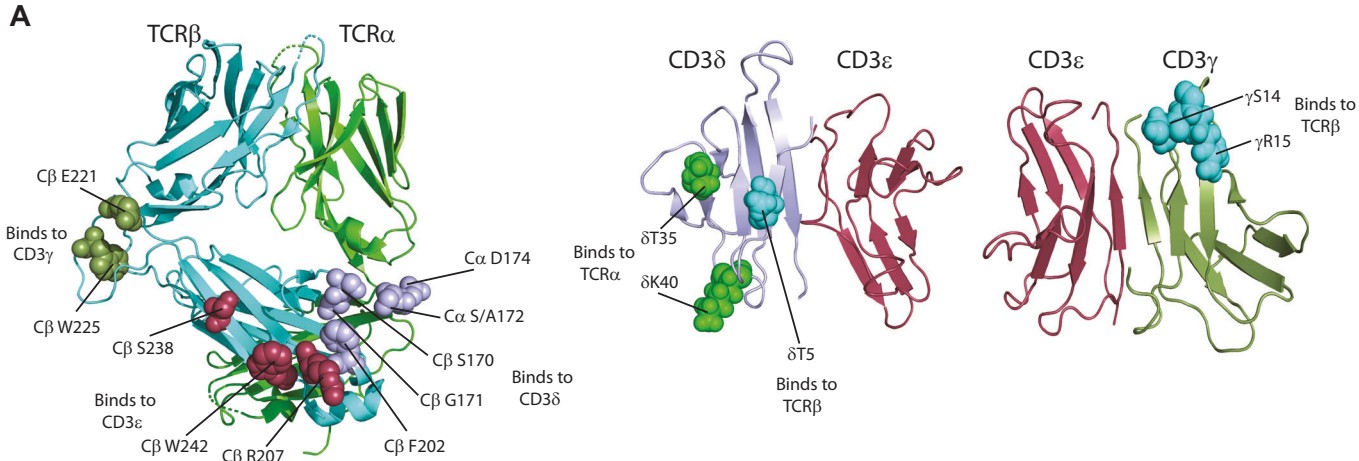

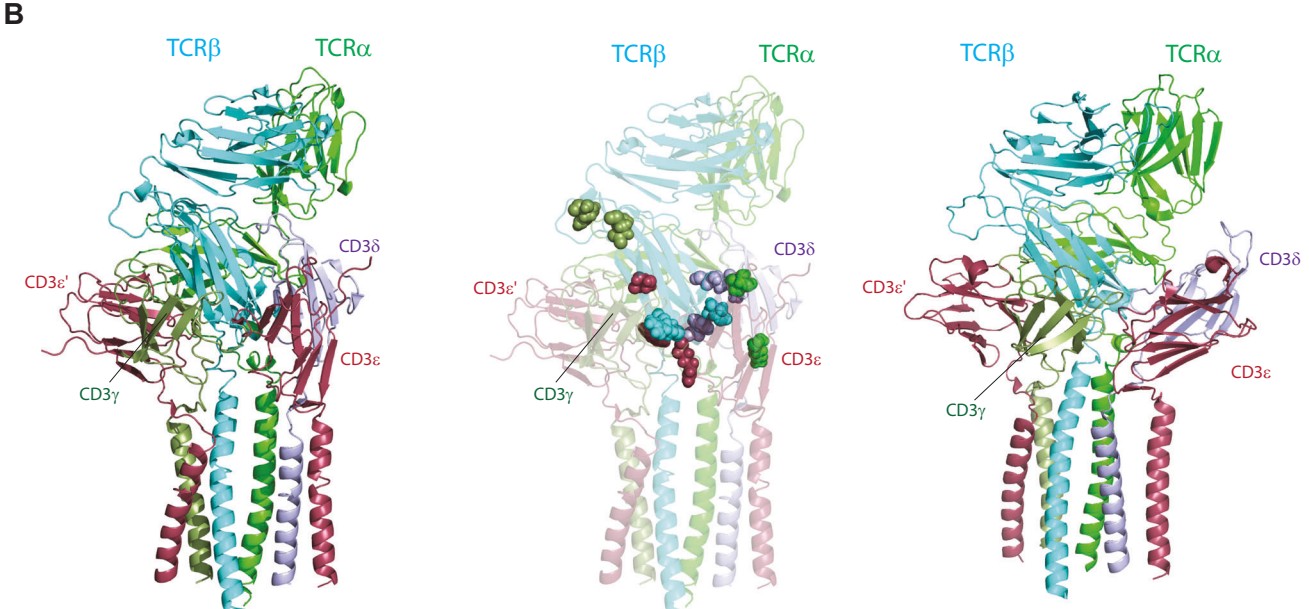

**Figure 4. Computational docking reveals a CD3ε'-CD3γ-CD3ε-CD3δ model for CD3 binding.**

(A) Left, crosslinking TCR residues indicated as spheres on the crystal structure of 2B4 TCR (with human constant domains, PDB: 3QJF). Residues interacting with CD3γ indicated in smudge green, with CD3ε indicated in raspberry and with CD3δ indicated in light blue. Center, crosslinking CD3δ residues indicated as spheres on the mouse CD3δε structure. Residues interacting with TCRα indicated in green, and with TCRβ indicated in cyan. Right, crosslinking CD3γ residues indicated as spheres on the mouse CD3γε structure. Residues interacting with TCRβ are indicated in cyan. (B) Left, docked TCR-CD3 complex structure based on crosslinking derived constraints. CD3γε interact primarily on the TCRβ face of the complex. CD3δε interacts in a region involving the interface of TCRα-TCRβ. The transmembrane bundle is derived from the cryo-EM human TCR-CD3 transmembrane helical bundle (PDB: 6JXR) (Dong et al, 2019). Center, TCR-CD3 crosslink-guided model depicted in cartoon representation (60% transparency) with crosslinking TCR-CD3 residues in spheres. Same color scheme for the residues in spheres as in (A). Right, Cryo-EM TCR-CD3 structure. TCRα, TCRβ, CD3γ, CD3δ, and CD3ε/ε' are indicated in green, cyan, smudge, light blue, and raspberry, respectively.

## Stabilizing TCR-CD3 extracellular interactions reduces TCR signaling

We employed in silico mutagenesis to identify point mutations in the TCR-CD3 complex interfaces of the crosslink-guided structure, targeting those that enhance TCR-CD3 subunit interactions (Appendix Table S6). Mutations were identified in Cβ G strand (N236R and S238K), Cβ FG loop (E227F), and Cα DE loop (D169K) (Fig. 5A) based on overall energy scores relative to the

wild-type structure. Subsequently, functional assays were conducted in T cell hybridoma 58-/- (Letourneur and Malissen, 1989) (expressing CD3 subunits but lacking TCRαβ) to assess whether these TCR mutants enhance T cell activation, measured by IL-2 production (Natarajan et al, 2016; Zhong et al, 2010). Mutant 2B4 TCR constructs were generated through site-directed mutagenesis, and T cell clones expressing these constructs were generated by retroviral transduction (Natarajan et al, 2016; Zhong et al, 2010). Cells were co-cultured with MHCII IE^k-expressing CHO cells

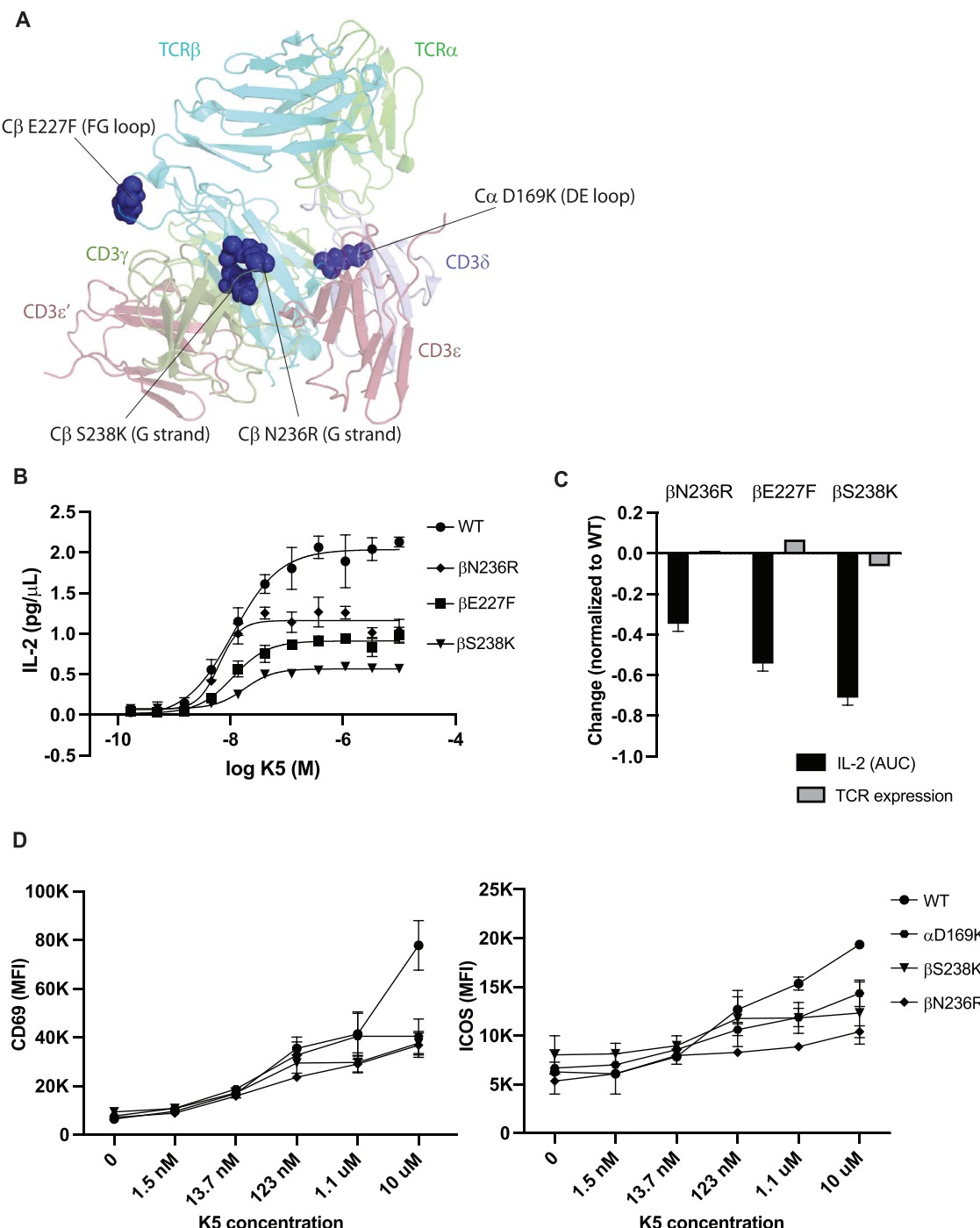

**Figure 5. Stabilizing TCR-CD3 extracellular interactions reduces TCR signaling.**

(**A**) Locations of the mutated residues indicated on the mouse TCR-CD3 crosslink-guided structure. E227F is located in the Cβ FG loop, N236R and S238K are located in the Cβ G strand and D169K is located in Cα DE loop. (**B**) ELISA assays (plot of IL-2 produced vs concentration of activating peptide) for mutant 2B4 T cell hybridoma clones activated with CHO/I-E$^k$/K5 in biological replicates ($n = 3$ for each peptide concentration). (**C**) Change in the area under the curve for IL-2 production (black) between the indicated mutant T cell hybridoma and wild type 2B4 hybridoma when activated with CHO cells expressing the cognate pMHC IE$^k$/K5 calculated based on (**B**). Change in TCR expression when compared to the wild type 2B4 TCR expression (MFI) is plotted in gray. (**D**) Plots of CD69 expression (MFI, Left) and ICOS expression (MFI, Right) on Jurkat cells expressing 2B4 mutant and wild-type TCR when activated with indicated concentrations of K5 peptide (CHO/I-E$^k$/K5) in biological replicates ($n = 3$ for each peptide concentration) for 16 h. Data information: In (**B, D**), data are presented as mean ± SD. In (**C**), area under the curves is represented as mean ± SEM. For (**D**), Appendix Table S7 shows non-parametric unpaired t-test values for wild type-mutant comparisons. Source data are available online for this figure.

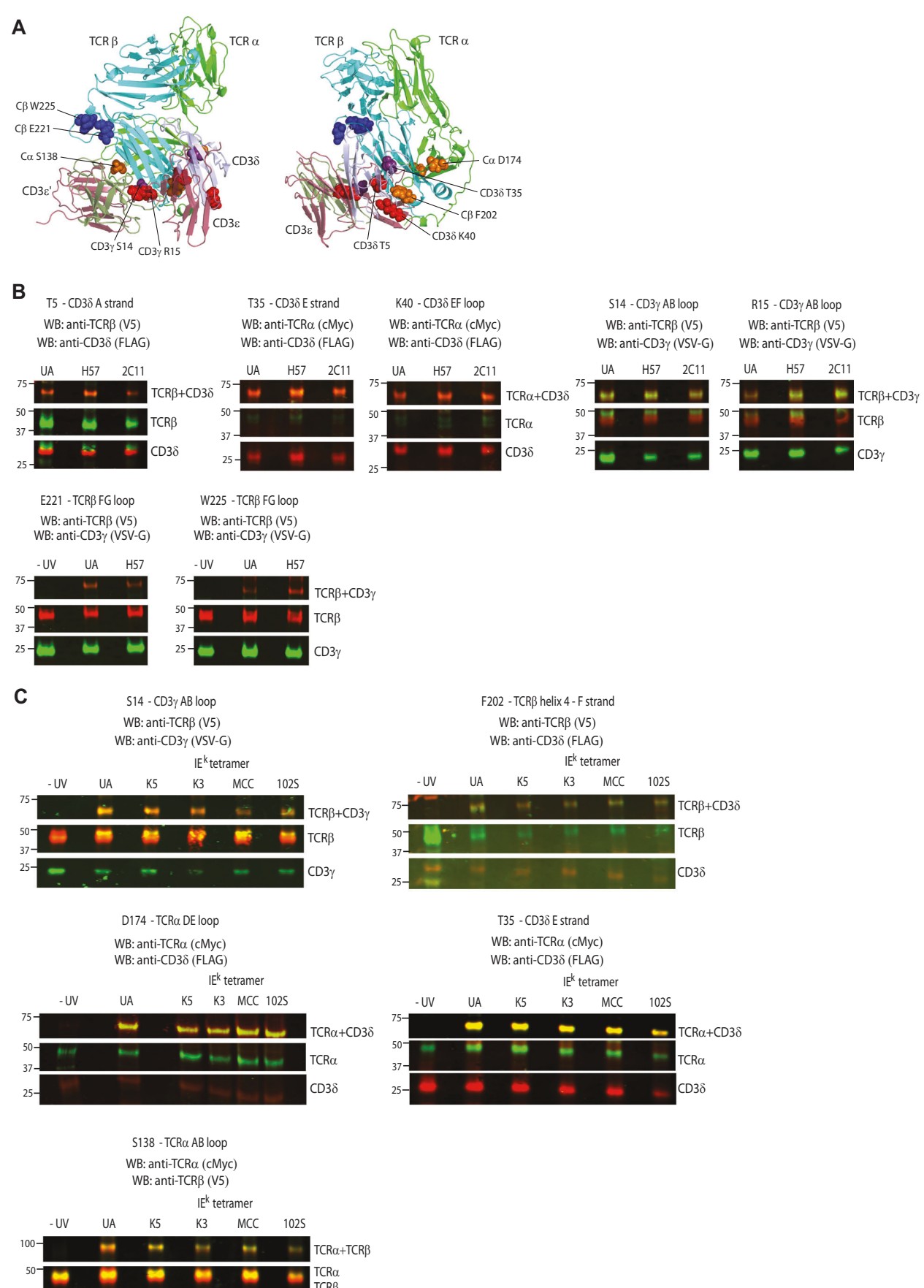

Figure 6. Crosslinking assay indicates minimal subunit reorientations or conformational changes upon activation.

(A) Locations of the mutated residues that crosslink to nearby subunits indicated on the mouse extracellular TCR-CD3 crosslink-guided structure. These mutants were tested for crosslink changes upon activation with surface-immobilized antibodies (H57/2C11) or soluble IE$^k$ tetramers. Crosslinks with H57 antibody activation are shown in blue (Cβ E221 and W225). Crosslinks with H57 and 2C11 activation are shown in red (δT5, δK40, and γR15). Crosslinks with IE$^k$ tetramer activation (K5, K3, MCC, and 102S) are shown in orange (αS138, αD174, and βF202). Crosslinks with H57, 2C11 antibody activation and IE$^k$ tetramer activation are shown in purple (δT35 and γS14). (B) Western blot analysis of the following crosslinking mutants: δT5, δT35, δK40, γS14, γR15, βE221, and βW225. TCRβ + CD3δ crosslinked band for T5 is present below 75 kDa. TCRα + CD3δ crosslinked bands for δT35 and δK40 are present below 75 kDa. TCRβ + CD3γ crosslinked bands for γS14 and γR15 are present below 75 kDa. In the forementioned blots all 3 conditions tested—not activated (UA), H57-coated, and 2C11-coated resulted in the same intensity of crosslinking bands. TCRβ + CD3γ crosslinked bands for βE221 and βW225 are present below 75 kDa for non activated and H57-coated conditions. (C) Western blot analysis of the following crosslinking mutants: γS14, βF202, αD174, δT35, and αS138. TCRβ + CD3γ crosslinked bands for γS14 are present below 75 kDa. TCRβ + CD3δ crosslinked bands for βF202 are present below 75 kDa. TCRα + CD3δ crosslinking bands for αD174 are present below 75 kDa. TCRα + CD3δ crosslinked bands for δT35 are present below 75 kDa. TCRα + TCRβ crosslinking band for αS138 is present below 100 kDa. Crosslinking bands were seen for all IE$^k$ tetramer activation conditions tested. Source data are available online for this figure.

(CHO-IE$^k$) loaded with K5 peptide, a superagonist peptide antigen for 2B4 TCR (Krogsgaard et al, 2003), and IL-2 production was quantified using an ELISA sandwich assay (Malecek et al, 2014; Natarajan et al, 2016).

Interestingly, the three mutants tested—βN236R, βS238K, and βE227F—produced lower amounts of IL-2 than the wild-type TCR (Fig. 5B,C), even though each mutant T cell clone exhibited similar TCR expression compared to the wild type (Fig. 5C). The αD169K mutant showed very poor surface expression of the TCR-CD3 complex and was excluded from the analysis. To independently verify these results, the same 2B4 TCR mutants were transfected into TCRβ deficient Jurkat cells (Ohashi et al, 1985), and T cell activation (CD69, ICOS) was quantified after co-culturing with CHO-IE$^k$ loaded with K5 peptide. All mutants tested—αD169K, βN236R, and βS238K—indicated lower levels of activation in comparison to the wild type, particularly at the highest concentration of K5 peptide tested (Fig. 5D; Appendix Table S7). The βE227F clone did not produce stable surface TCR-CD3 complex and was consequently removed from our analyses.

In addition, a similar in silico analysis using the reported cryo-EM structure of Dong et al (Dong et al, 2019) was performed, identifying mutations that are calculated biophysically to enhance TCR-CD3 interface stability, including G$_{182}$H (Cβ CC′ loop), S$_{216}$R (Cβ helix 4), E$_{257}$K and W$_{259}$Y (Cβ G strand) (Fig. EV2A; Appendix Table S6), all of which are conserved in both mouse and human constant domains. Subsequent IL-2 ELISA assays on T cell hybridomas expressing these mutants upon activation revealed diminished IL-2 production compared to the wild-type 2B4 TCR (Fig. EV2B,C). Similarly, transfection of these mutants into Jurkat cells followed by activation resulted in reduced CD69 expression, particularly at the highest K5 peptide concentrations (Fig. EV2D; Appendix Table S7). Overall, our in silico mutant design and activation assays consistently demonstrate that mutations intended to enhance or stabilize TCR-CD3 extracellular interactions had an opposite effect on T cell signaling.

In addition, we performed photo-crosslinking experiments on mutant-stabilized TCR-CD3 complexes (Fig. 5A) to identify whether cross-linking efficiencies improve. The triple mutant—βN236R/S238K/W242TAG, double mutants—βE227F/W242TAG and βE227F/W225TAG (TAG indicates the location for pApa substitution) had poor surface expression of the TCR-CD3 in comparison to the single mutants that cross-linked—βW242TAG and βW225TAG (Appendix Fig. S13A). Cross-linking analysis on βW225TAG and βE227F/W225TAG indicated crosslinks between

TCRβ and CD3γ in both instances even though βE227F/W225TAG complex expression was lower than βW225TAG expression (Appendix Fig. S13A,B). However, we weren't able to ascertain that mutant-stabilized TCR-CD3 complexes lead to improved crosslinking.

## Crosslinking assay indicates minimal subunit reorientations or conformational changes upon activation

Our photo-crosslinking assay provides an excellent opportunity to examine conformational changes and subunit reorientations in the TCR-CD3 complex upon activation, utilizing both surface-immobilized antibodies (Leo et al, 1987) and soluble pMHC tetramers. The alteration of existing crosslinks or the emergence of new crosslinks upon activation would indicate potential large conformational changes or movements of subunits relative to each other in the complex. While fast dynamical transient structural changes can be captured by photo-crosslinking corresponding to fast half-life of the reactive intermediate, ultra-fast dynamical changes (<ns–μs timescales) remain beyond its scope (Pham et al, 2013).

We performed a tetramer binding assay before and after photo-crosslinking for γS14 and δK40 mutant TCR-CD3 complexes to evaluate whether arresting conformational flexibility via cross-linking in the TCR-CD3 complex influences pMHC tetramer binding. Our results indicate no difference in pMHC tetramer binding between UV-crosslinked and non-crosslinked TCR-CD3 complexes (Appendix Fig. S14A,B) indicating that fixing the TCR-CD3 complex via photo-cross-links did not influence antigen binding. To analyze for conformational changes, we examined crosslinks for specific residues (Fig. 6A): αS138 (Cα AB loop) (Fig. 2D), αD174 (Cα DE loop) (Fig. 2B), βE221 and βW225 (Cβ FG loop) (Fig. 2E), βF202 (Cβ helix 4-F strand) (Fig. 2H), δT5 (CD3δ A strand) (Fig. 3B), δT35 (CD3δ E strand) (Fig. 3B), δK40 (CD3δ EF loop) (Fig. 3B), and S14 and R15 (CD3γ AB loop) (Fig. 3C). No changes were observed in the CD3 mutant crosslinks —TCRβ + CD3δ for δT5, TCRα + CD3δ for δT35 and δK40, and TCRβ + CD3γ for γS14 and γR15 after incubation with coated activating antibodies—H57 (anti-TCRβ) and 2C11 (anti-CD3ε) (Leo et al, 1987) (Fig. 6B). Similarly, no changes were observed in the Cβ FG loop mutant crosslinks—CD3γ + TCRβ for βE221 and βW225 after incubation with coated H57 antibody (Fig. 6B). Subsequently, we selected five crosslinks involving different

subunits of the TCR-CD3 complex and distributed throughout the complex—αS138 (Cα AB loop), αD174 (Cα DE loop), βF202 (Cβ helix 4-F strand), δT35 (CD3δ E strand), and γS14 (CD3γ AB loop) and probed for changes in crosslinks upon incubation with pMHC tetramers.

Using different variations of the 2B4 TCR antigen MCC—K5 (super agonist), K3 (agonist), 102S (weak agonist), each of which elicit different degrees of T cell stimulation (Krogsgaard et al, 2003), no changes were observed in the cross-links—TCRβ + TCRα for αS138, CD3δ + TCRα for αD174, CD3δ + TCRβ for βF202, TCRα + CD3δ for δT35, and TCRβ + CD3γ for γS14 after incubation with the different tetramers (Fig. 6C). Overall, the crosslinking assay indicates that antibody or tetramer activation does not elicit substantial stable conformational changes or subunit reorientations in the probed regions in the TCR-CD3 complex.

Recently, a cryo-EM structure of the unliganded TCR-CD3 complex in nanodiscs was reported by Notti et al (2023). This work reported the presence of 'open' and 'closed' conformations in the TCR-CD3 complex wherein 'detergent-based' cryo-EM structures were postulated to represent the 'open' activated complex and 'nano discs-based' structure represents the 'closed' native complex. A conformational change model for activation was proposed involving the TCRα-CD3δ regions and TCRα/β hinge regions. Therefore, we performed crosslinking analysis at the interface between TCRα-CD3δ in the absence and presence of pMHC tetramer binding which should provide more insights into conformational change model. We chose CD3δ residues N16 and E55 (corresponding human residues based on sequence alignment being $N_{38}$ and $D_{77}$), CD3δ K59 and TCRα R39 (both based on the crosslink structure) to test for crosslinking (Appendix Fig. S16A). We also used a control residue αS41 (that crosslinks to TCRβ) which is near the hinge region for our crosslinking analysis (Appendix Fig. S16A). Out of the mutants, CD3δ N16 and CD3δ K59 had very poor surface expression of the TCR-CD3 complex (Appendix Fig. S16B). αR39 mutant did not reveal any noticeable crosslinks. Both control αS41 mutant and CD3δ E55 mutant crosslinked with TCRβ and TCRα, respectively, although the CD3δ-TCRα crosslink for δE55 was much weaker in comparison to the TCRα-TCRβ crosslink for αS41 (Appendix Fig. S16C). However, in both instances, there was no change in the observed crosslinks upon binding with IE$^k$/MCC tetramer (Appendix Fig. S16C). Therefore, based on evidence from our study, we could not infer the occurrence of conformational change involving the TCRα-CD3δ region.

## CD3 tetramer assays reveal no major TCR-CD3 subunit reorganization upon activation

The accumulation of TCR-pMHC complexes at the T cell-APC interface, culminating in the formation of an immunological synapse along with other co-stimulatory molecules, is essential for T cell activation (Grakoui et al, 1999; Purtic et al, 2005). This clustering process could hypothetically involve CD3 subunits from adjacent TCR-CD3 complexes interacting with each other. To explore this possibility, we generated fluorophore (PE)-tagged CD3γε and CD3δε tetramers, comprising extracellular CD3γ or CD3δ subunits linked to extracellular CD3ε (Kim et al, 2000), and examined their ability to bind purified biotinylated TCR-CD3 complexes (Dong et al, 2019) coated on streptavidin beads

(DynaBeads) (Fig. 7A). The TCR-CD3 complex was immunoprecipitated either with CD3ε-specific 2C11 antibody or with 2B4-specific Vα 11.1, 11.2 antibody. Both purified complexes demonstrated binding to the IE$^k$/MCC tetramer, although tetramers exhibited weaker binding to the Vα 11.1, 11.2 immunoprecipitated complexes, likely due to competition with Vα antibody (Fig. 7B, Left). Notably, both immunoprecipitated complexes bound CD3γε and CD3δε tetramers (Fig. 7B, Center, Right) suggesting the presence of CD3 binding sites on the TCR-CD3 complex that are distinct from the Vα 11.1, 11.2 and 2C11 antibody epitopes.

To determine whether CD3 tetramers bind to the TCR or CD3 subunits in the TCR-CD3 complex, we assessed the CD3 tetramer binding ability of individual CD3γε, CD3δε or TCR heterodimers. Biotinylated versions of these proteins were generated recombinantly (Natarajan et al, 2016) and bound to streptavidin-coated beads. Both CD3 tetramers were able to bind individual CD3γε or CD3δε heterodimer-coated beads, indicating that CD3γε-CD3γε, CD3δε-CD3δε, and CD3γε-CD3δε interactions are all possible (Fig. 7C) concerning CD3 tetramer binding to CD3 subunits in the TCR-CD3 complex. Interestingly, CD3 tetramers also bound TCR-coated beads, albeit much weaklier, suggesting that CD3 tetramers specifically interact with the TCR-CD3 complex on the side harboring the CD3 subunits (Fig. 7C).

Based on this observation, we hypothesized that potential reorientations of CD3 subunits upon pMHC tetramer binding might alter the surface of the complex where CD3 tetramers bind, resulting in changes in CD3 tetramer binding (Fig. 7A). Indeed, we initially observed low binding of CD3γε and CD3δε tetramers to the complex prebound with different IE$^k$ tetramers—K3, MCC, and K5, with no significant difference among them (Appendix Fig. S17). However, upon using increased amounts of CD3 tetramers (350 ng/mL as compared to 50 ng/uL used previously), we discovered that the binding of CD3γε (Fig. 7D,E Left) and CD3δε (Fig. 7D,E Right) tetramers to the complex prebound with IE$^k$/K5 or IE$^k$/K3 or IE$^k$/MCC tetramers was comparable to CD3 tetramer binding to the IE$^k$ tetramer-free complex. These findings suggest that the earlier reduction in CD3 tetramer binding (Appendix Fig. S17) might be attributed to steric hindrance between the CD3 and IE$^k$ tetramers present, which can be overcome by using higher amounts of CD3 tetramers. In summary, the CD3 tetramer assay indicates that pMHC binding to the TCR-CD3 complex does not induce substantial subunit movements within the complex.

## Discussion

The bio-translational significance of molecular observations can be hindered by the structural accuracy of 3D receptor models, constituting a critical aspect in bridging the gap between in vitro molecular insights and clinical relevance. Established structural biology techniques such as X-ray crystallography, NMR, cryo-EM and molecular modeling provide detailed and tangible atomistic information about protein structures and interactions in vitro. However, challenges arise when aiming to extrapolate these findings to the complex, in situ states within cellular environments. To address this, site-specific crosslinking, together with the incorporation of unnatural amino acids and computational analysis, offers a robust alternative. This approach not only provides atomistic details but also offers species-specific

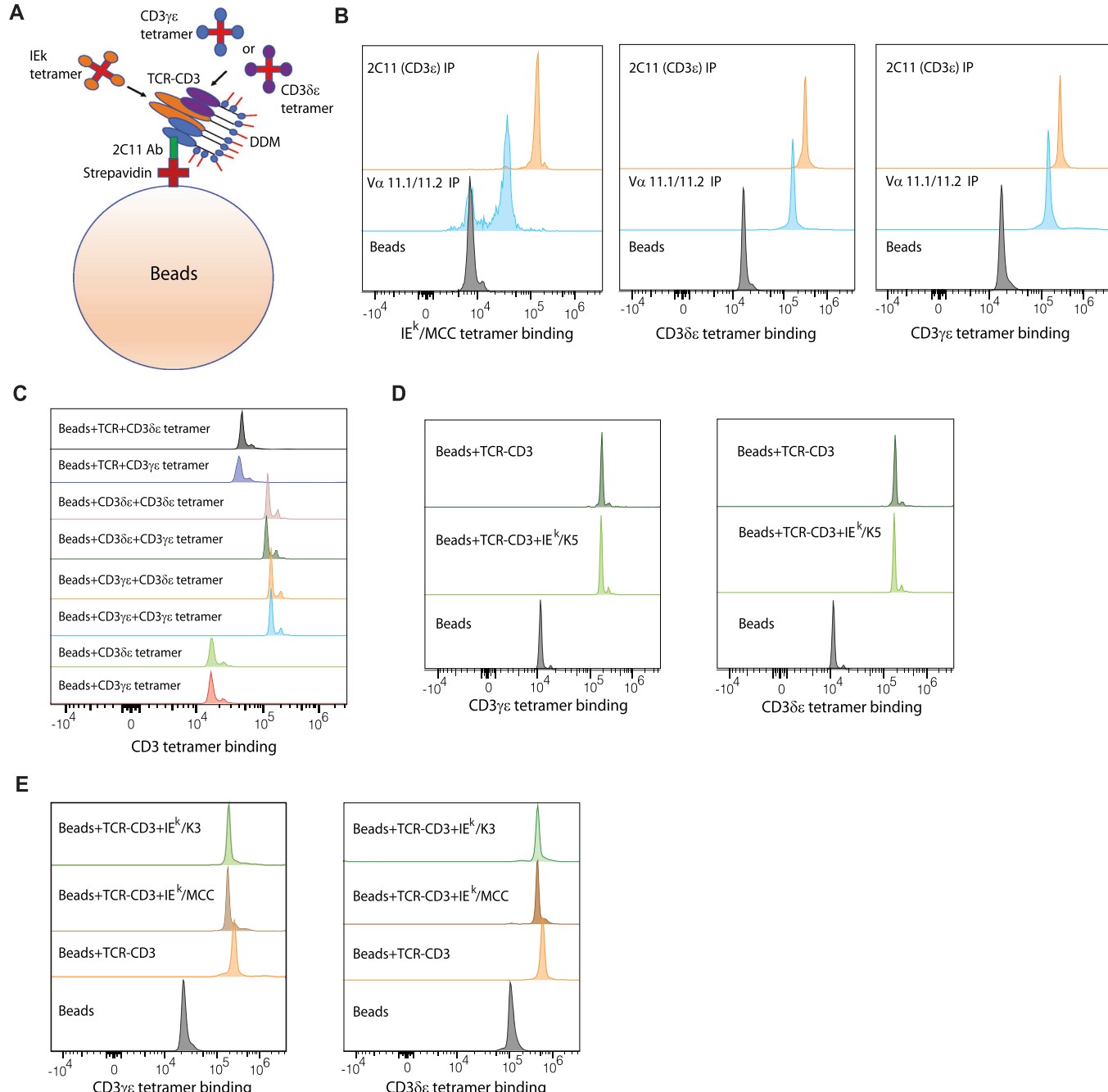

**Figure 7. CD3 tetramer assays reveal no major TCR-CD3 subunits reorganization upon activation.**

(A) Schematic illustration of the CD3 tetramer assay used to identify subunit orientations upon pMHC tetramer binding. (B) Histograms of IE$^k$/MCC tetramer-APC binding (Left), CD3γε tetramer-PE binding (Center), and CD3δε tetramer-PE binding (Right) to 2C11 immunoprecipitated complex (orange), Vα 11.1, 11.2 immunoprecipitated complex (cyan) and control Dynabeads (black). (C) Histogram of beads coated with TCR bound with CD3δε tetramer-PE (black) and CD3γε tetramer-PE (blue), beads coated with CD3δε bound with CD3δε tetramer-PE (brown) and CD3γε tetramer-PE (dark green), beads coated with CD3γε bound with CD3δε tetramer-PE (orange) and CD3γε tetramer-PE (cyan), control beads bound with CD3δε tetramer-PE (green) and CD3γε tetramer-PE (red). (D) Left, Histogram of beads coated with the complex (dark green), beads with complex with bound IE$^k$/K5 tetramer (green) and control beads (black), each stained with 350 ng/µL CD3γε tetramer-PE. Right, Histogram of beads coated with the complex (dark green), beads with complex with bound IE$^k$/K5 tetramer (green), and control beads (black), each stained with 350 ng/µL CD3δε tetramer-PE. (E) Left, Histogram of beads with complex with bound IE$^k$/K3 tetramer (green), beads with complex with bound IE$^k$/MCC tetramer (brown), beads coated with the complex (orange) and control beads (black), each stained with 350 ng/µL CD3γε tetramer-PE. Right, Histogram of beads with complex with bound IE$^k$/K3 tetramer (green), beads with complex with bound IE$^k$/MCC tetramer (brown), beads coated with the complex (orange), and control beads (black), each stained with 350 ng/µL CD3δε tetramer-PE. Source data are available online for this figure.

information on structures while requiring minimal protein quantities. Importantly, it enables the exploration of in situ states within complex cellular environment, contributing to a more comprehensive understanding of molecular phenomena (Coin, 2018; Grunbeck et al, 2011; Valentin-Hansen et al, 2014).

Photo-cross-linking is well-suited for investigating weak protein-protein interactions, a characteristic prominent in the TCR-CD3 complex. The assembly of this complex is primarily driven through interactions among transmembrane subunits (Call et al, 2002; Call and Wucherpfennig, 2005; Xu et al, 2006), with weak extracellular interactions (He et al, 2015; Natarajan et al, 2016). To our knowledge, our study is the first to successfully employ photo-crosslinking, subsequently providing a native structural model for an immune protein receptor complex. The observed overall similarity between cryo-EM structures (Chen et al, 2022; Dong et al, 2019; Saotome et al, 2023; Susac et al, 2022) and the crosslinking model validates the photo-crosslinking-docking technique as an attractive option for structural and in situ analysis of protein complexes. This approach can be extended to study other immune protein complexes in their native states, including the γδTCR-CD3 complex, B cell receptor complex, CD19/CD21 coreceptor complex, checkpoint inhibitor-ligand complexes such as LAG3-FGL-1, TIM-3-CEACAM1 as well as other multi-unit membrane receptors.

For the incorporation of unnatural amino acid pAzpa and subsequent crosslinking, we conducted co-transfections of tRNA-aaRS and mouse TCR and CD3 plasmids, incorporating amber mutations at specific sites in TCR or CD3, in 293T cells. A total of 48 mutants were expressed and UV crosslinked, which yielded 15 specific TCR-CD3 subunit crosslinks. Noteworthy crosslinks included residues in the Cα DE loop crosslinking with CD3δ, the Cβ CC′ loop crosslinking to CD3δ, the Cβ FG loop crosslinking to CD3γ, the Cβ G strand crosslinking to CD3ε and the Cβ helix 4-F strand crosslinking to CD3δ and CD3ε (Fig. 4A). In addition, the CD3δ A strand exhibited crosslinks to TCRβ, CD3δ E strand, the EF loop crosslinks to TCRα, and the CD3γ AB loop crosslinked to TCRβ (Fig. 4A). Employing these specific crosslinks as constraints in a biophysical conformational search, we successfully visualized an in situ cell-surface model of the mouse TCR-CD3 complex (Fernandez-Recio et al, 2003).

Prior investigations into CD3 interacting sites on the TCR employed diverse methodologies, including mutagenesis (Kuhns and Davis, 2012), modeling (Sun et al, 2004), and NMR spectroscopy (He et al, 2015; Natarajan et al, 2016). These studies have yielded conflicting models, with some proposing one-sided interactions (He et al, 2015; Kuhns et al, 2010) and others suggesting two-sided models (Birnbaum et al, 2014; Natarajan et al, 2016; Sun et al, 2004). In light of this, we compared our crosslink-guided model with a recent cryo-EM structure (Dong et al, 2019), which depicted a one-sided arrangement of CD3 subunits around the TCR in the resting state.

Our analysis revealed that the CD3 arrangement from the crosslink-guided model is largely consistent with the cryo-EM structure, particularly regarding the gross locations of the CD3-TCR interfaces within the complexes (Fig. 4B). However, a notable difference was observed in the positioning of the Cβ FG loop, which, in the crosslink-guided model is situated above and in-between CD3γ and CD3ε′, unlike the cryo-EM structures (Chen et al, 2022; Dong et al, 2019; Saotome et al, 2023; Susac et al, 2022),

where the FG loop is positioned above CD3ε′ in the CD3γε heterodimer. A potential explanation for this discrepancy could be species-specific variations in the amino acid composition of the electrostatic surfaces of the mouse CD3 subunits used in our study compared to the human CD3 subunits in the cryo-EM study, despite the high conservation of TCR interface residues (Appendix Fig. S11). Furthermore, our crosslinking assay supports the notion that the cell-surface conformation of the TCR-CD3 complex closely mirrors the conformation of the 'detergent-purified' complex (Fig. EV1B) as evidenced by the consistent presence of the same crosslinks under both conditions.

Our current crosslink-guided model deviates significantly from our previously proposed NMR-based model (Natarajan et al, 2016), which relied on CSP data that indicated changes in peak intensities or peak shifts in the TCR upon addition of CD3γε and CD3δε. These sites include Cβ helix 3, Cβ helix 4-F strand, Cβ FG loop, Cα F strand, Cα C strand, and Cα tail (Natarajan et al, 2016). In our earlier work, the top-scoring TCR-CD3γε and TCR-CD3δε docking models suggested CD3 heterodimer interactions occurring on opposite sides of the TCR (Natarajan et al, 2016). A potential explanation for this discrepancy could be the absence of membrane, CPs and TMs in the soluble protein domains used in the NMR study. This absence may permit orientation of the CD3 molecules away from the membrane, rather than in proximity. Further, CSP analysis cannot distinguish between 'direct' and 'indirect' contact between two interacting proteins because spectral changes can occur in sites that are away from interacting partner due to changes in overall complex tumbling time or allosteric effects (Zuiderweg, 2002). A more concrete method such as NMR cross-saturation analysis (Takahashi et al, 2000) was not applicable here due to weak affinity between extracellular TCR and CD3 subunits (He et al, 2015). Further, IL-2 production upon activation of Cβ helix 3 mutants identified in the NMR study indicated a loss of >50% compared to the wild-type TCR (Natarajan et al, 2016). In addition, other studies have implicated some of the same TCR sites (Cβ helix 3 and helix 4) as CD3 interaction regions, along with Cβ FG loop and Cα AB loop, all of which are involved in allosteric interactions upon antigen ligation (He et al, 2015; Natarajan et al, 2017; Rangarajan et al, 2018). Consequently, the roles of Cβ helix 3 and other sites such as the Cα F and C strands in TCR signaling remains unresolved, as they are not directly implicated in CD3 interaction.

Significantly, in this study, we explored the potential of enhancement of TCR-CD3 extracellular interactions through in silico mutagenesis, utilizing both the crosslink-guided model and cryo-EM structure (Dong et al, 2019). Despite efforts to strengthen the TCR-CD3 complex interface, we observed no improvement in T cell functionality, as evidenced by IL-2 production, CD69 and ICOS expression. This suggests that T cell signaling operates at a highly dynamic macromolecular level, necessitating the interplay between the "on" and "off" states of CD3 on the TCR. It also suggests that strengthening the complex interface could lead to a loss of flexibility or fluidity in overall T-cell synapse, which is crucial for effective T cell signaling (van Eerden, 2023; Lanz et al, 2021; Rangarajan et al, 2018). Ongoing studies are actively exploring strategies to optimize TCR-CD3 extracellular interactions with the aim of enhancing T cell signaling. Based on this observation of signal dampening, it is important to acknowledge that mutating certain wild-type residues to pAzpa could indeed alter the sidechain dynamics involving these amino acids that could

impact signaling. However, we could only verify the surface expressions and antigen binding abilities of these mutant complexes and not their signaling capacities in our current experimental setup.

Based on insights gained from earlier NMR and cryo-EM studies (Arechaga et al, 2010; Birnbaum et al, 2014; Dong et al, 2019; He et al, 2015; Natarajan et al, 2016), we posited that the extracellular segment of the TCR-CD3 complex may adopt multiple biologically-relevant conformations, particularly in its activated state. Such a conformational switch is not uncommon in the TCR-CD3 transmembrane space, as evidenced by the existence of CD3ζ juxtamembrane regions in open and closed conformations (Lee et al, 2015), reduction of CD3ζ cohesion from αβTCR upon pMHC binding (Lanz et al, 2021), and TCRβ transitioning between inactive and active conformations in response to cholesterol binding and unbinding (Swamy et al, 2016).

Our crosslinking assay offered some evidence supporting the requirement of cholesterol for proper TCR-CD3 complex assembly and supports prior observations that cholesterol maintains the TCR-CD3 complex in 'resting conformation' (Chen et al, 2022; Molnar et al, 2012; Swamy et al, 2016). Utilizing the photo-crosslinking assay in conjunction with an antibody and antigen activation system, along with CD3 tetramer-based assays, we discerned that significant subunit movements or reorientations within the extracellular regions of the TCR-CD3 complex do not occur upon activation. This finding aligns with recent cryo-EM structures indicating minimal structural changes in the complex in presence of a single pMHC (Saotome et al, 2023; Susac et al, 2022). Also, we did not identify evidences for the 'conformational change' model (Notti et al, 2023) involving TCRα-CD3δ interface region. But this evidence of lack of conformational change is not concrete because two out of four mutants tested resulted in poor complex surface expression and only mutant produced noticeable crosslink (Appendix Fig. S16C). Although the TCR-CD3 crosslinks used in our activation analysis (Fig. 6A) are distributed throughout the interface between the subunits of the TCR-CD3 complex, identifying more crosslinks between the TCR and CD3 subunits and exploring potential changes in these crosslinks could provide clearcut evidence of conformational changes or lack thereof. However, it is crucial to note that our observation does not rule out the possibility of dynamic or flexibility changes occurring within individual TCR-CD3 subunit components, potentially influencing signaling (Lanz et al, 2021; Natarajan et al, 2017; Rangarajan et al, 2018). Previous NMR analyses and MD simulations suggested a loss of flexibility in Cβ helix 3, Cβ FG loop, and Cα AB loop upon pMHC binding to the TCR (Rangarajan et al, 2018). Our collective data lead us to hypothesize that pMHC binding to the TCR induces dynamic or flexibility changes without causing substantial conformational changes or subunit reorientations in the constant domains of the TCR and extracellular CD3 subunits but still inducing notable conformational changes in the transmembrane regions of the complex, potentially leading to CD3ζ separation and subsequent initiation of signaling events (Chen et al, 2022; Minguet et al, 2007; Swamy et al, 2016).

The formation of catch-bonds during the interaction between TCR and agonist pMHC, under the influence of mechanical force across T cell-APC interface, has been proposed to induce conformation changes in the TCR-CD3 complex (Das et al, 2015;

Liu et al, 2014). However, our current experimental setup did not allow us to explore this phenomenon. Moreover, it is important to acknowledge that HEK293T cells, utilized in our study, lack other essential T cell signaling components and co-receptors that could potentially play a pivotal role in eliciting significant structural changes within the TCR-CD3 complex upon activation. In the future, conducting crosslinking assays in T cell systems like Jurkat cells (Ohashi et al, 1985) or T cell hybridoma (Letourneur and Malissen, 1989) lacking native TCRs in which other signaling components of a T cell are intact, where the functional capacities of crosslinker substituted TCR-CD3 complexes can be quantified. These systems are particularly insightful in the context of antigen-presenting cell systems, transient interactions between different adapter and effector molecules using the experimental framework we have outlined. Such additional systems could offer valuable insights into the nuanced interplay of mechanical forces and cellular components influencing the dynamic structural alterations within the TCR-CD3 complex during T cell activation.

## Methods

**Reagents and tools table**

| Reagent/Resource | Reference or Source | Identifier or Catalog Number |
|---|---|---|
| **Experimental models** | | |
| HEK 293T (*H. sapiens*) | ATCC | CRL-3216 |
| 58-/- hybridoma (*M. musculus*) | (Letourneur and Malissen, 1989) | From David Kranz, University of Illinois |
| J.RT3-T3.5 (*H. sapiens*) | ATCC | TIB-153 |
| CHO (*C. griseus*) | (Natarajan et al, 2016) | From Mark. M. Davis, Stanford University |
| Platinum-GP | Cell Biolabs | RV-103 |
| One Shot BL21 (DE3) | Invitrogen | C600003 |
| **Oligonucleotides and other sequence-based reagents** | | |
| PU6-pAzpa | (Wang et al, 2014) | Modified from PU6-pBpa, from Peter G. Schultz, Scripps Research Institute |
| pCDNA3.1/Zeo (+) | Life Technologies (ThermoFisher) | V86020 |
| pMXSIB | Cell Biolabs | RTV-010 |
| pMX-Puro | Cell Biolabs | RTV-012 |
| pVSV-G | Addgene | 138479 |
| **Antibodies** | | |
| APC anti-TCRβ (clone H57-597) | eBioscience | 17-5961-82 |
| PE anti-CD3ε (clone 145-2C11) | eBioscience | 12-0031-82 |
| Biotin mouse anti-CD3ε (clone 145-2C11) | BD Pharmingen | 553060 |
| The c-Myc tag antibody, mouse | Genscript | A00704-100 |
| V5 tag antibody, rabbit | Genscript | A00623-40 |

| Reagent/Resource | Reference or Source | Identifier or Catalog Number |
|---|---|---|
| FLAG tag antibody, rabbit | Genscript | A00170-40 |
| The V5 tag antibody, mouse | Genscript | A01724-100 |
| The FLAG tag antibody, mouse | Genscript | A00187-100 |
| Anti-VSV-G antibody, mouse | Abcam | Ab50549 |
| MonoRab HA tag antibody, rabbit | Genscript | A01963 |
| The HA tag antibody, mouse | Genscript | A01244-100 |
| IRDye 680LT-conjugated donkey anti-rabbit IgG (H + L) | LI-COR | 926-68023 |
| IRDye 800CW-conjugated donkey anti-mouse IgG (H + L) | LI-COR | 926-32212 |
| IRDye 680RD-conjugated donkey anti-mouse IgG (H + L) | LI-COR | 926-68072 |
| IRDye 800CW-conjugated Goat anti-rabbit IgG (H + L) | LI-COR | 926-32211 |
| Biotin Rat Anti-mouse Vα 11.1, 11.2 [b,d] TCR antibody | BD Pharmingen | 553221 |
| CD3 monoclonal antibody (OKT3) PE | eBioscience | 12-0037-42 |
| Purified NA/LE Hamster anti-mouse CD3ε, clone 145-2C11 | BD Pharmingen | 553057 |
| Purified NA/LE Hamster anti-mouse TCRβ, clone H57-597 | BD Pharmingen | 553166 |
| **Chemical, enzymes and reagents** | | |
| Not1 | New England Biolabs | R0189S |
| Xho1 | New England Biolabs | R0146S |
| Quikchange Mutagenesis Kit | Agilent | 200519 |
| DMEM | Gibco | 10-017-CV |
| FBS | Sera Prime | F11713-500 |
| Sodium pyruvate | Gibco | 11360-070 |
| MEM NEAA | Gibco | 11140-050 |
| Gluatmax-1 | Gibco | 35050-061 |
| Pen-strep | Gibco | 15140-122 |
| β-mercaptoethanol | Gibco | 21985-023 |
| p-azido-phenylalanine | Chem-Impex Int'l | 06162 |
| Xfect transfection kit | TaKaRa | 631318 |
| Collagen plates | Gibco | A11428-01 |
| PBS | Corning | 21-040-CV |

| Reagent/Resource | Reference or Source | Identifier or Catalog Number |
|---|---|---|
| HBSS | Corning | 14025-092 |
| Sodium azide | ThermoFisher | S2271-100 |
| IGEPAL CA-630 | Sigma | I3021-50ML |
| Dynabeads M-280 streptavidin | Invitrogen, ThermoFisher | 11206D |
| Nitrocellulose membrane | ThermoFisher | 88024 |
| Complete protease cocktail inhibitor | Roche | 11697498001 |
| n-dodecyl-β-D-maltopyranoside (DDM) | Millipore Sigma | D4641-5G |
| Cholesteryl hemisuccinate tris salt | Millipore Sigma | C6013-5G |
| Digitonin | Millipore Sigma | D141-500MG |
| PMSF | MP medicals | 195381 |
| HEPES | Sigma | H3375-500G |
| Sodium Chloride | ThermoFisher | BP358-10 |
| Lipofectamine 2000 | Invitrogen | 11668019 |
| Opti-MEM | Invitrogen | 31985070 |
| Protamine Sulfate | MP Biomedicals | 102752 |
| BirA500 protein ligase kit | Avidity LLC | BIRA500 |
| Streptavidin/Biotin blocking kit | Vector Laboratories | SP-2002 |
| Streptavidin PE | eBioscience | 12-4317-87 |
| **Other** | | |
| FACSCalibur | BD | NA |
| Cytek Aurora 5 | Cytek Bio | NA |
| Odyssey CLx | LI-COR | NA |
| **Softwares** | | |
| Flowjo | BD | 10.8.1 |
| Image Studio Lite | LI-COR | 5.2.5 |
| Prism | Graphpad | 10.1.1 |
| ICM-Pro software | Molsoft LLC | |

## Plasmid construction

The tRNA synthetase for recognition of pBpa in PU6-pBpa plasmid was replaced with tRNA synthetase for pAzpa to create PU6-pAzpa plasmid, both generous gifts from Peter G. Schultz, Scripps Research Institute (Wang et al, 2014). This PU6-pAzpa plasmid contains mutant E.coli tyrosyl-tRNA synthetase (EcTyrRS), one copy of *B. stearothermophilus* tRNAs (BstRNA) and human U6 small nuclear promoter (U6) (Wang et al, 2014). cDNA encoding mouse TCR 2B4 α (with c-Myc-tag) and β (with V5-tag) sequences with 2A sequence linking each other were cloned into pCDNA3.1/Zeo(+) vector (Life Technologies) using Not1 and Xho1 restriction enzymes. Similarly, cDNA encoding mouse CD3δ (with FLAG-tag), mouse CD3γ (with VSV-G-tag), mouse CD3ε (with HA-tag) and CD3ζ interconnected with 2A sequence were

cloned in pCDNA3.1/Zeo(+) vector using Not1 and Xho1 restriction enzymes. Amber (TAG) codons were introduced site-specifically in the 2B4 TCR and CD3 plasmid using Quikchange mutagenesis kit (Agilent).

## Transfections into HEK293T cells

HEK293T cells (ATCC) were cultured in Dulbecco's modified Eagle medium (DMEM) media, supplemented with 10% FBS, sodium pyruvate, non-essential amino acids, glutaMAX-1, penicillin-streptomycin and β-mercaptoethanol and grown at 37 °C, 5% $CO_2$ to 80% confluency in a collagen-coated 6-well plate before transfections. To incorporate unnatural amino acids, pAzpa (p-azido-phenylalanine), into predetermined sites on the TCR and CD3 extracellular regions (Appendix Tables S1 and S2), plasmid expressing amber suppressor tRNA-aminoacyl-tRNAsynthetase (tRNA-aaRS) – PU6-pAzpa was co-transfected with plasmids expressing full-lengths mutant 2B4 TCR and CD3 subunits using Xfect transfection kit (TaKaRa). For incorporating pAzpa into TCR sites, as optimized previously, 7.5 μg of TCR:CD3 in 8:1 ratio and 2.5 μg of PU6-pAzpa plasmids were co-transfected into HEK293T cells (Wang et al, 2014). For incorporating pAzpa into CD3 sites, 7.5 μg of TCR:CD3 in 4:1 ratio and 2.5 μg of PU6-pAzpa plasmids were co-transfected into HEK293T cells. After 4 h of culture at 37 °C, 5% $CO_2$, the media was replaced with fresh DMEM media containing 1 mM pAzpa (Chem-Implex International) and cultured for 48 h.

## Flow cytometry analysis

After 48 h of culture, the cells were harvested and washed in FACS buffer (PBS + 2% FBS). A small portion of the cells were treated with allophycocyanin (APC) anti-TCRβ (clone H57-597, eBioscience) and phycoerythrin (PE) anti-CD3ε (clone 145-2C11, eBioscience) in FACS buffer for 30 min. Subsequently, the samples were analyzed for TCRβ and CD3ε expression in FACSCalibur (BD Biosciences) and data was analyzed using FlowJo (ver 10.8.1).

## Photo-crosslinking, immunoprecipitation, and western blotting

Cells were photo-crosslinked by exposing them to 360 nm UV light source for 30 min on ice. For IE$^k$/MCC-APC (NIH Tetramer Core Facility) tetramer staining to test viability of the assembled TCR-CD3 complex, 100,000 of the UV irradiated cells were washed with FBS buffer and incubated with 100 ng/μL tetramer in 50 μL at RT for 1 h. Subsequently, the samples were analyzed for tetramer binding in Cytek Aurora (Cytek Bio) and data was analyzed using FlowJo (ver 10.8.1). Following that, the cells were washed in antibody buffer - Hank's balanced salt solution/2% FBS/0.05% (m/v) sodium azide. The cells were treated with 25 μg/mL biotinylated mouse anti-CD3ε (clone 145-2C11, BD Pharmingen) for 30 min and washed in 1X TBS (Tris-buffered saline). The cells were lysed in TBS/1% (v/v) IGEPAL-630 (Sigma) containing 1X Complete protease cocktail inhibitors (Roche). The TCR-CD3 complex was purified from the lysate using Dynabeads M-280 streptavidin (Invitrogen). The beads were subsequently washed with 1X TBS and boiled with SDS-PAGE reducing buffer with β-mercaptoethanol. The subunits were resolved by SDS-PAGE

electrophoresis and transferred to nitrocellulose membranes (ThermoFisher). For Western blot analysis, the following pairs of primary antibodies were used: (1) TCRα-TCRβ cross-linking: mouse anti-cMyc (Genscript) and rabbit anti-V5 (Genscript); (2) TCRα-CD3δ cross-linking: mouse anti-cMyc and rabbit anti-FLAG (Genscript); (3) TCRβ-CD3δ cross-linking: mouse anti-V5 and rabbit anti-FLAG (Genscript); (4) TCRβ-CD3δ cross-linking: rabbit anti-V5 and mouse anti-FLAG; (5) TCRβ-CD3γ cross-linking: rabbit anti-V5 (Genscript) and mouse anti-VSV-G (Abcam); (6) TCRβ-CD3ε cross-linking: rabbit anti-V5 and mouse anti-HA (Genscript); (7) TCRβ-CD3ε cross-linking: mouse anti-V5 and rabbit anti-HA (Genscript); (8) TCRα-CD3γ cross-linking: rabbit anti-cMyc and mouse anti-VSV-G. The following secondary antibodies were used for detection: IRDye 680LT-conjugated donkey anti-rabbit IgG (H + L) (LI-COR), IRDye 800CW-conjugated donkey anti-mouse IgG (H + L) (LI-COR), IRDye 680RD-conjugated donkey anti-mouse IgG (H + L) (LI-COR) and IRDye 800CW-conjugated Goat anti-rabbit IgG (H + L) (LI-COR). Images were collected using LI-COR Odyssey and analyzed using Image Studio Lite (LI-COR, ver 5.2.5).

## TCR-CD3 complex purification from HEK293T cells

After 48 h of culture, the TCR-CD3 expressing cells were harvested and washed in antibody buffer. The cells were treated with 25 μg/mL biotinylated mouse anti-CD3ε (clone 145-2C11, BD Pharmingen) or biotinylated anti-mouse Vα 11.1, 11.2 (BD Pharmingen) for 30 min at RT. Following that, the cells were washed in 25 mM HEPES pH 7.5, 150 mM NaCl and 1 mM PMSF. Then, the cell membrane was lysed in 25 mM HEPES pH 7.5, 150 mM NaCl, 1% n-dodecyl-β-D-maltopyranoside (DDM), 0.2% cholesteryl hemi-succinate tris salt and 1 mM PMSF for 30 min at RT (Dong et al, 2019). After removal of the debris via centrifugation, the TCR-CD3 complex was purified from the lysate using Dynabeads M-280 streptavidin (Invitrogen). The Dynabeads coated complex was then suspended in 25 mM HEPES pH 7.5, 150 mM NaCl, 0.06% digitonin (Dong et al, 2019) and used for subsequent experiments. The resuspended Dynabeads resin was photo-crosslinked by exposing them to 360 nm UV light source for 30 min on ice. The beads were subsequently washed and boiled with SDS-PAGE reducing buffer with β-mercaptoethanol. The subunits were resolved by SDS-PAGE electrophoresis and transferred to nitro-cellulose membranes (ThermoFisher) for Western blot analysis.

## Functional analysis with mutant T cell hybridoma

Mouse 58-/- T cell hybridoma cells (Letourneur and Malissen, 1989), which expresses mouse CD3 but not TCRαβ (from David Kranz, University of Illinois) and Chinese hamster ovary (CHO) cells expressing I-E$^k$ (Krogsgaard et al, 2005) (generous gift from Mark M. Davis, Stanford University) were cultured in RPMI 1640 medium and DMEM, respectively, supplemented with 10% FBS, sodium pyruvate, non-essential amino acids, glutaMAX-1, penicillin-streptomycin and β-mercaptoethanol. The mutant mouse 2B4 TCR constructs were generated by PCR using overlapping primers containing the mutant sequences and cloned into the pMXSIB vector (Natarajan et al, 2016). Retroviral transductions of the hybridoma cells were done as described previously (Zhong et al, 2010). Transduced cells were stained with PE anti-CD3ε (clone

145-2C11) and APC anti-TCRβ (clone H57-597) antibodies. Transduced cells were sorted, expanded for 6 days, quantified for TCRβ/CD3ε expression, and prepared for the cytokine assay. $1 \times 10^4$ CHO-IE$^k$ cells were loaded with different concentrations of a variant of moth cytochrome c (K5) (Krogsgaard et al, 2005) peptide and incubated with $1 \times 10^4$ T cell hybridoma clones (wild type and mutants) in triplicates for 16 h at 37 °C, 5% $CO_2$. A standard ELISA sandwich was used to quantify cytokine IL-2 production (Malecek et al, 2013). The area under the curve for wild type and mutant IL-2 production, a cumulative response measure, was calculated after non-linear fitting using Prism (GraphPad software).

## Retroviral transduction in Jurkat cells and T cell activation

To quantify how TCR mutants impact T cell functionality in human T cells, we employed retroviral infection in TCRβ negative Jurkat cell line (J.RT3-T3.5, ATCC). The retroviral vector PMXs-puro-TCRs were generated by inserting both WT 2B4 TCR and mutant TCRs into PMXs vector (Cell Biolabs) backbone. Then PMXs plasmids and the envelope plasmid (pVSV-G, Addgene) were transfected into platinum-GP (plate GP, Cell Biolabs) cell line using lipofectamine (lipofectamines 2000; Invitrogen) and incubated with glucose-free Opti-MEM (Invitrogen). After culturing for 48 h, the supernatant was collected and filtered through a 0.45-μm-pore filter. Jurkat cell pellets were resuspended in virus supernatant at $10^6$ cells per 1 ml and 10 μg/ml of protamine sulfate. 1 ml of cells suspension was added to each well of a 24-well plate. Cell containing plates were centrifuged for 90 min at $2000 \times g$ at 32 °C with no brake. 1 ml of complete medium was added to the plate after centrifugation and incubated at 37 °C. At day 6, the TCR's transduced Jurkat cells were sorted for both TCR and CD3 expression using TCRβ H57 and OKT3 CD3ε antibodies. Two additional sorting steps for TCR and CD3 expression were performed within next 14 days to ensure continuous expression. For the functional studies, $5 \times 10^4$ CHO-IE$^k$ cells were loaded with different concentrations of K5 peptide (Krogsgaard et al, 2005) and incubated with $5 \times 10^4$ Jurkat clones (wild type and mutants) in triplicates for 16 h at 37 °C, 5% $CO_2$. After activation, cells were washed and stained with TCRβ (H57-797, eBioscience) antibody and activation markers, CD69 and ICOS for 30 min at room temperature. Cells were then washed twice. After the final wash, cells were resuspended in 100 μl FACS buffer (PBS + 2% FBS). Data collection was performed on Cytek Aurora 5 laser spectral flow cytometer and data was analyzed using FlowJo (ver 10.8.1).

## TCR-CD3 complex antibody and pMHC tetramer activation and crosslinking

For antibody experiments, anti-TCRβ (clone H57-597, BD Pharmingen) and anti-CD3ε (clone 145-2C11, BD Pharmingen) antibodies were coated at 5 μg/mL on 12-well plates overnight at 4 °C. $1 \times 10^6$ 293T cells expressing TCR-CD3 complexes were added to the coated plates and incubated for 2 h at RT. Photo-crosslinking, immunoprecipitation and Western blotting were performed as mentioned above. For pMHC tetramer activation experiments, IE$^k$/K5, IE$^k$/K3, IE$^k$/MCC, and IE$^k$/102S (Krogsgaard et al, 2003) were obtained from NIH Tetramer Core Facility. $1 \times 10^6$ 293T cells expressing TCR-CD3 complexes were incubated with

100 μg/mL tetramer (50 μL) for 2 h at RT. Appendix Fig. S15A,B illustrates IE$^k$/MCC-APC tetramer binding at 100 μg/mL concentration for $1 \times 10^6$ cells for select crosslinking mutants and IL-2 production of 2B4 wild-type hybridoma with 100 ng/μL tetramer concentration. Photo-crosslinking, immunoprecipitation and Western blotting were performed as mentioned above.

## Recombinant proteins, CD3 tetramer generation, and staining experiments

A biotinylation sequence (GLNDIFEAQKIEWHE) was added to N-terminus of extracellular constructs of 2B4 TCRα and C-terminus of human CD3γε and CD3δε subunits by PCR (Beckett et al, 1999). 2B4 TCR construct consists of 2B4 variable region and human LC13 constant region to aid in protein expression (Natarajan et al, 2016). Proteins were expressed in One Shot™ E. coli BL21 (DE3) (Invitrogen) as insoluble inclusion bodies, refolded and purified as previously described (Natarajan et al, 2016). For all proteins, purified fractions were biotinylated using a BirA protein ligase kit (Avidity LLC). The biotinylated proteins were further purified by gel filtration (S200, Cytiva) in PBS buffer pH 7.4. The proteins were finally tested for biotinylation efficiency by gel-shift analysis by incubating them in excess streptavidin. The proteins were incubated with Dynabeads M-280 Streptavidin (Invitrogen) (10 μg proteins for 15 μL beads resin) for 1 h at RT to prepare protein-coated Dynabeads. CD3γε and CD3δε tetramers were made by stepwise addition of 1-part streptavidin-PE to 10 parts biotinylated CD3 subunits in 10-min intervals in ice and subsequently concentrating it into 1X PBS, pH 7.4. For, CD3 tetramer staining experiments, purified TCR-CD3 complex or CD3 subunits or TCR-coated Dynabeads were first blocked with streptavidin and biotin solutions (Streptavidin/Biotin blocking kit, Vector Laboratories). The washed beads were then incubated with 50 or 350 ng/mL CD3γε or CD3δε tetramers and incubated for 2 h at RT. For IE$^k$ tetramer binding, Dynabeads coated TCR-CD3 complex was incubated with 100 ng/μL IE$^k$ tetramers (K5, K3, and MCC) (at 50 μL volume) for 2 h at RT prior to CD3 tetramer staining. The samples were analyzed for tetramer binding in Cytek Aurora 5 (Cytek Bio) and data was analyzed using FlowJo (ver 10.8.1).

## TCR and CD3 subunit structure generation, complex docking, and in silico mutagenesis

### Crosslink-guided model

3D models of individual murine CD3 domains (CD3γ, CD3δ) were built by homology modeling using ICM-Pro software (Molsoft LLC., La Jolla, CA) (Fernandez-Recio et al, 2003) applied to different structures as templates found in the PDB database as shown in Appendix Table S3. The coordinates of the TMs of all subunits relative to the TCR subunits were inherited from the human cryo-EM structure (PDB: 6JXR) (Dong et al, 2019). Full atom, 3D models of the two CD3 hetero-oligomers were docked to a grid potential representation of the 3D model of the 2B4 murine TCR. Diverse conformations without clashes and compact conformations exhibiting reasonable protein-protein interfaces were identified by calculated, estimated free energy of the complex, as previously described (Fernandez-Recio et al, 2002a, b; Fernandez-Recio et al, 2003; Garzon et al, 2009), which includes

terms for van der Waals, electrostatics, hydrogen bonding and solvation and the energy units approximate kcals in free energy calculations. All compact (all CD3 and TCR domains contacting at least one other domain) and conformations of the complexes without clashes were retained and their calculated free energy score was re-weighted by their contact surface area of the UAA side-chains with the CD3 domains and the distance between these UAA side-chains and the CD3. Docked conformations whose termini were inconsistent with the length of the linking segments that connect the termini of the TMs with the CD3 folded domains were discarded. Conformations impinging on the membrane or the 2C11 antibody were discarded. Unclashed, compact, cross-link compatible conformations of the complexes that did not impinge on the membrane or 2C11 were retained. In order to find mutations of interest in the model, both interfaces, CD3γδ and CD3εδ, were interrogated using the mutation analysis suite of algorithms in ICM. All 19 naturally occurring amino acid structures (not proline) were tried in all the positions that were in the interface of TCRαβ. Briefly, the binding free energies (composite of van der Waals, electrostatics, torsional strain and entropy) were calculated by the difference in binding energy in between the wild type and mutated amino acid residue (Neves et al, 2012). Monte Carlo simulations were run in order to relax any potential clashes caused by the mutation (Fernandez-Recio et al, 2003). The mutations that produced the lowest predicted $\Delta E_{binding}$ were chosen for further in vitro analysis.

## Statistics

T cell activation experiments of mutants in T cell hybridoma and Jurkat cells are done in triplicates for every peptide concentration. For comparisons of activation parameters of individual mutants with wild type in Jurkat cells, non-parametric unpaired tests were used.

## Data availability

This study includes no data deposited in external repositories.

The source data of this paper are collected in the following database record: biostudies:S-SCDT-10_1038-S44319-024-00314-3.

## Peer review information

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

## Acknowledgements

We thank Peter G. Schultz, Scripps Research Institute for the PU6-pBpa plasmid. We also thank David Kranz (University of Illinois) for providing us 58−/−T-cell hybridoma and Mark M. Davis (Stanford University) for providing us with Chinese hamster ovary (CHO) cells expressing I-E^k. We also thank Thomas P. Sakmar (Rockefeller University) for protocols and advice on photo-crosslinking technique. We thank Duane Moogk (McMaster University) and Yury Patskovsky (NYU Grossman School of Medicine) for helpful discussions and critical reading of the manuscript. We thank Eric Ni (Yale University) for helpful discussions. Mass spectrometry experiments for protein identification were supported in part by NYU Langone Health and Laura and Isaac Perlmutter Cancer Center support grant P30CA016087 from NCI. This work was supported by the NIH grants NIGMS R01 GM124489 (to MK), NCI R01 CA284604 (to MK), NCI R01 CA243486 (to MK), and NCI R21 CA263378 (to AN).

## Author contributions

**Aswin Natarajan**: Conceptualization; Data curation; Formal analysis; Funding acquisition; Visualization; Methodology; Writing—original draft. **Yogambigai Velmurugu**: Formal analysis; Investigation. **Manuel Becerra Flores**: Data curation; Software; Validation; Visualization. **Fatoumatta Dibba**: Data curation; Investigation. **Saikiran Beesam**: Data curation; Investigation. **Sally Kikvadze**: Investigation. **Xiaotian Wang**: Investigation. **Wenjuan Wang**: Conceptualization. **Tianqi Li**: Investigation. **Hyewon Shin**: Investigation. **Timothy Cardozo**: Conceptualization; Supervision; Writing—review and editing. **Michelle Krogsgaard**: Conceptualization; Supervision; Funding acquisition; Writing—review and editing.

Source data underlying figure panels in this paper may have individual authorship assigned. Where available, figure panel/source data authorship is listed in the following database record: biostudies:S-SCDT-10_1038-S44319-024-00314-3.

## Disclosure and competing interests statement

AN declares no competing interests. MK serves on the scientific advisory boards of Genentech and Merck and Co. and received research support from Merck Sharp & Dohme Corp., a subsidiary of Merck and Co., Inc., Genentech, Biogen, Novartis and the Mark Foundation.

# Expanded View Figures

## Western blots of purified TCR-CD3 complex

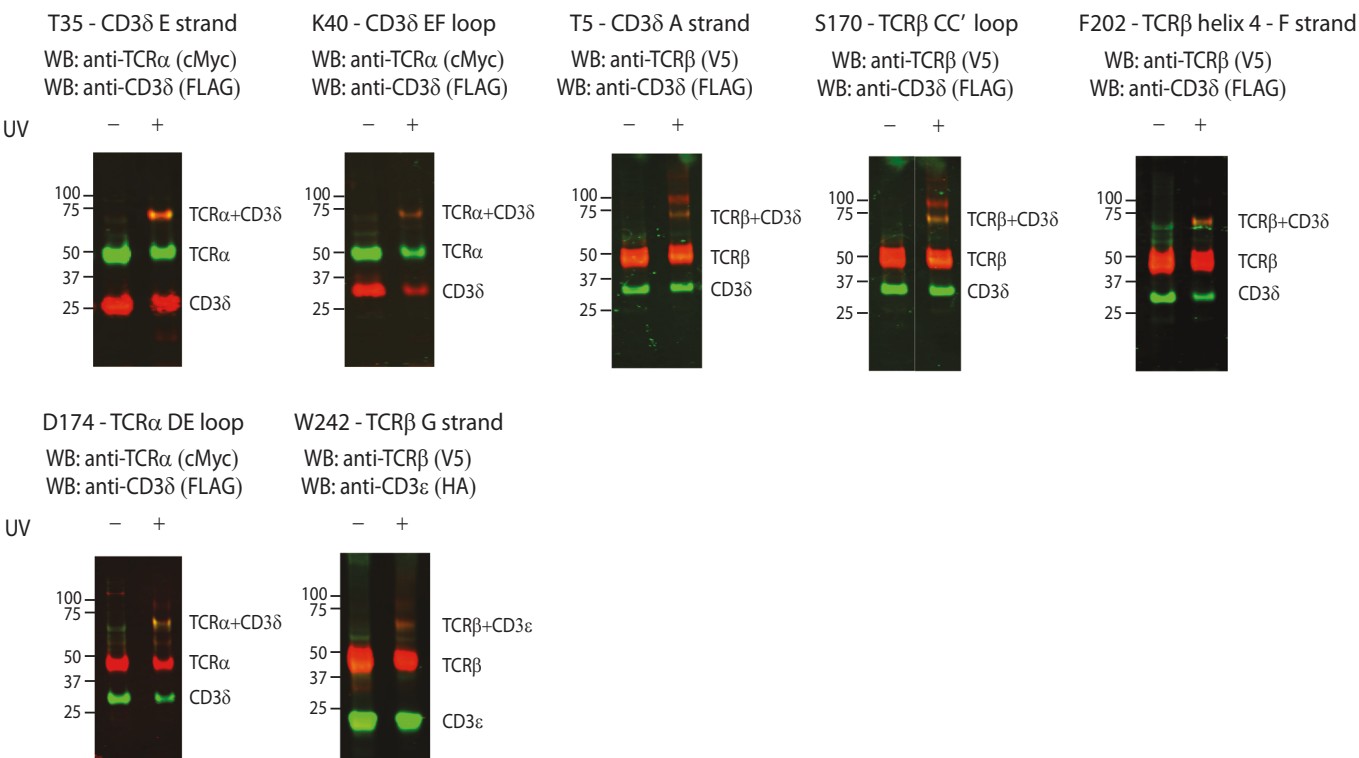

**Figure EV1. Crosslinking in purified TCR-CD3 complex.**

Western blots of purified mutant TCR-CD3 complexes—δT35, δK40, δT5, βS170, βF202, αD174, and βW242. TCRα + CD3δ crosslinked bands for δT35, δK40 and αD174 are present below 75 kDa. These blots were stained with anti-TCRα (cMyc) antibody and anti-CD3δ (FLAG). TCRβ + CD3δ crosslinked bands for δT5, βS170 and βF202 are present below 75 kDa. These blots were stained with anti-TCRβ (V5) antibody and anti-CD3δ (FLAG). TCRβ + CD3ε crosslinked band for βW242 is present below 75 kDa. These blots were stained with anti-TCRβ (V5) antibody and anti-CD3ε (HA). Anti-rabbit IRDye 680LT- and anti-mouse IRDye 800CW were used as secondary antibodies for detection for all blots.

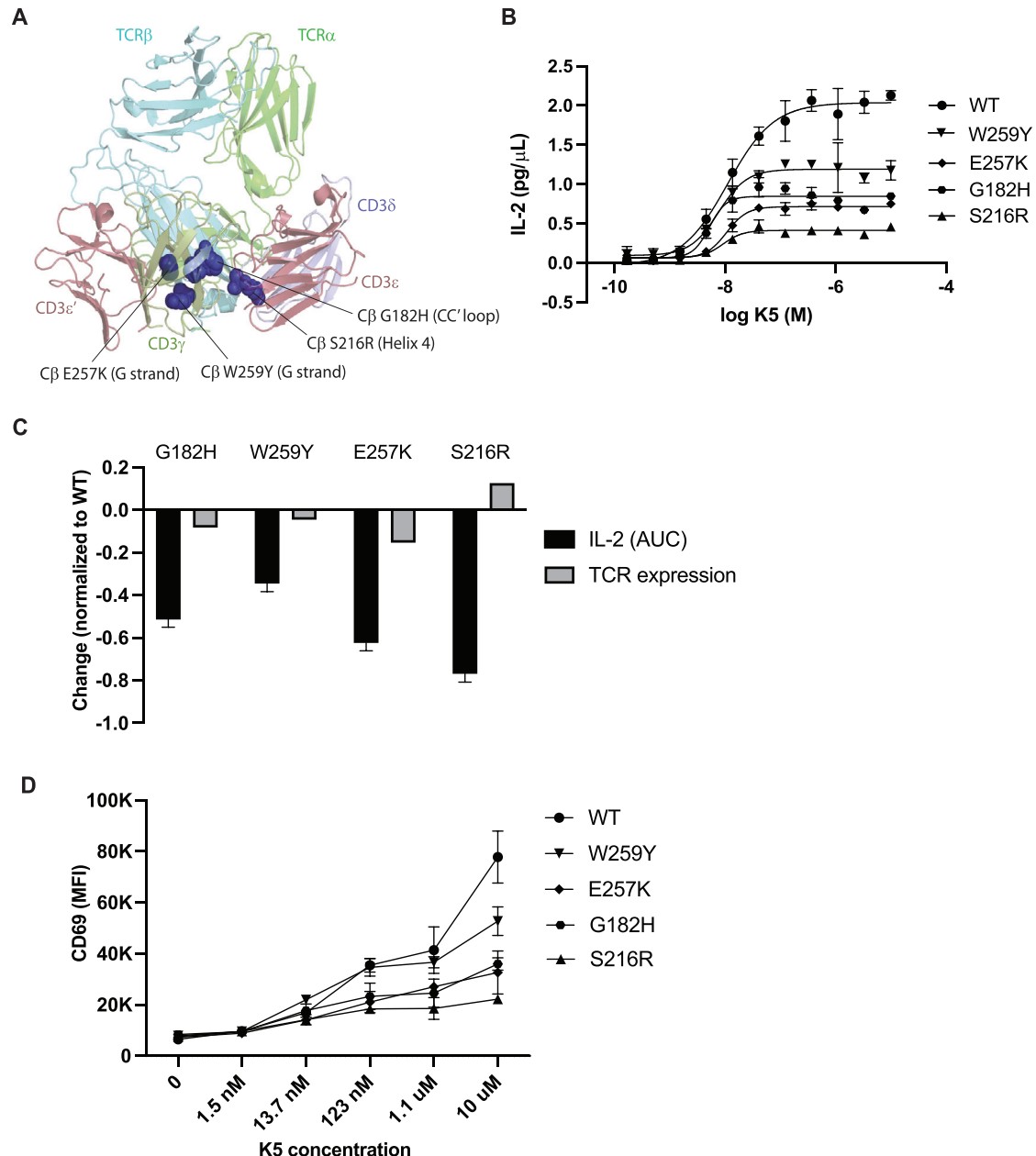

**Figure EV2. Stabilizing TCR-CD3 extracellular interactions reduces TCR signaling.**

(A) Locations of the mutated residues indicated on the human TCR-CD3 cryo-EM structure (PDB: 6JXR). $G_{182}H$ is located in the Cβ CC' loop, $E_{257}K$ and $W_{259}Y$ are located in the Cβ G strand and $S_{216}R$ is located in Cβ Helix 4. (B) ELISA assays (plot of IL-2 produced vs concentration of activating peptide) for mutant 2B4 T cell hybridoma clones activated with CHO/I-E$^k$/K5 in biological replicates ($n = 3$ for each peptide concentration). (C) Change in the area under the curve for IL-2 production (black) between the indicated mutant T cell hybridoma and wild type 2B4 hybridoma when activated with CHO cells expressing the cognate pMHC IE$^k$/K5 calculated based on (B). Change in TCR expression when compared to the wild type 2B4 TCR expression (MFI) is plotted in gray. (D) Plots of CD69 expression (MFI) on Jurkats expressing 2B4 mutant and wild type TCR when activated with indicated concentrations of K5 peptide (CHO/I-E$^k$/K5) in biological replicates ($n = 3$ for each peptide concentration) for 16 h. Data information: In (B, D), data are presented as mean ± SD. In (C), area under the curves is represented as mean ± SEM. For (D), Appendix Table S7 shows non-parametric unpaired t-test values for wild type-mutant comparisons.

