## [Peer Review File · EMBO Reports]

In situ cell-surface conformation of the TCR-CD3 signaling complex

Aswin Natarajan, Yogambigai Velmurugu, Manuel Becerra Flores, Fatoumatta Dibba, Saikiran Beesam, Sally Kikvadze, Xiaotian Wang, Wenjuan Wang, Tianqi Li, Hyewon Shin, Timothy Cardozo, and Michelle Krogsgaard

Corresponding author(s): Michelle Krogsgaard (Michelle.Krogsgaard@nyulangone.org)

Review Timeline:

Submission Date:	30th Dec 23
Editorial Decision:	29th Feb 24
Revision Received:	22nd Aug 24
Editorial Decision:	7th Oct 24
Revision Received:	22nd Oct 24
Accepted:	26th Oct 24

Editor: Achim Breiling

Transaction Report:

Dear Dr. Krogsgaard,

Thank you for the submission of your research manuscript to EMBO reports. I have now received the reports from the three referees that were asked to evaluate your study, which can be found at the end of this email.

As you will see, the referees think that the findings are of interest. However, they have several comments, concerns, and suggestions, indicating that a major revision of the manuscript is necessary to allow publication of the study in EMBO reports. From the analysis of the referee comments it is clear that a significant revision is required before publication can be considered, and I would also understand your decision if you chose to rather seek rapid publication elsewhere at this stage.

However, I would like to give you the opportunity to address the concerns and would be willing to consider a revised manuscript with the understanding that all referee concerns must be addressed in the revised manuscript and in a detailed point-by-point response. Acceptance of your manuscript will depend on a positive outcome of a second round of review. It is EMBO reports policy to allow a single round of revision only and acceptance of the manuscript will therefore depend on the completeness of your responses included in the next, final version of the manuscript.

- 1) a .docx formatted version of the final manuscript text (including legends for main figures, EV figures and tables), but without the figures included. Figure legends should be compiled at the end of the manuscript text.
- 2) individual production quality figure files as .eps, .tif, .jpg (one file per figure), of main figures and EV figures. Please upload these as separate, individual files upon re-submission.

- 4) a complete author checklist, which you can download from our author guidelines

(<https://www.embopress.org/page/journal/14693178/authorguide>). Please insert page numbers in the checklist to indicate where the requested information can be found in the manuscript. The completed author checklist will also be part of the RPF.

5) that primary datasets produced in this study (e.g. RNA-seq, CHIP-seq, structural and array data) are deposited in an appropriate public database. If no primary datasets have been deposited, please also state this in a dedicated section (e.g. 'No primary datasets have been generated and deposited'), see below.

The accession numbers and database should be listed in a formal "Data Availability" section (placed after Materials & Methods) that follows the model below. This is now mandatory (like the COI statement). Please note that the Data Availability Section is restricted to new primary data that are part of this study. This section is mandatory. As indicated above, if no primary datasets have been deposited, please state this in this section

Data availability

8) Regarding data quantification and statistics, please make sure that the number "n" for how many independent experiments were performed, their nature (biological versus technical replicates), the bars and error bars (e.g. SEM, SD) and the test used to calculate p-values is indicated in the respective figure legends (also for EV figures and all those in an Appendix). Please also check that all the p-values are explained in the legend, and that these fit to those shown in the figure. Please provide statistical testing where applicable. Please avoid the phrase 'independent experiment', but clearly state if these were biological or technical replicates. Please also indicate (e.g. with n.s.) if testing was performed, but the differences are not significant. In case n=2, please show the data as separate datapoints without error bars and statistics. See also: <http://www.embopress.org/page/journal/14693178/authorguide#statisticalanalysis>

9) Please add scale bars of similar style and thickness to microscopic images, using clearly visible black or white bars (depending on the background). Please place these in the lower right corner of the images themselves. Please do not write on or near the bars in the image but define the size in the respective figure legend.

10) Please also note our reference format:

12) We now use CRediT to specify the contributions of each author in the journal submission system. CRediT replaces the author contribution section. Please use the free text box to provide more detailed descriptions and do not provide your final manuscript text file with an author contributions section. See also our guide to authors: <https://www.embopress.org/page/journal/14693178/authorguide#authorshipguidelines>

13) We would encourage you to use 'Structured Methods', our new Materials and Methods format. According to this format, the Materials and Methods section should include a Reagents and Tools Table (listing key reagents, experimental models, software and relevant equipment and including their sources and relevant identifiers), uploaded as separate file, followed by a Methods and Protocols section in which we encourage the authors to describe their methods using a step-by-step protocol format with bullet points, to facilitate the adoption of the methodologies across labs. More information on how to adhere to this format as well as downloadable templates (.doc or .xls) for the Reagents and Tools Table can be found in our author guidelines (section 'Structured Methods'):

14) Please shorten the abstract to not more than 175 words and order the manuscript sections like this, using these names: Title page - Abstract - Keywords - Introduction - Results - Discussion - Materials and Methods - Data availability section - Acknowledgements - Disclosure and Competing Interests Statement - References - Figure legends - Expanded View Figure legends

Finally, please note that all corresponding authors are required to supply an ORCID ID for their name upon submission of a revised manuscript. Please find instructions on how to link the ORCID ID to the account in our manuscript tracking system in our Author guidelines: <http://www.embopress.org/page/journal/14693178/authorguide#authorshippinguidelines>

I look forward to seeing a revised version of your manuscript when it is ready. Please let me know if you have questions or comments regarding the revision.

Yours sincerely,

Referee #1:

Natarajan et al. have attempted to map close contacts between the subunits of the mouse TCR-CD3 antigen receptor (mTCR $\alpha\beta\gamma\delta\epsilon\zeta$ octamer), specifically those between TCR $\alpha\beta$ constant (C) domains and the ectodomain (ECD) of CD3 $\epsilon\delta$ and CD3 $\epsilon\gamma$ dimers. Using a well-established technical protocol, they introduced single amber mutations at specific sites of either TCR $\alpha\beta$ C domains or CD3 δ or CD3 γ ECD to biosynthetically incorporate an unnatural amino acid (UAA), pAzpa, instead of the natural counterpart. Upon UV irradiation, covalent bonding may ensue if pAzpa and a natural AA are sufficiently close and for sufficient time (ideally, ms). Identification of the target AA cross-linked to pAzpa may be obtained by accurate mass spectrometry. This technique may allow, though not without failing, an approximate mapping of intra- and inter-protein contacts (e.g., relatively stable ligand-receptor, enzyme-substrate, interactions). Usually employed to corroborate structural data, this technique has considerable intrinsic limitations. Positions in TCR $\alpha\beta$ and CD3 δ and CD3 γ were selected using bona fide contacts reported in recently published cryo-EM structures of detergent-extracted human TCR-CD3 (at 3.7-3.0 Å resolution, by three groups between 2019 and 2022) or NMR chemical shift perturbation upon solution binding of human CD3 ectodomains with human TCR $\alpha\beta$ ECD (work by Orban & Mariuzza and co-workers and by Natarajan et al.). Aided by the cross-linking data, molecular docking was implemented using well-known computational tools to corroborate proximity of TCR $\alpha\beta$ to CD3 $\epsilon\delta$ and CD3 $\epsilon\gamma$. Also, detergent extraction with or without a supply of a cholesterol analogue was used to probe TCR-CD3 stability by pAzpa crosslinking.

Two goals were hoped to be achieved. Confirm or refute the cryo-EM and/or NMR data and find evidence for changes in AA proximity at specific interaction site between TCR $\alpha\beta$ and CD3 upon ligand binding or in the absence of cholesterol in detergent-extracted TCR-CD3. The latter questions were aimed to corroborate recent evidence that pMHC binding induces allosteric changes that relax TCR-CD3 quaternary structure implicating modifications of TCR-CD3 TMDs bundle configuration (Lanz et al. 2021)

Some results of the present work agree with site-directed mutagenesis (Khuns and Davis) and with more trust-worthy cryo-EM (Dong and al.) and NMR (Orban & Mariuzza), all suggesting a "sided" arrangement of CD3 $\epsilon\delta$ and CD3 $\epsilon\gamma$ next to TCR $\alpha\beta$. A sizable number of cross-linking experiments could not be exploited as pAzpa substitution caused considerable reduction in TCR-CD3 cell surface expression or did not confirm close contacts sites assigned by cryo-EM. The role of cholesterol, suggested to keep a "resting state" of TCR-CD3 complex was confirmed by the pAzpa approach, suggesting (but not clearly discussed in the manuscript) allosteric connection between TCR-CD3 and TMDs, as previously suggested by Lanz et al. 2021 and by extensive MDS of TCD-CD3 by Prakaash et al. (Plos Comp. Biol. 2021). The cross-linking experiments appear to be technically well executed. The conclusions drawn are essentially confirmatory, with some small differences with respect to contacts assignment by NMR and cryo-EM. However, other potentially interesting inferences could be made possibly because, due to the use of

pAzpa, the experimental system has intrinsic limitation (see below) and/or because the technical protocol used to assess the structural integrity of pAzpa-containing TCR-CD3 is largely insufficient and flawed. For this reason some pAzpa substitution compounds mutational effects that cannot be disentangled from the intended readout (see below). It is therefore hard to establish with certainty if the findings shown here are solid enough for rejecting cryo-EM data and some crystal structure data (for TCR α AB loop) as well as evidence pro or contra ligand-dependent allosteric regulation suggested by NMR, genetic, biochemical and functional data and MDS.

Specific critics and suggestions:

1 - pAzpa, which features an aromatic side chain, introduces almost invariably non-conservative substitutions in TCR-CD3 (e.g., suppression of a net charge, or a polar or small side chain substituted with the bulky benzene ring). Such changes are highly susceptible of introducing local and long-distance structural perturbations in a large and dense spatially connected TCR-CD3. Close connections made by single AA side chains often via multiple bonds, van der waals and hydrophobic forces with other AA side chains or main chain atoms can be altered by pAzpa, causing stable rearrangements locally, distally and/or changes of AA side chains and main chain dynamics. These alterations, that are very difficult to anticipate, have perturbing effects before cross-linking, the mutation likely causing already in some instances a structural change, hence incurring the risk of testing mutational perturbation and not wt interactions. This limitation compromises the possibility of inferring with certainty close contacts and ligand-induced structural changes. Moreover, the relatively low resolution of current TCR-CD3 cryo-EM structures limits accurate modelling of all bonds, which is usually better afforded by high-resolution crystal structures, not obviously (except by NMR) local and long-distance dynamics. However, the manuscript keeps quiet on the limitation of pAzpa substitutions that are all but neutral, as well the limitations of cryo-EM and NMR studies. To indicate pAzpa-substituted TCR-CD3, the authors simply refer to "mutants" but do not discuss case-by-case the nature of the substitution and its potential effects, in an evident effort to minimize the limitations of the AUU-based biochemical approach.

Indeed, in a few instances pAzpa substitution results in drastically or considerable reduced surface expression of TCR-CD3, a well-known phenotype reporting on TCR-CD3 structural stability. Moreover, the FACS analysis (double staining with anti-CD3 and anti-C β Abs or anti-CD3 ϵ Abs and MHC-tetramer - the latter binding to V α V β), used to assess TCR-CD3 structural stability, is inadequate on at least two grounds. First, over-expression of TCR-CD3 in 293 cells, while apparently a handy short-cut, precludes accurate verification of potential effects of pAzpa substitutions on signaling. pAzpa mutations may increase or decrease signaling, presumably by changing the dynamics of TCR-CD3 and/or perturb allosteric trajectories, hence introducing biases for a correct interpretation of the results. Thus, the FACS analysis set up is a rather weak assay to answer to this crucial questions. Second, FACS analysis of wt TCR-CD3 overexpressed in 293 cells reveals two distinct and spread cell populations, while cell surface expression of physiological amounts of TCR-CD3 in normal T cells or in transformed T-cell line model systems (e.g., Jurkat cells) is a single gaussian-distributed population. It is known that overexpression of TCR and/or CD3 components eludes cellular quality controls (especially in transformed cell line) for ensuring a stoichiometric TCR-CD3 assembly for surface expression. Extreme over-expression (as the second peak of the FACS profile shows) might also lead to some clustering of a fraction of surface TCR-CD3. The two peaks observed here for TCR-CD3 wt in 293 cells could be the consequence of the high expression peak being incompletely assembled TCR-CD3 (e.g., lacking $\zeta\zeta$). Whether both peaks contain identical fully assembled TCR-CD3 is difficult to establish. Moreover, ligand-independent ζ -ITAM phosphorylation upon overexpressed TCR-CD3 in 293 cells (see, earlier work by James and Vale, Nature 2006) may lead to subtle, yet functionally relevant structural changes (critical in an allosterically regulated model) as well as to receptor downregulation. Some of these possibilities can be experimentally verified by accurate FACS staining. Also, I assume that UV-induced cross-linking will be effective on both surface and internalized TCR-CD3. If a sizable proportion of TCR-CD3 is internalized under over-expression conditions or large amounts of incompletely assembled complexes are stuck in intracellular compartments, these affects will compromise a rigorous comparison of wt and mutants as the molecular populations observed are not homogeneous, inevitably blurring data interpretation. Over-expression may also hide subtle assembly defects due pAzpa substitutions because quantitative compensation can result in "wt expression" that mask altered TCR-CD3 structure and/or generating populations of on/off switch receptors at steady state. Adding to the uncertainty, some pAzpa substitutions decrease the size of TCR-CD3 high expression peak and considerably modify the shape of the lower-expressor population. Why is so? Any deviation (revealed or concealed) of surface expression in "mutants" compared to wt constitutes structural alterations by pAzpa and possible functional alteration.

Accurate double staining FACS plots reporting direct comparisons of pAzpa substitution with wt must be shown. More properly, bar-coding staining distinguishing cells expressing wt and pAzpa substitutions must be used to achieve great accuracy in comparison, as a rigorous experimental test to assess the neutrality or not of pAzpa substitution. Ideally, a much safer and neat approach is to use near-physiological expression of wt and pAzpa TCR-CD3 "mutants" in Jurkat cell line lacking TCR $\alpha\beta$ and in Jurkat variants crispr ko for CD3 δ or CD3 γ (easy to generate). Determination of the EC50 of peptide dose for each pAzpa replacement showing identical surface expression and unaltered cell activation will guarantee that potential structural effects by pAzpa are safely evaluated before cross-linking. Because pAzpa technique does not require large amount of material it is unclear why the authors did not use a better controlled experimental set-up as the one suggested above.

2 - Another worrying aspect of this investigation is the fact that pAzpa substitutions are carried out in mouse TCR-CD3 while the structural reference are the Cryo-EM and NMR data of human TCR-CD3. Inspection of human and mouse TCR C α and C β and CD3 reveals a non-negligible number of interspecies substitutions (including non-conservative ones). The authors only briefly touch on this critical point, yet they do not openly discuss this limitation. In the absence of at least a mouse TCR-CD3 cryo-EM

structure, it is hard to establish the pertinence of the pAzpa substitutions.

3 - Another point not discussed here is why crosslinking in many instances does not lead to quantitative ligation of two contiguous subunits? Do the authors have an explanation? Moreover, another serious limitation of the work (this one commented on) is that highly dynamical changes activating allosteric trajectories upon ligand binding (a likely possibility discussed by other authors (see Mariuzza et al. JBC 2020 and Lanz et al. Cell Reports 2021) can hardly be sensed by the cross-linking approach. Moreover, cross-linking probing used here would be largely unreliable if major hypothetical conformation changes occur in regions of TCR-CD3 not covered by pAzpa substitutions.

4 - I found somewhat disturbing that in several instances the authors cite papers to give for granted notions what are purely speculative, as based either on artificial experimental systems (mechano-sensing) or only very indirectly deduced empirically or having no substantial experimental support. Also, they do not cite properly some literature or ignore it tout-court (see below).

All in all, I find that this investigation leads to a stalemate. It is impossible to make firm conclusions considering that the current cryo-EM structures, used as a guide for pAzpa substitutions, can be biased because obtained in detergent and has poor spatial resolution. Conversely, mapping obtained by pAzpa substitutions has serious intrinsic limitations, worsened by poor phenotypic analysis of TCR-CD3 intactness upon pAzpa substitutions. It may be worth it for the authors to have a close look at the Notti/Walz et al. paper uploaded on BioRxiv (to be published relatively soon) reporting an interesting new structure of TCR-CD3 obtained in a nanodisk composed of several lipids (including cholesterol and charged lipids).

The summary of this manuscript states: "This study contributes valuable insights into the structural dynamics and functional aspects of the TCR-CD3 complex, emphasizing the broader relevance of photo-crosslinking methodologies in unravelling immune receptor mechanisms". This a highly optimistic statement that does not much the approximative experimental design, the poor quality of the FACS analyses and the discussion. The authors should considerably tone down such an enthusiastic statement and engage instead in a rigorous discussion on the limitation of the cross-linking approach to highlight only what can be extracted from the work at this stage, that, in my opinion, provides very little original contribution to the understanding of TCR-CD3 structure-function.

Minor points:

4 - Citations and statements. Some important citations of the TCR-CD3 literature are missing, others are inexactly cited or at times not cited where they should. Moreover, some statements should be better formulated and toned down.

a)Page 3. Referring to the membrane-proximal tetracysteine motif present in the CD3 ϵ , CD3 γ and CD3 δ subunits, major contributions by Alarcon and co-workers and Kappler and co-workers are not cited. These papers convey the notion of a more subtle/signalling role of the highly conserved rigidified CD3 stems, discordant with the conclusions of the cited paper by Wucherpfennig and co-workers.

b)Page 4. The authors state: "However, a previous report suggests that engagement of multiple TCR-CD3 complexes by dimeric or tetrameric pMHC is required to induce CD3 ϵ cytosolic conformational changes (Minguet et al., 2007)." To make this notion more substantial, as based on more general molecular changes of TCR-CD3, the authors should cite also Lanz et al (2021). This work has provided substantial biochemical evidence that monomeric pMHC ligation allosterically modifies TCR-CD3 quaternary structure cohesion, a conclusion backed by mutational analysis.

c)Page 4. The authors state: "Photo-crosslinking of incorporated unnatural amino acids (UAA) is a powerful tool for investigating complex protein-protein interactions, molecular mechanisms, and spatiotemporal conformational states (Coin, 2018; Coin et al., 2013). Then: "Photo-crosslinking is well-suited for investigating weak protein-protein interactions" and later: "Photo-crosslinking, distinguished by its precision and specificity in capturing dynamic interactions...". I strongly recommend toning down the enthusiasm for such a useful technique as it carries a number of intrinsic shortcomings (as mentioned above) and it is largely inadequate to capture dynamic interactions/spatiotemporal conformational changes. The very same negative result in this investigation, despite a substantial number of evidence now accumulated on allosteric regulation of TCR-CD3 activation, does not lay well with professing an enthusiastic support for this technique. I also disagree that photo-crosslinking can effectively document weak protein-protein interactions.

d)Page 35. It is stated that Lanz et al. (2021) would have observed "...CD3- ζ dissociation from TCR upon activation. To my knowledge, no data in Lanz et al. showed that this event happens in the native TCR-CD3 embedded in the cell membrane. Rather, a reduced recovery of CD3- ζ was observed after detergent solubilization in pMHC liganded vs. unliganded TCR-CD3. I suggest to be more precise about this, in order not to convey a wrong messages about what has been actually shown in published data.

e)Page 11. In discussing the potential role of the C β FG loop as important for ligand discrimination the authors cite Das et al 2015, in which in vitro evidence using an artificial devise imposing considerable force ramp on TCR-CD3 suggest this role. Recent data by Schütz and Huppa suggest instead that the magnitude of forces experienced at realistic IS by TCR-CD3 upon physiological ligand engagement is one-order of magnitude lower than those generated by artificial devises. This suggest that

claims for mechano-sensing experience by TCR)CD3 should be downsized. Independently of this, what is portrayed on page 11 is a restrictive (or perhaps incorrect) view of the FG loop function if considering genetic data in mice by Reinherz and co-workers indicating that deletion of the entire FG loop does not impair thymocyte positive selection, though signalling seems to be attenuated or increased depending on T-cell developmental context, as shown by two studies by Reinherz and co-workers (cited by in tis manuscript). If the FG loop was critical for ligand discrimination, positive selection should be dramatically altered. In presenting such an interesting aspect of TCR-CD3 structure-function, I suggest being more precise, namely quote back-to-back Reinherz in vivo and vitro work and Mariuzza & Orban work to compare and contrast potential role(s) of the C β FG loop.

f)Page 34. The quote: "This suggests that T cell signaling operates at a highly dynamic macromolecular level, necessitating the interplay between the "on" and "off" states of CD3 on the TCR." This is not a novel notion. This possibility was largely discussed in Lanz et al 2021 who showed for first time that various single mutations (in the TMD of β or ζ) can spontaneously activate TCR-CD3 (i.e., an "off"-to-"on" transition) and augment ligand-induced signaling efficacy. THis was invariably correlated to loosening of TCR-CD3 quaternary structure. It would help to cite here Lanz et al. 2021 to adequately back up the authors statement.

g)Page 34. The sentence: "It also suggests that strengthening the complex interface could lead to a loss of flexibility or fluidity in overall T-cell synapse, which is crucial for effective T cell signaling (Floris 731 J. van Eerden, 2023; Rangarajan et al., 2018). This observation aligns with the 'serial triggering' model for T cell activation, where the high off-rates and optimal affinities of TCR-pMHC". Frankly, I don't understand the connection between TCR-CD3 "flexibility/fluidity",the IS and serial triggering (by now, a model largely recognised inconsistent and obsolete on several ground by a plethora of data - not to be confused with elegant and plausible ligand "binding/rebinding" proposed by others). This paragraph could be deleted as it is a too far reaching and not strictly necessary for discussing the data presented and their implications.

h)Page 35. "However, it is crucial to note that our observation does not rule out the possibility of dynamic or flexibility changes occurring in the TCR-CD3 subunit components, potentially influencing signaling (Natarajan et al., 2017; Rangarajan et al., 2018). This paragraph alludes to the fact that the cross-linking strategy used has failed to provide evidence for consistent and stable conformational changes upon ligand binding (i.e., a stable and large structural change as opposed to changes in dynamics, both of which leading to allosteric communication). Such an option to explain ligand induced allosteric activation for TCR-CD3 in the form of well-established allosteric changes based largely on "dynamic allostery" or "entropic allostery" has been discussed for the first time in Lanz et al (and see references therein). I may be worth quoting here this published contribution.

i)Page 38. It is perplexing to see that the authors cite Brazin et al. paper of 2018 as an example suggesting "conformational switch" TCR-CD3. Brazin et al. reported NMR data obtained with synthetic single TCR α TMD alone in lipid micelles, un enterily articial reductivist condition hardly matching the reality that consists of an $\alpha\beta$ TMD pair. Brazin et al. proposed the hypothetical existence in a natural setting (the entire TCR-CD3 in a realistic membrane) of two distinct TCR α TMD configurations. It is clear now that, after we all see how TCR-CD3 bundles of TMRs are arranged in space (see also, Notti et al BioRxiv paper and Prakaash et al. Plos Comput. Biol. 2021) such a highly speculative supposition should be dismantled and considered obsolete. Along the same lines, the idea of "mechanical switch coupled to T-cell receptor triggering to the cytoplasmic juxta-membrane regions of CD3 ζ " is also totally inconsistent with the cryo-EM data, with more recent cryo-EM data by Notti et al in BioRxiv and Prakaash et al. Plos Comput. Biol. 2021, both investigations revealing by physico-chemical data or computational simulations that (as correctly speculated more than ten years ago by Kai Wuckerpennig) the cytoplasmic juxta-membrane regions are floppy as they lack secondary structure. THis is evident from the primary sequence of the tails indicating formation of random coils that are invisible in the cryo-EM structures as extremely mobile as opposed the behaviour of folded sequences. Such a condition renders transmission of mechanical force from TMDs to the CD3 ζ intracellular tail practically impossible. Such condition is entirely different from the intrinsic rigidity of the oxidised tetra-cystestein motif of CD3 ϵ , CD3 γ and CD3 δ that can instead ensure mechanical trasmission of vectorial force between CD3 ECDs and the TMDs.

The above are warning to keep in mind that in order to build solid paradigms on complex molecular structure-function mechanisms we must discriminate between solid experimental fact consistent with fundamental physical and chemical laws and "wishful thinking". Commenting on notions/speculations having evidently lost consistency and credibility might be remembered in review articles as historical relics but should be avoided in original articles.

Referee #2:

The manuscript "In situ cell-surface conformation of the TCR-CD3 signaling complex" by Natarajan et al., deals with a very timely and controversial topic, namely how the quaternary structure of the most important receptor in T cells, the TCR, looks like and implications for the TCR's function. Recent cryo-EM structures of the TCR in detergent micelles or in a membrane bilayer resulted in different structures. Thus, it is still unclear how the TCR looks like in a living T cell, i.e. in its native environment. Natarajan et al. answer this question by expression of the TCR in HEK cells, incorporating the unnatural amino acid p-azido-phenylalanine (pAzpa) at specific sites in the TCR, followed by photo-crosslinking and analysing the cross-links by WB and MS. Using 48 different TCR mutants 15 specific crosslinks were identified that allowed to computational construct a model of the TCR's quaternary arrangement that is similar to the detergent-derived cryo-EM structure. Upon pMHC tetramer binding changes

were not detected, however evidence for a role of cholesterol in the stability of the TCR was obtained.

I find this a very elegant and important study and recommend publication, if the following points of criticism were addressed.

1. The WBs show that much less than 50% of the molecules of a certain subunit get crosslinked (in the case where a crosslink is seen). What are the reasons for not seeing nearly 100% of the molecules being cross-linked? Is there a large excess of non-assembled subunits in the ER that will not crosslink? Is the cross-linking inefficient, i.e. even in a complete TCR not all proximities will lead to a crosslink? Or, do two different conformational pools of the TCR exist on the cell surface - maybe in a very fast equilibrium? In one pool the X-link occurs, and in the other pool it does not? This should be discussed. If feasible, an experiment could be done in which TCRs are UV-crosslinked, then surface TCRs are bound to an anti-CD3 antibody, TCRs purified via this antibody and analysed by WB. This would allow to look only at the surface TCR pool.
2. For the different regions, such as TCRbeta FG loop or G strand, it would be good to mention how the amino acids for the mutagenesis were selected. Why were the ones that were mutated chosen, and not others.
3. As explained nicely in the manuscript, conformational flexibility might be too rapid to be captured by the cross-link method. To gain more insight, one could use the point mutants that lead to a stabilized TCR and then repeat the crosslink to test whether a higher percentage of the crosslinked molecules would crosslink.
4. Conformational flexibility might be too rapid to be captured by the cross-link method but relevant of pMHC tetramer binding or signaling. It would be good to UV-crosslink a TCR, in order to fix the TCR in a certain conformation, and then (i) quantify in a dose response pMHC tetramer binding and (ii) quantify downstream signaling after tetramer activation.
5. Legend to Figure 1: " S41 (blue)" should be changed to " S41 (red)".
6. To the reader that is not expert in MS it would be good to explain what the PSM (peptide-spectrum match) is and what the values mean.
7. In lines 553 and 554 the authors write: "consistent with previous evidence from pMHC-TCR-CD3 cryo-EM structures (Saotome et al., 2023; Susac et al., 2022), revealing no structural changes in the TCR-CD3 complex upon complexation with pMHC." To my opinion this is misleading. In their manuscript Natarajan et al. used pMHC tetramers that activate the TCR. In contrast, in the cited papers the authors used soluble monomeric pMHC that do not activate and do not induce structural changes (Minguet et al., 2007). Thus, in contrast to the study by Natarajan et al., the cited papers are not looking at TCRs that were bound to a stimulating ligand. This has to be made clear.
8. When discussing cholesterol binding it would be good to mention the original paper that discovered cholesterol binding to the TCR (PMID: 23091059).
9. Line 460 misses a full stop.
10. Line 469: "Stabilizing TCR-CD3 extracellular interactions does not improve T cell functionality" would it not be more informative to write "Stabilizing TCR-CD3 extracellular interactions reduces TCR signaling" ?

Referee #3:

The T-cell mediated immune response against invading pathogens is initiated from the recognition of antigenic peptide-MHC (pMHC) molecules by T-cell receptors (TCR) on T cell surface. As an octamer complex, TCR comprises an antigen-binding subunit (TCR $\alpha\beta$) and three CD3 signaling subunits (CD3 $\zeta\zeta$, CD3 $\delta\epsilon$, and CD3 $\gamma\epsilon$). Structural characterization of the specificity of TCR-pMHC interactions and the assembly of TCR-CD3 complex are important for understanding TCR triggering mechanism and function. Compared with previous studies that reported structural information of TCR piece by piece using NMR and X-ray crystallography, recently, the whole pictures of unliganded and even liganded TCR-CD3 cryo-EM structures at high resolutions are coming up. Here, the authors have combined the use of photo-crosslinking of incorporated unnatural amino acids (UAA) with computational docking, making it possible to investigate the in situ cell-surface conformation of the TCR-CD3 complex in a site specific manner. With the coming up of several structures reporting the whole picture of TCR, TCR triggering models, especially the conformation change model, will be open up for more discussion.

Major findings of this paper:

With combinatory use of photo-crosslinking of incorporated UAA with computational docking, together with CD3 tetramer-based assays and mutagenesis, the authors characterized the in situ cell-surface conformation of the TCR-CD3 complex. The authors first established and validated their photo-crosslinking-docking technique for in situ structural study, as they observed similar structural arrangement of their crosslinking model with the cryo-EM structures reported. They identified and confirmed the CD3 ϵ' -CD3 γ -CD3 ϵ -CD3 δ arrangement around the $\alpha\beta$ TCR in situ. Interestingly, they demonstrated the involvement of cholesterol

in maintaining the stability of TCR complex. Finally, the authors observed minimal subunit movements or reorientations upon activation by antibodies and pMHC tetramers, thus suggesting the absence of 'inactive-active' conformational states in the TCR constant regions and the extracellular CD3 subunits.

To the best of my knowledge, this paper is the first trial for the application of photo-crosslinking technique in the structural study of immune receptors. It provides an alternative strategy for structural and in situ analysis of protein complexes in their native states. Overall, I find the study is carefully designed and the conclusion is supported by their data.

On the other hand, besides the four cryo-EM structures of the TCR-CD3 complex in detergents cited by the authors (Dong et al., 2019; Chen et al., 2022; Susac et al., 2022; Saotome et al., 2023), there is also a cryo-EM structure of TCR-CD3 complex in proteolipid nanodiscs, a much more native-like bilayer environment, reported in 2023 (Notti et al., bioRxiv, 2023). Notti et al. reported that their structures reveal novel conformations that distinct from those in detergents, and represent the resting state of an unstimulated TCR on T cells combined with their cross-linking data. Therefore, it will be intriguing to ask which structure is more close to the native one, and how the conformation change model applies in TCR triggering. Here I have a few important questions that should be addressed for clarity.

Major points:

1. An important contribution by this paper is the characterization of the in situ TCR-CD3 conformation on the cell surface. Based on the structure comparison with the cryo-EM structures reported (Dong et al., 2019; Chen et al., 2022; Susac et al., 2022; Saotome et al., 2023), the authors suggested the absence of 'inactive-active' conformational states, though they also observed several specific differences, particularly in the TCR-CD3 interface residues. As described by the authors, the cryo-EM structures used for structure comparison are 'detergent-purified' complexes. Very recently, an unliganded human TCR in a native-like lipid bilayer has been reported by Notti et al. in bioRxiv. Notti et al. observed structure differences in the orientation arrangements of $\alpha\beta$ TCR-CD3 ectodomain and JM regions, suggesting a "closed and compacted" resting state of the human TCR. Although the structure PDB coordinates have not been open yet for public, it will be interesting to discuss based on the opened paper. Specifically, Notti et al. observed structural differences in the interface between CD3 δ and TCR α (e.g. TCR α G79-CD3 δ N38, TCR α G79-CD3 δ D77, TCR α S78-CD3 δ D77) and the region around the TCR α and TCR β interdomain hinges (e.g. TCR α S104-V182, TCR β T130-H172).

It seems that Natarajan et al. in this paper selected the residues for photo-crosslinking mainly based on the structure from Dong et al. (Dong et al., 2019). Therefore, I suggest to perform the crosslinking assay for the above regions (i.e. , the interface between CD3 δ and TCR α , and the region around the TCR α and TCR β interdomain hinges), and give more insights on the conformation change model.

2. The authors discussed several times about the inconsistency of the extracellular binding modes of CD3 $\gamma\epsilon/\delta\epsilon$ to the $\alpha\beta$ TCR: single-sided mode reported here and double-sided mode reported by their own group before (Natarajan et al., 2016). They attributed the inconsistency to the absence of membrane, connecting peptides (CPs) and transmembrane domains in the soluble protein domains used in the previous NMR study. In the previous NMR study, they used NMR chemical shift perturbation (CSP) method to characterize the TCR-CD3 binding mode. In fact, CSP has limitation in distinguishing the direct and indirect contact between two interacting proteins. The NMR method that uses cross-saturation phenomena is suitable to identify the direct contact residue more accurately. The authors are suggested to make more discussion on this point.

3. The involvements of cholesterol molecules have been reported in the published cryo-EM structures. It is interesting that the authors also demonstrated the importance of cholesterol in maintaining the stability of the complex. They observed poor crosslinking for the tested mutants locating at the extracellular domain, such as C β E221, CD3 γ S14 and C α A172. Is this observation regarded as an allosteric effect caused by the addition of cholesterol molecules? As cholesterol usually binds to the TM region of transmembrane proteins.

Dr. Achim Breiling,
Senior Scientific Editor,
EMBO Reports

New York, August 22, 2024

Dear Dr. Breiling,

We are pleased that our manuscript "*In situ* cell-surface conformation of the TCR-CD3 signaling complex" was reviewed with many positive comments and suggestions to improve the manuscript. Here, we respond point-by-point to the comments/revisions from the reviewers, including highlighting new experiments we have done to support the data. Overall, we feel that our results fill important gaps that remain in understanding $\alpha\beta$ TCR-CD3 complex signaling. We performed the following new experiments as suggested by the reviewers – 1) quantifying structural perturbations introduced by pAzpa substitutions; 2) photo-crosslinking experiments on mutant-stabilized TCR-CD3 complexes; 3) tetramer-binding experiments in the absence and presence of UV-irradiation to activate crosslinking; 4) photo-crosslinking experiments in new mutants to study 'conformational change' model proposed by Notti et al.; and 5) additional photo-crosslinking experiments on mutants to study effect of cholesterol on complex stability. We hope that the additional experiments add sufficient value and clarity about TCR signaling through the extracellular TCR and CD3 subunits during T cell activation to warrant acceptance of our manuscript in EMBO Reports.

Reviewer 1:

Major comment 1:

pAzpa, which features an aromatic side chain, introduces almost invariably non-conservative substitutions in TCR-CD3 (e.g., suppression of a net charge, or a polar or small side chain substituted with the bulky benzene ring). Such changes are highly susceptible of introducing local and long-distance structural perturbations in a large and dense spatially connected TCR-CD3. Close connections made by single AA side chains often via multiple bonds, van der Waals and hydrophobic forces with other AA side chains or main chain atoms can be altered by pAzpa, causing stable rearrangements locally, distally and/or changes of AA side chains and main chain dynamics. These alterations, that are very difficult to anticipate, have perturbing effects before cross-linking, the mutation likely causing already in some instances a structural change, hence incurring the risk of testing mutational perturbation and not wt interactions. This limitation compromises the possibility of inferring with certainty close contacts and ligand-induced structural changes. Moreover, the relatively low resolution of current TCR-CD3 cryo-EM structures limits accurate modelling of all bonds, which is usually better afforded by high-resolution crystal structures, not obviously (except by NMR) local and long-distance dynamics. However, the manuscript keeps quiet on the limitation of pAzpa substitutions that are all but neutral, as well the limitations of cryo-EM and NMR studies. To indicate pAzpa-substituted TCR-CD3, the authors simply refer to "mutants" but do not discuss case-by-case the nature of the substitution and its potential effects, in an evident effort to minimize the limitations of the AUU-based biochemical approach.

Indeed, in a few instances pAzpa substitution results in drastically or considerably reduced surface expression of TCR-CD3, a well-known phenotype reporting on TCR-CD3 structural stability. Moreover, the FACS analysis (double staining with anti-CD3 and anti-C β Abs or anti-CD3 ϵ Abs and MHC-tetramer - the latter binding to V α V β), used to assess TCR-CD3 structural stability, is inadequate on at least two grounds. First, over-expression of TCR-CD3 in 293 cells, while apparently a handy short-cut, precludes accurate verification of potential effects of pAzpa substitutions on signaling. pAzpa mutations may increase or decrease signaling, presumably by changing the dynamics of TCR-CD3 and/or perturb allosteric trajectories, hence introducing biases for a correct interpretation of the results. Thus, the FACS analysis set up is a rather weak assay to answer to these crucial questions. Second, FACS analysis of wt TCR-CD3 overexpressed in 293 cells reveals two distinct and spread cell populations, while cell surface expression of physiological amounts of TCR-CD3 in normal T cells or in transformed T-cell line model systems (e.g., Jurkat cells) is a single gaussian-distributed population. It is known that overexpression of TCR and/or CD3 components eludes cellular quality controls (especially in transformed cell line) for ensuring a stoichiometric TCR-CD3 assembly for surface expression. Extreme over-expression (as the second peak of the FACS profile shows) might also lead to some clustering of a fraction of surface TCR-CD3. The two peaks observed here for TCR-CD3 wt in 293 cells could be the consequence of the high expression peak being incompletely assembled TCR-CD3 (e.g., lacking $\zeta\zeta$). Whether

both peaks contain identical fully assembled TCR-CD3 is difficult to establish. Moreover, ligand-independent ζ -ITAM phosphorylation upon overexpressed TCR-CD3 in 293 cells (see, earlier work by James and Vale, Nature 2006) may lead to subtle, yet functionally relevant structural changes (critical in an allosterically regulated model) as well as to receptor downregulation. Some of these possibilities can be experimentally verified by accurate FACS staining. Also, I assume that UV-induced cross-linking will be effective on both surface and internalized TCR-CD3. If a sizable proportion of TCR-CD3 is internalized under over-expression conditions or large amounts of incompletely assembled complexes are stuck in intracellular compartments, these affects will compromise a rigorous comparison of wt and mutants as the molecular populations observed are not homogeneous, inevitably blurring data interpretation. Over-expression may also hide subtle assembly defects due pAzpa substitutions because quantitative compensation can result in "wt expression" that mask altered TCR-CD3 structure and/or generating populations of on/off switch receptors at steady state. Adding to the uncertainty, some pAzpa substitutions decrease the size of TCR-CD3 high expression peak and considerably modify the shape of the lower-expressor population. Why is so? Any deviation (revealed or concealed) of surface expression in "mutants" compared to wt constitutes structural alterations by pAzpa and possible functional alteration.

Accurate double staining FACS plots reporting direct comparisons of pAzpa substitution with wt must be shown. More properly, bar-coding staining distinguishing cells expressing wt and pAzpa substitutions must be used to achieve great accuracy in comparison, as a rigorous experimental test to assess the neutrality or not of pAzpa substitution. Ideally, a much safer and neat approach is to use near-physiological expression of wt and pAzpa TCR-CD3 "mutants" in Jurkat cell line lacking TCR $\alpha\beta$ and in Jurkat variants crispr ko for CD3 δ or CD3 γ (easy to generate). Determination of the EC50 of peptide dose for each pAzpa replacement showing identical surface expression and unaltered cell activation will guarantee that potential structural effects by pAzpa are safely evaluated before cross-linking. Because pAzpa technique does not require large amount of material it is unclear why the authors did not use a better controlled experimental set-up as the one suggested above.

Response:

Structural perturbations introduced by pAzpa substitutions:

pAzpa, p-azido-L-phenylalanine is an effective photo-crosslinker that can be genetically encoded, the other being pBpa, a benzophenone derivative. Previously, our lab systematically compared the two crosslinkers and found that pAzpa has higher levels of incorporation and crosslinking efficiency in the TCR-CD3 complex (Wang et al., 2014). pAzpa is one of the smallest crosslinkers in terms of sidechain length in comparison to other crosslinkers such as pBpa and diazirine derivatives (Coin, 2018). Further, crystallographic structures of a protein complex with and without photo-crosslinker, pBpa, exhibited no structural variations between native structure and crosslinked structure (Sato et al., 2011). Based on these reports, we anticipate that the level of perturbation due to pAzpa substitution to be minimal. Nevertheless, to respond to the Reviewer's concern about this issue and to provide a quantitative measure for the level of perturbation introduced upon pAzpa substitution, we biophysically modeled the pAzpa-mutated (both crosslinked and non-crosslinked amino acids) *in silico* within the TCR, CD3 $\gamma\epsilon$ and CD3 $\delta\epsilon$ 3D structures. This allowed us to quantify and visualize the resulting perturbation of the local structure and electrostatic surface in the vicinity of the pAzpa mutant side chain, including calculation of the changes in van der Waals, electrostatic and solvation energy. This analysis does not reveal a correlation between pAzpa substitutions predicted to perturb the structure by resulting in higher energy and its ability to crosslink. Appendix Table S4 shows that mutations that resulted in worse energy are in the minority, and are distributed randomly between crosslinks that occurred (blue shaded) and those that didn't. Therefore, local perturbations caused by pAzpa mutations constitute a standard error in the study and did not bias the study towards a particular 3D model. This analysis suggests that bias in the study due to perturbation of the protein by pAzpa mutations is unlikely.

Viability of the pAzpa-incorporated TCR-CD3 complex:

Regarding the viability of the complex and its signaling ability after pAzpa substitution, we have checked the viability through H57 (TCR β antibody), 2C11 (CD3 ϵ antibody- recognizes contiguous surface epitope) and pMHC tetramer staining. We believe this as an accurate measure of viability of the complex assembled. Dong et al., 2019 used glutaraldehyde crosslinkers to stabilize TCR-CD3 complex purified from HEK cells and used anti-CD3 antibody to prove viability of the complex using surface plasmon resonance. In some instances, the poor expression of the crosslinker incorporated complex could also be the result of poor incorporation of the crosslinker and abrupt stoppage of protein synthesis during translation (due to the presence of amber stop

codon) and not always due to structural destabilization caused by pAzpa incorporation. The reviewer's assertion that introducing crosslinkers destabilizes the native conformation at every instance brings into question all previous structural studies (Grunbeck et al., 2011; Kramer et al., 2014; Krishnamurthy et al., 2011; Rannversson et al., 2016; Valentin-Hansen et al., 2014) on multiple protein complexes involving photo-crosslinkers.

Presence of two populations of TCR-CD3 complex in the transfected cell line:

We would like to bring to attention here that we do not see two distinct populations in the expression profile but rather a distribution with two maxima in many instances. Our laboratory has expertise in TCR transduction across different cell lines including 58/- T cell hybridoma and Jurkat lines and we have consistently observed a wide range of TCR/CD3 expression levels post-transduction with multiple local maxima. Even retroviral transduction of TCRs into primary mouse T cell results in wide range of TCR/CD3 expressions (Zhong et al., 2013). We had already considered the reviewer's observation that one of the populations in the expression flow plots could contain incompletely assembled complexes and UV-irradiation could also crosslink internalized unviable complexes. However, our protocol uses 2C11 (CD3 ϵ antibody)-IP to pull down complexes following UV-irradiation and prior to detergent solubilization so that only the surface-assembled complexes are immunoprecipitated. Our Western blot analysis of the crosslinked complexes indicates the presence of all individual components of the complex (TCR α , TCR β , CD3 γ , CD3 δ and CD3 ϵ). The presence of CD3 ζ in the complex was identified through mass spectrometry analysis (Dataset EV1-EV3).

Usage of other T cell lines instead of HEK cells for photo-crosslinking experiments:

Regarding the reviewer's suggestion to use a T cell system to perform crosslinking analysis, we feel this as out-of-scope for the research intended. Considerable time and effort are required for optimizing transfections of TCR, CD3 and orthogonal aminoacyl-tRNA synthetase/tRNA pairs into the new cell system. Further, the presence of endogenous proteins, for example: CD3 subunits in Jurkat β -deficient T cell line, could lead to mis-pairings. However, work by James and Vale (James and Vale, 2012) show that HEK cells are an ideal system for reconstituting T cell proteins and for studying T cell triggering mechanisms. In a recent study HEK cell system was used to identify the CD3 components that are required for optimal TCR surface expression (Degirmencay et al., 2024). In another work, HEK293FT system was used to study CD3 ζ phosphorylation following activation (Zheng et al., 2024). These studies indicate that HEK293 cells as valuable *in vitro* systems to study T cell signaling processes.

Major comment 2:

Another worrying aspect of this investigation is the fact that pAzpa substitutions are carried out in mouse TCR-CD3 while the structural reference are the Cryo-EM and NMR data of human TCR-CD3. Inspection of human and mouse TCR C α and C β and CD3 reveals a non-negligible number of interspecies substitutions (including non-conservative ones). The authors only briefly touch on this critical point, yet they do not openly discuss this limitation. In the absence of at least a mouse TCR-CD3 cryo-EM structure, it is hard to establish the pertinence of the pAzpa substitutions.

Response:

Experiments carried out in mouse system while pAzpa substitution locations were based from human structures:

In our experiments, we use pAzpa substitutions in mouse TCR and CD3 subunits and use the crosslinking constraints obtained to model the mouse TCR-CD3 extracellular complex using computational docking. We used experimental data from previous studies that includes cryo-EM, NMR, mutagenesis, fluorescence and structural analysis (Appendix Table S1, S2) not to identify individual amino acids but rather structural regions (loops, strands and helices) where we can introduce multiple pAzpa substitutions for crosslinking to neighboring subunits. Our study includes pAzpa substitutions at multiple sites that are conserved between mouse and human TCR-CD3 subunits (examples: C β CC' loop, C β helix 4 – F strand, C β helix 3 and C β G strand).

Major comment 3:

Another point not discussed here is why crosslinking in many instances does not lead to quantitative ligation of two contiguous subunits? Do the authors have an explanation? Moreover, another serious limitation of the work (this one commented on) is that highly dynamical changes activating allosteric trajectories upon ligand

binding (a likely possibility discussed by other authors (see Mariuzza et al. JBC 2020 and Lanz et al. Cell Reports 2021) can hardly be sensed by the cross-linking approach. Moreover, cross-linking probing used here would be largely unreliable if major hypothetical conformation changes occur in regions of TCR-CD3 not covered by pAzpa substitutions.

Response:

Lack of ligation of two contiguous subunits:

We respectfully disagree with the reviewer as we have crosslinked contiguous subunits during multiple instances in our crosslinking assays. For example, our positive control α S41 was identified based on its proximity to the TCR β subunit based on the 2B4 TCR crystal structure (PDB:3QJF) and we were able to successfully crosslink pAzpa substituted at α S41 to the TCR β subunit (Fig 1D, 1G). Similarly, pAzpa substitutions in C β helix 3 regions resulted in crosslinks between C β helix 3 and TCR α subunit (Fig 2G). Examination of the 2B4 TCR structure suggests the close proximity between TCR α subunit and C β helix 3.

Crosslinking does not capture dynamical changes and conformational changes occurring at sites not covered by pAzpa substitutions:

Although photo-crosslinking is the ideal technique to capture weak transient states, we acknowledge the possibilities of highly dynamical motions occurring within individual components in the extracellular TCR-CD3 complex upon antigen binding which may not be identified by photo-crosslinking. Our goal was to identify the existence (if any) of large conformational changes with respect to different components in the extracellular regions of the TCR-CD3 complex upon antigen binding which we believe we successfully accomplished through our results. The crosslinks that were examined for changes during antigen binding are spread throughout the TCR-CD3 complex (Fig 6A). We revised the Results section to highlight that major conformational changes were not seen in the regions tested – page 20 in the revised manuscript.

Major comment 4:

I found somewhat disturbing that in several instances the authors cite papers to give for granted notions what are purely speculative, as based either on artificial experimental systems (mechano-sensing) or only very indirectly deduced empirically or having no substantial experimental support. Also, they do not cite properly some literature or ignore it tout-court (see below).

Response:

We have added appropriate references and revised statements as suggested by the reviewer. See in minor comments.

Final comment:

All in all, I find that this investigation leads to a stalemate. It is impossible to make firm conclusions considering that the current cryo-EM structures, used as a guide for pAzpa substitutions, can be biased because obtained in detergent and has poor spatial resolution. Conversely, mapping obtained by pAzpa substitutions has serious intrinsic limitations, worsened by poor phenotypic analysis of TCR-CD3 intactness upon pAzpa substitutions. It may be worth it for the authors to have a close look at the Notti/Walz et al. paper uploaded on BioRxiv (to be published relatively soon) reporting an interesting new structure of TCR-CD3 obtained in a nanodisk composed of several lipids (including cholesterol and charged lipids).

The summary of this manuscript states: "This study contributes valuable insights into the structural dynamics and functional aspects of the TCR-CD3 complex, emphasizing the broader relevance of photo-crosslinking methodologies in unravelling immune receptor mechanisms". This a highly optimistic statement that does not much the approximative experimental design, the poor quality of the FACS analyses and the discussion. The authors should considerably tone down such an enthusiastic statement and engage instead in a rigorous discussion on the limitation of the cross-linking approach to highlight only what can be extracted from the work at this stage, that, in my opinion, provides very little original contribution to the understanding of TCR-CD3 structure-function.

Response:

We would like to point out that the previous TCR-CD3 structural studies involving mutagenesis, fluorescence, NMR spectroscopy and cryo-EM were all used as guides to select mutations for pAzpa substitutions (Appendix

Table S1). For example, C β helix 3 and C α AB loop regions were tested based on evidence from NMR analysis (He et al., 2015; Natarajan et al., 2016) and fluorescence-based experiments (Beddoe et al., 2009). The viability of the pAzpa-substituted TCR-CD3 complexes have been established as indicated in our response to major comment 1. The article suggested by the reviewer – Notti et al., BioRxiv is still not peer-reviewed at the time of our revision and we could not access the PDB files. Despite the limitation of available data to us, we still introduced pAzpa substitutions in the TCR V α and CD3 δ regions based on the preprint article to identify crosslinks and change in crosslinks upon pMHC tetramer binding. Nonetheless, based on our crosslinking experiments, we could not get clearcut evidence for changes in crosslinks between TCR α and CD3 δ upon pMHC binding, suggesting an absence of conformational change involving these regions (Appendix Fig S16).

Overall, we respectfully disagree with the Reviewer's statement that our work provides very little contribution to the understanding of the TCR-CD3 structure function. We believe our work establishes the following: 1) *in situ* cell-surface extracellular TCR-CD3 structure matches the detergent-solubilized cryo-EM structures, 2) dynamic interplay between TCR and CD3 subunits that is crucial for TCR signaling, and 3) lack of substantial stable conformational changes in the TCR constant domains and CD3 extracellular regions upon antigen binding as previously speculated which we would argue is of outmost significance in terms of providing a detailed and physiological relevant understanding of TCR-CD3 complex signaling.

Minor comments:

- a) Page 3. Referring to the membrane-proximal tetracysteine motif present in the CD3 ϵ , CD3 γ and CD3 δ subunits, major contributions by Alarcon and co-workers and Kappler and co-workers are not cited. These papers convey the notion of a more subtle/signalling role of the highly conserved rigidified CD3 stems, discordant with the conclusions of the cited paper by Wucherpfennig and co-workers.

Response: The citations indicated by the reviewer regarding membrane-proximal tetracysteine motif are added on page 3 of the revised manuscript.

- b) Page 4. The authors state: "However, a previous report suggests that engagement of multiple TCR-CD3 complexes by dimeric or tetrameric pMHC is required to induce CD3 ϵ cytosolic conformational changes (Minguet et al., 2007)." To make this notion more substantial, as based on more general molecular changes of TCR-CD3, the authors should cite also Lanz et al (2021). This work has provided substantial biochemical evidence that monomeric pMHC ligation allosterically modifies TCR-CD3 quaternary structure cohesion, a conclusion backed by mutational analysis.

Response: The work by Lanz et al., is added to the introduction on page 4 of the revised manuscript.

- c) Page 4. The authors state: "Photo-crosslinking of incorporated unnatural amino acids (UAA) is a powerful tool for investigating complex protein-protein interactions, molecular mechanisms, and spatiotemporal conformational states (Coin, 2018; Coin et al., 2013). Then: "Photo-crosslinking is well-suited for investigating weak protein-protein interactions" and later: "Photo-crosslinking, distinguished by its precision and specificity in capturing dynamic interactions...". I strongly recommend toning down the enthusiasm for such a useful technique as it carries a number of intrinsic shortcomings (as mentioned above) and it is largely inadequate to capture dynamic interactions/spatiotemporal conformational changes. The very same negative result in this investigation, despite a substantial number of evidence now accumulated on allosteric regulation of TCR-CD3 activation, does not lay well with professing an enthusiastic support for this technique. I also disagree that photo-crosslinking can effectively document weak protein-protein interactions.

Response: We respectively disagree with the reviewer's contention that photo-crosslinking technique doesn't capture transient or weak protein-protein interactions. Cell-based photo-crosslinking techniques were used to identify transient complex formations in multiple protein systems involving transcription activators VP16 and Gal4 with Swi/Snf chromatin remodeling complex (Krishnamurthy et al., 2011), during nuclear transport involving nucleoporins (Yu et al., 2012), chaperone-assisted protein folding

(Zhang et al., 2011) among others. These examples are added to the main text – page 4 of the revised manuscript.

- d) Page 35. It is stated that Lanz et al. (2021) would have observed "...CD3- ζ dissociation from TCR upon activation. To my knowledge, no data in Lanz et al. showed that this event happens in the native TCR-CD3 embedded in the cell membrane. Rather, a reduced recovery of CD3- ζ was observed after detergent solubilization in pMHC liganded vs. unliganded TCR-CD3. I suggest to be more precise about this, in order not to convey a wrong message about what has been actually shown in published data.

Response: We apologize for the oversight regarding Lanz et al. work and the text is corrected now (Page 27 in the revised manuscript).

- e) Page 11. In discussing the potential role of the C β FG loop as important for ligand discrimination the authors cite Das et al 2015, in which in vitro evidence using an artificial device imposing considerable force ramp on TCR-CD3 suggest this role. Recent data by Schütz and Huppa suggest instead that the magnitude of forces experienced at realistic IS by TCR-CD3 upon physiological ligand engagement is one-order of magnitude lower than those generated by artificial devices. This suggest that claims for mechano-sensing experience by TCR-CD3 should be downsized. Independently of this, what is portrayed on page 11 is a restrictive (or perhaps incorrect) view of the FG loop function if considering genetic data in mice by Reinherz and co-workers indicating that deletion of the entire FG loop does not impair thymocyte positive selection, though signalling seems to be attenuated or increased depending on T-cell developmental context, as shown by two studies by Reinherz and co-workers (cited by in tis manuscript). If the FG loop was critical for ligand discrimination, positive selection should be dramatically altered. In presenting such an interesting aspect of TCR-CD3 structure-function, I suggest being more precise, namely quote back-to-back Reinherz in vivo and vitro work and Mariuzza & Orban work to compare and contrast potential role(s) of the C β FG loop.

Response: We understand the concern raised by the reviewer regarding the usage of artificial systems (single molecule tether assays) to study TCR signaling (Das et al., 2015). However, we feel it is important to cite all studies that examined the potential role of FG loop in TCR signaling irrespective of the technique used. The text is edited to reflect the importance of C β FG loop in TCR signaling and T cell development context (Page 10 in the revised manuscript).

- f) Page 34. The quote: "This suggests that T cell signaling operates at a highly dynamic macromolecular level, necessitating the interplay between the "on" and "off" states of CD3 on the TCR." This is not a novel notion. This possibility was largely discussed in Lanz et al 2021 who showed for first time that various single mutations (in the TMD of β or ζ) can spontaneously activate TCR-CD3 (i.e., an "off"-to-"on" transition) and augment ligand-induced signaling efficacy. THis was invariably correlated to loosening of TCR-CD3 quaternary structure. It would help to cite here Lanz et al. 2021 to adequately back up the authors statement.

Response: Lanz et al., is cited accordingly in Page 27 in the revised manuscript.

- g) Page 34. The sentence: "It also suggests that strengthening the complex interface could lead to a loss of flexibility or fluidity in overall T-cell synapse, which is crucial for effective T cell signaling (Floris 731 J. van Eerden, 2023; Rangarajan et al., 2018). This observation aligns with the 'serial triggering' model for T cell activation, where the high off-rates and optimal affinities of TCR-pMHC". Frankly, I don't understand the connection between TCR-CD3 "flexibility/fluidity", the IS and serial triggering (by now, a model largely recognised inconsistent and obsolete on several ground by a plethora of data - not to be confused with elegant and plausible ligand "binding/rebinding" proposed by others). This paragraph could be deleted as it is a too far reaching and not strictly necessary for discussing the data presented and their implications.

Response: The statement regarding serial triggering model has been removed from the revised manuscript.

- h) Page 35. "However, it is crucial to note that our observation does not rule out the possibility of dynamic or flexibility changes occurring in the TCR-CD3 subunit components, potentially influencing signaling (Natarajan et al., 2017; Rangarajan et al., 2018). This paragraph alludes to the fact that the cross-linking strategy used has failed to provide evidence for consistent and stable conformational changes upon ligand binding (i.e., a stable and large structural change as opposed to changes in dynamics, both of which leading to allosteric communication). Such an option to explain ligand induced allosteric activation for TCR-CD3 in the form of well-established allosteric changes based largely on "dynamic allostery" or "entropic allostery" has been discussed for the first time in Lanz et al (and see references therein). I may be worth quoting here this published contribution.

Response: Lanz et al., is cited accordingly in Page 28 in the revised manuscript.

- i) Page 38. It is perplexing to see that the authors cite Brazin et al. paper of 2018 as an example suggesting "conformational switch" TCR-CD3. Brazin et al. reported NMR data obtained with synthetic single TCR α TMD alone in lipid micelles, an entirely artificial reductivist condition hardly matching the reality that consists of an $\alpha\beta$ TMD pair. Brazin et al. proposed the hypothetical existence in a natural setting (the entire TCR-CD3 in a realistic membrane) of two distinct TCR α TMD configurations. It is clear now that, after we all see how TCR-CD3 bundles of TMRs are arranged in space (see also, Notti et al BioRxiv paper and Prakaash et al. Plos Comput. Biol. 2021) such a highly speculative supposition should be dismantled and considered obsolete. Along the same lines, the idea of "mechanical switch coupled to T-cell receptor triggering to the cytoplasmic juxta-membrane regions of CD3 $\zeta\zeta$ " is also totally inconsistent with the cryo-EM data, with more recent cryo-EM data by Notti et al in BioRxiv and Prakaash et al. Plos Comput. Biol. 2021, both investigations revealing by physico-chemical data or computational simulations that (as correctly speculated more than ten years ago by Kai Wuckerpennig) the cytoplasmic juxta-membrane regions are floppy as they lack secondary structure. This is evident from the primary sequence of the tails indicating formation of random coils that are invisible in the cryo-EM structures as extremely mobile as opposed the behaviour of folded sequences. Such a condition renders transmission of mechanical force from TMDs to the CD3 ζ intracellular tail practically impossible. Such condition is entirely different from the intrinsic rigidity of the oxidised tetra-cysteine motif of CD3 ϵ , CD3 γ and CD3 δ that can instead ensure mechanical transmission of vectorial force between CD3 ECDs and the TMDs.

Response: Branzin et al., documenting conformational switch in the TCR α subunit is removed from the manuscript. However, we would argue that citing Lee et al., 2015 is relevant here as this study used *in situ* proximity assays to determine the existence of different conformations in the CD3 $\zeta\zeta$ juxtamembranal regions before and after pMHC binding. The cryo-EM structures could not resolve these regions so we could not overrule the existence of these structures in physiological conditions. NMR structures of cytosolic CD3 ϵ and CD3 ζ revealed the presence of helical turns in them which provides existence of such states as a possibility (Xu et al., 2008; Zimmermann et al., 2017). The works cited by the reviewer – Notti et al., BioRxiv is not peer-reviewed and the PDB coordinates from the cryo-EM structures are still embargoed; Prakaash et al., is a full computational study the evidences from which cannot be used to undermine experimental evidence completely from Lee et al., in our humble opinion.

Final comment:

The above are warning to keep in mind that in order to build solid paradigms on complex molecular structure-function mechanisms we must discriminate between solid experimental fact consistent with fundamental physical and chemical laws and "wishful thinking". Commenting on notions/speculations having evidently lost consistency and credibility might be remembered in review articles as historical relics but should be avoided in original articles.

Response:

Given our limited knowledge in many techniques such as single-molecule study, molecular sensors that have been used to study T cell signaling mechanisms we are not at the liberty to judge all prior works in this subject. We try to evaluate all prior works to the best of our knowledge and cite them accordingly in our manuscripts.

Reviewer 2:

Comment 1:

The WBs show that much less than 50% of the molecules of a certain subunit get crosslinked (in the case where a crosslink is seen). What are the reasons for not seeing nearly 100% of the molecules being cross-linked? Is there a large excess of non-assembled subunits in the ER that will not crosslink? Is the cross-linking inefficient, i.e. seen in a complete TCR not all proximities will lead to a crosslink? Or, do two different conformational pools of the TCR exist on the cell surface - maybe in a very fast equilibrium? In one pool the X-link occurs, and in the other pool it does not? This should be discussed. If feasible, an experiment could be done in which TCRs are UV-crosslinked, then surface TCRs are bound to an anti-CD3 antibody, TCRs purified via this antibody and analysed by WB. This would allow to look only at the surface TCR pool.

Response:

We agree with the reviewer's observation that the crosslinking is not 100% for any mutant tested. This, in general, is the limitation of the photo-crosslinking technique as the yield of photo-crosslinked product is usually low. This technique is combined with mass spectrometric analysis to obtain qualitative results about protein-protein interactions (Coin, 2018). One reason for the lower crosslinks yield could be that the codon intended to add non-canonical amino acid is skipped in the ribosome leading to production of proteins without photo-crosslinker. In accordance with the suggestion by Reviewer 2, our protocol did involve UV-irradiation followed by anti-CD3 antibody binding and subsequent detergent solubilization and purification to select for only the surface TCR-CD3 complex pool (see Materials and Method section).

Comment 2:

For the different regions, such as TCRbeta FG loop or G strand, it would be good to mention how the amino acids for the mutagenesis were selected. Why were the ones that were mutated chosen, and not others.

Response:

For large regions such as C β FG loop, C β G strand and C α AB loop we chose alternating residues in the regions for introducing crosslinkers as it reduced the number of mutants that should be tested. A line has been added in page 6 to address this.

Comment 3:

As explained nicely in the manuscript, conformational flexibility might be too rapid to be captured by the cross-link method. To gain more insight, one could use the point mutants that lead to a stabilized TCR and then repeat the crosslink to test whether a higher percentage of the crosslinked molecules would crosslink.

Response:

We thank the reviewer for this suggestion and acknowledge that conformational flexibility could be one of the reasons for poor crosslinking efficiency. We performed photo-crosslinking experiments on mutant-stabilized TCR-CD3 complexes (Fig 5A) to identify whether crosslinking efficiencies improve. The triple mutant – β N236R/S238K/W242TAG, double mutants – β E227F/W242TAG and β E227F/W225TAG (TAG indicates the location for pApa substitution) had poor surface expression of the TCR-CD3 in comparison to the single mutants that crosslinked – β W242TAG and β W225TAG (Appendix Figure S13A). This was expected because T cell hybridoma and Jurkat expressing the single mutants – β E227F, β N236R, β S238K had to be sorted multiple times to select for T cell expressing stable TCR-CD3 complexes. Nonetheless, we tried crosslinking on β E227F/W225TAG mutant and crosslink between TCR β and CD3 γ subunit was identified similar to the one observed for β W225TAG even though β E227/W225TAG complex surface expression was much lower (Appendix Figure S13A, B). However, we weren't able to ascertain that mutant-stabilized TCR-CD3 complexes lead to improved crosslinking.

Comment 4:

Conformational flexibility might be too rapid to be captured by the cross-link method but relevant of pMHC tetramer binding or signaling. It would be good to UV-crosslink a TCR, in order to fix the TCR is a certain conformation, and then (i) quantify in a dose response pMHC tetramer binding and (ii) quantify downstream signaling after tetramer activation.

Response:

Again, we thank the reviewer for this insightful suggestion. Our mutant TCR-CD3 complexes were tested for viability after crosslinking using pMHC tetramer and we identified that the tetramer binding ability of the complex correlated to the surface expression. As suggested, we performed a dose-response tetramer binding assay before and after photo-crosslinking for couple of mutant TCR-CD3 complexes to evaluate whether conformational flexibility in the TCR-CD3 complex influences pMHC tetramer binding. Our results indicate that there is very little difference in pMHC tetramer binding between UV-crosslinked and non-crosslinked TCR-CD3 complexes (Appendix Figure S14A, B) indicating that fixing the TCR-CD3 complex via photo-crosslinks did not influence antigen binding. However, quantifying downstream signaling after tetramer activation is not possible in our current cross-linking setup as it is not a T cell system but rather using HEK293 cells.

Comment 5:

Legend to Figure 1: " α S41 (blue)" should be changed to " α S41 (red)"

Response: We have changed the legend in Figure 1 in the revised manuscript.

Comment 6:

To the reader that is not expert in MS it would be good to explain what the PSM (peptide-spectrum match) is and what the values mean.

Response: We have added an explanation in the result section. Page – 11 in the revised manuscript.

Comment 7:

In lines 553 and 554 the authors write: "consistent with previous evidence from pMHC-TCR-CD3 cryo-EM structures (Saotome et al., 2023; Susac et al., 2022), revealing no structural changes in the TCR-CD3 complex upon complexation with pMHC.". To my opinion this misleading. In their manuscript Natarajan et al. used pMHC tetramers that activate the TCR. In contrast, in the cited papers the authors used soluble monomeric pMHC that do not activate and do not induce structural changes (Minguet et al., 2007). Thus, in contrast to the study by Natarajan et al., the cited papers are not looking at TCRs that were bound to a stimulating ligand. This has to be made clear.

Response: We agree with the reviewer and the text is modified. We also visit this concept in the introduction section – page 4 in the revised manuscript.

Comment 8:

When discussing cholesterol binding it would be good to mention the original paper that discovered cholesterol binding to the TCR (PMID: 23091059).

Response: The reference is added – Page – 27 in the revised manuscript.

Comment 9:

Line 460 misses a full stop.

Response: The punctuation is added to the legend.

Comment 10:

Line 469: "Stabilizing TCR-CD3 extracellular interactions does not improve T cell functionality" would it not be more informative to write "Stabilizing TCR-CD3 extracellular interactions reduces TCR signaling"?

Response: We have modified the header as suggested.

Reviewer 3:**Major point 1:**

1. An important contribution by this paper is the characterization of the in situ TCR-CD3 conformation on the cell surface. Based on the structure comparison with the cryo-EM structures reported (Dong et al., 2019; Chen et al., 2022; Susac et al., 2022; Saotome et al., 2023), the authors suggested the absence of 'inactive-active' conformational states, though they also observed several specific differences, particularly in the TCR-CD3 interface residues. As described by the authors, the cryo-EM structures used for structure comparison are 'detergent-purified' complexes. Very recently, an unliganded human TCR in a native-like lipid bilayer has been reported by Notti et al. in bioRxiv. Notti et al. observed structure differences in the orientation arrangements of $\alpha\beta$ TCR-CD3 ectodomain and JM regions, suggesting a "closed and compacted" resting state of the human TCR. Although the structure PDB coordinates have not been open yet for public, it will be interesting to discuss based on the opened paper. Specifically, Notti et al. observed structural differences in the interface between CD3 δ and TCR α (e.g. TCR α G79-CD3 δ N38, TCR α G79-CD3 δ D77, TCR α S78-CD3 δ D77) and the region around the TCR α and TCR β interdomain hinges (e.g. TCR α S104-V182, TCR β T130-H172). It seems that Natarajan et al. in this paper selected the residues for photo-crosslinking mainly based on the structure from Dong et al. (Dong et al., 2019). Therefore, I suggest to perform the crosslinking assay for the above regions (i.e., the interface between CD3 δ and TCR α , and the region around the TCR α and TCR β interdomain hinges), and give more insights on the conformation change model.

Response:

We thank the reviewer for this important suggestion based on the Notti et al. pre-print on bioRxiv, which was also mentioned by Reviewer 1. We weren't aware of this pre-print at the time of our original manuscript submission, and the 3D coordinates of this structure remain embargoed at this time. However, we performed crosslinking analysis at the interface between TCR α -CD3 δ in the absence and presence of pMHC tetramer binding which should provide more insights into conformational change model. Based on the information from the Notti et al. manuscript, we chose CD3 δ residues N16 and E55 (corresponding human residues based on sequence alignment being N38 and D77), CD3 δ K59 and TCR α R39 (both based on the crosslink structure) to test for crosslinking (Appendix Figure S16A). We also used a control residue α S41 (that crosslinks to TCR β) which is near the hinge region for our crosslinking analysis (Appendix Fig S16A). Out of the mutants, CD3 δ N16 and CD3 δ K59 had very poor surface expression of the TCR-CD3 complex (Appendix Figure S16B). α R39 mutant did not reveal any noticeable crosslinks. Both control α S41 mutant and CD3 δ E55 mutant crosslinked with TCR β and TCR α respectively, although the CD3 δ -TCR α crosslink for δ E55 was much weaker in comparison to the TCR α -TCR β crosslink for α S41 (Appendix Fig S16C). However, in both instances, there was no change in the observed crosslinks upon binding with IE^k/MCC tetramer (Appendix Fig S16C). Therefore, based on evidences from our study, we could not infer the occurrence of conformational change involving the TCR α -CD3 δ region. Regarding the TCR interdomain hinges (α S104-V182, β T130-H172) we cannot test it with our current experimental set-up as these linkages would be intradomain crosslinks that cannot be identified by the Western blot analysis.

Major point 2:

2. The authors discussed several times about the inconsistency of the extracellular binding modes of CD3 $\gamma\epsilon/\delta\epsilon$ to the $\alpha\beta$ TCR: single-sided mode reported here and double-sided mode reported by their own group before (Natarajan et al., 2016). They attributed the inconsistency to the absence of membrane, connecting peptides (CPs) and transmembrane domains in the soluble protein domains used in the previous NMR study. In the previous NMR study, they used NMR chemical shift perturbation (CSP) method to characterize the TCR-CD3 binding mode. In fact, CSP has limitation in distinguishing the direct and indirect contact between two interacting proteins. The NMR method that uses cross-saturation phenomena is suitable to identify the direct contact residue more accurately. The authors are suggested to make more discussion on this point.

Response:

We agree with the reviewer's opinion that cross-saturation NMR is the best suitable technique to identify interface residues in large protein complexes. However, cross-saturation experiments work best if the interactions between the individual components are strong, which is not the case for extracellular TCR and CD3 subunits as identified by He et al. (He et al., 2015) and by us (Natarajan et al., 2016). Natarajan et al. previously (Natarajan et al., 2017) performed cross-saturation experiments to identify interfaces between TCR α - and β - subunits, where the interactions are much stronger than extracellular TCR-CD3 interactions. We have included a discussion of cross-saturation experiments in the revised manuscript – Page 26 in the revised manuscript.

Major point 3:

The involvements of cholesterol molecules have been reported in the published cryo-EM structures. It is interesting that the authors also demonstrated the importance of cholesterol in maintaining the stability of the complex. They observed poor crosslinking for the tested mutants locating at the extracellular domain, such as C β E221, CD3 γ S14 and C α A172. Is this observation regarded as an allosteric effect caused by the addition of cholesterol molecules? As cholesterol usually binds to the TM region of transmembrane proteins.

Response:

We thank the reviewer for this observation. For the revision, we retested the crosslinks in a controlled setting in the presence and absence of cholesterol and digitonin detergent which is used in the final sample preparation before UV irradiation (see Materials and Methods – TCR-CD3 complex purification from HEK cells) to clearly establish the role of cholesterol in the complex. Interestingly, crosslinks were observed for the mutants tested in the absence of cholesterol and digitonin although it was much weaker for mutants – δ K40, α A172 and β E221 (Appendix Fig S10). This seemingly suggests that cholesterol plays more of an allosteric role in maintaining TCR-CD3 extracellular conformation rather than being critical for the TCR-CD3 complex stability. The main text is revised appropriately according to our analysis.

Literature Cited

- Beddoe, T., Chen, Z., Clements, C.S., Ely, L.K., Bushell, S.R., Vivian, J.P., Kjer-Nielsen, L., Pang, S.S., Dunstone, M.A., Liu, Y.C., *et al.* (2009). Antigen ligation triggers a conformational change within the constant domain of the alphabeta T cell receptor. *Immunity* *30*, 777-788.
- Coin, I. (2018). Application of non-canonical crosslinking amino acids to study protein-protein interactions in live cells. *Current opinion in chemical biology* *46*, 156-163.
- Das, D.K., Feng, Y., Mallis, R.J., Li, X., Keskin, D.B., Hussey, R.E., Brady, S.K., Wang, J.H., Wagner, G., Reinherz, E.L., *et al.* (2015). Force-dependent transition in the T-cell receptor beta-subunit allosterically regulates peptide discrimination and pMHC bond lifetime. *Proc Natl Acad Sci U S A* *112*, 1517-1522.
- Degirmencay, A., Thomas, S., Holler, A., Burgess, S., Morris, E.C., and Stauss, H.J. (2024). Exploitation of CD3zeta to enhance TCR expression levels and antigen-specific T cell function. *Front Immunol* *15*, 1386132.
- Grunbeck, A., Huber, T., Sachdev, P., and Sakmar, T.P. (2011). Mapping the ligand-binding site on a G protein-coupled receptor (GPCR) using genetically encoded photocrosslinkers. *Biochemistry* *50*, 3411-3413.
- He, Y., Rangarajan, S., Kerzic, M., Luo, M., Chen, Y., Wang, Q., Yin, Y., Workman, C.J., Vignali, K.M., Vignali, D.A., *et al.* (2015). Identification of the Docking Site for CD3 on the T Cell Receptor beta Chain by Solution NMR. *The Journal of biological chemistry* *290*, 19796-19805.
- James, J.R., and Vale, R.D. (2012). Biophysical mechanism of T-cell receptor triggering in a reconstituted system. *Nature* *487*, 64-69.
- Kramer, K., Sachsenberg, T., Beckmann, B.M., Qamar, S., Boon, K.L., Hentze, M.W., Kohlbacher, O., and Urlaub, H. (2014). Photo-cross-linking and high-resolution mass spectrometry for assignment of RNA-binding sites in RNA-binding proteins. *Nature methods* *11*, 1064-1070.
- Krishnamurthy, M., Dugan, A., Nwokoye, A., Fung, Y.H., Lancia, J.K., Majmudar, C.Y., and Mapp, A.K. (2011). Caught in the act: covalent cross-linking captures activator-coactivator interactions in vivo. *ACS chemical biology* *6*, 1321-1326.
- Natarajan, A., Nadarajah, V., Felsovalyi, K., Wang, W., Jeyachandran, V.R., Wasson, R.A., Cardozo, T., Bracken, C., and Krosgaard, M. (2016). Structural Model of the Extracellular Assembly of the TCR-CD3 Complex. *Cell reports* *14*, 2833-2845.
- Natarajan, K., McShan, A.C., Jiang, J., Kumirov, V.K., Wang, R., Zhao, H., Schuck, P., Tilahun, M.E., Boyd, L.F., Ying, J., *et al.* (2017). An allosteric site in the T-cell receptor Cbeta domain plays a critical signalling role. *Nature communications* *8*, 15260.
- Rannversson, H., Andersen, J., Sorensen, L., Bang-Andersen, B., Park, M., Huber, T., Sakmar, T.P., and Stromgaard, K. (2016). Genetically encoded photocrosslinkers locate the high-affinity binding site of antidepressant drugs in the human serotonin transporter. *Nature communications* *7*, 11261.

Sato, S., Mimasu, S., Sato, A., Hino, N., Sakamoto, K., Umehara, T., and Yokoyama, S. (2011). Crystallographic study of a site-specifically cross-linked protein complex with a genetically incorporated photoreactive amino acid. *Biochemistry* *50*, 250-257.

Valentin-Hansen, L., Park, M., Huber, T., Grunbeck, A., Naganathan, S., Schwartz, T.W., and Sakmar, T.P. (2014). Mapping substance P binding sites on the neurokinin-1 receptor using genetic incorporation of a photoreactive amino acid. *The Journal of biological chemistry* *289*, 18045-18054.

Wang, W., Li, T., Felsovalyi, K., Chen, C., Cardozo, T., and Krogsgaard, M. (2014). Quantitative analysis of T cell receptor complex interaction sites using genetically encoded photo-cross-linkers. *ACS chemical biology* *9*, 2165-2172.

Xu, C., Gagnon, E., Call, M.E., Schnell, J.R., Schwieters, C.D., Carman, C.V., Chou, J.J., and Wucherpfennig, K.W. (2008). Regulation of T cell receptor activation by dynamic membrane binding of the CD3epsilon cytoplasmic tyrosine-based motif. *Cell* *135*, 702-713.

Yu, S.H., Boyce, M., Wands, A.M., Bond, M.R., Bertozzi, C.R., and Kohler, J.J. (2012). Metabolic labeling enables selective photocrosslinking of O-GlcNAc-modified proteins to their binding partners. *Proc Natl Acad Sci U S A* *109*, 4834-4839.

Zhang, M., Lin, S., Song, X., Liu, J., Fu, Y., Ge, X., Fu, X., Chang, Z., and Chen, P.R. (2011). A genetically incorporated crosslinker reveals chaperone cooperation in acid resistance. *Nat Chem Biol* *7*, 671-677.

Zheng, J., Zhang, Y., Cai, Y., Han, W., and Chen, W. (2024). An optimized non-T cell transfection system based on HEK293FT cells for CD3zeta phosphorylation and ubiquitination. *Journal of immunological methods* *528*, 113664.

Zhong, S., Malecek, K., Johnson, L.A., Yu, Z., Vega-Saenz de Miera, E., Darvishian, F., McGary, K., Huang, K., Boyer, J., Corse, E., *et al.* (2013). T-cell receptor affinity and avidity defines antitumor response and autoimmunity in T-cell immunotherapy. *Proc Natl Acad Sci U S A* *110*, 6973-6978.

Zimmermann, K., Eells, R., Heinrich, F., Rintoul, S., Josey, B., Shekhar, P., Losche, M., and Stern, L.J. (2017). The cytosolic domain of T-cell receptor zeta associates with membranes in a dynamic equilibrium and deeply penetrates the bilayer. *The Journal of biological chemistry* *292*, 17746-17759.

Dear Dr. Krogsgaard,

Thank you for the submission of your revised manuscript to our editorial offices. I have now received the report from the three referees that were asked to re-evaluate the study, you will find below. As you will see, the referees now support the publication of the study in EMBO reports. Referees #1 and 3# have some further comments you will find useful for the final revision of the manuscript text. Please also discuss the experimental limitations, future perspectives and expectations, as indicated by referee #3 (point 1).

Moreover, I have these editorial requests I ask you to address in a final revised manuscript:

- Please provide the abstract written in present tense throughout (and remove the word count information).
- Please add contact information (e-mail) for the corresponding author to the title page.
- We updated our journal's competing interests policy in January 2022 and request authors to consider both actual and perceived competing interests. Please review the policy <https://www.embopress.org/competing-interests> and update your competing interests if necessary. Please name this section 'Disclosure and Competing Interests Statement' and put it after the Acknowledgements section.
- We now use CRediT to specify the contributions of each author in the journal submission system. CRediT replaces the author contribution section. Please use the free text box to provide more detailed descriptions and do NOT provide your final manuscript text file with an author contributions section. See also our guide to authors: <https://www.embopress.org/page/journal/14693178/authorguide#authorshipguidelines>
- Please order the manuscript sections like this, using these names:
Title page - Abstract - Keywords - Introduction - Results - Discussion - Methods - Data availability section - Acknowledgements - Disclosure and Competing Interests Statement - References - Figure legends - Expanded View Figure legends
- Please move the paragraph about statistics into the methods section.
- Please use our reference format:
<http://www.embopress.org/page/journal/14693178/authorguide#referencesformat>

Please note that 'et al.' should be used after 10 author names and the year should be in brackets.

- Please make sure that all the funding information is also entered into the online submission system and that it is complete and similar to the one in the acknowledgement section of the manuscript text file. Presently, the grants 'NYU Langone Health and Laura and Isaac Perlmutter Cancer Center support grant P30CA016087 from NCI' is missing in the submission system. Please check.
- There are three datasets uploaded. Please remove their legends from the manuscript text file. Please add the legends for each instead as a separate TAB/sheet to the corresponding Excel files.
- Please make sure that the number "n" for how many independent experiments were performed, their nature (biological versus technical replicates), the bars and error bars (e.g. SEM, SD) and the test used to calculate p-values is indicated in the respective figure legends. Please also check that all the p-values are explained in the legend, and that these fit to those shown in the figure. Please provide statistical testing where applicable. Please avoid the phrase 'independent experiment', but clearly state if these were biological or technical replicates. Please also indicate (e.g. with n.s.) if testing was performed, but the differences are not significant. In case n=2, please show the data as separate datapoints without error bars and statistics. See also: <http://www.embopress.org/page/journal/14693178/authorguide#statisticalanalysis>

If n<5, please show single datapoints for diagrams. Could statistics be added to panels 5B-D and EV2B-D? Moreover:

- Please provide information related to n that is missing in the legends of figures 5c; EV 2b-d.
- Please define the error bars in the legends of figures 5b-d; EV 2b-d.
- As Appendix Fig. 15b has n=2, please remove the error bar and show the two datasets separate.
- Please add to each legend (main, EV and Appendix figures, where applicable) a 'Data Information' section explaining the statistics used or providing information regarding replicates and scales. See:

- Please add page numbers to the table of content of the Appendix file.

- Please upload the Reagents and Tools Table as a separate file and remove it from the Methods section. Please find downloadable templates (.doc) for the Reagents and Tools Table in our author guidelines (section 'Structured Methods'):

- Thank you for providing the source data (SD). Please upload the final SD as one ZIPed folder per figure. Within this folder, the files should be organized in subfolders, one subfolder for each panel. E.g. presently the SD for Figure 6 is not clearly labeled and it is unclear what belongs to panel B or to C. Please check.

In addition, I would need from you uploaded separately:

Best,

Referee #1:

The authors have provided answers to all my concerns and suggestions. I am sufficiently satisfied with the way they made changes in the manuscript with new controls and sensible arguments and new citations. These changes make the work experimentally stronger and more reliable. Overall, the answers show the authors' professionalism and unquestionable competence in their area of expertise. I recommend publication of the manuscript without revision.

However, in the interest of clarity, rigor and fairness, I have added a few comments for the authors to consider. They are intended to mitigate potential misunderstandings and to provide information that may be useful in future studies by these highly competent workers.

Response by the authors followed by my comments:

1 - Structural perturbations introduced by pAzpa substitutions:

"This analysis suggests that bias in the study due to perturbation of the protein by pAzpa mutations is unlikely".

However, local dynamics could be altered by pAzpa substitutions without necessarily result in major structural perturbations. Changes in local dynamics of side chains and/or main chain could diminish or increase TCR-CD3 functional response.

2 - "Viability of the pAzpa-incorporated TCR-CD3 complex".

Viability checks by Ab probing should be carried out by Ab titration, a most accurate quantitative test for probing potential minor, major or no effects by the pAzpa substitution

"The reviewer's assertion that introducing crosslinkers destabilizes the native conformation at every instance brings into question all previous structural studies".

Not necessarily. It depends on how potential destabilization is checked, that is, the technique used for this purpose and its stringency to inform about minor or major structural changes.

For instance, for a biological object whose function is known and measurable quantitatively in a cell, a simple, robust and accurate biological tests (e.g., a T-cell activation readout upon TCR-CD3 natural ligand titration) comparing in parallel WT vs. mutant (pAzpa substitutions, in this case) is by far the best and more powerful test - i.e., comparing in vitro the functional fitness of a T-cell (or an acceptable surrogate T-cell line).

At times, investigators applying their knowledge of fundamental physics and physical chemistry (not a single biologist would object the criticality of such "hard science" disciplines to explain Biology) forget simple and powerful notions in Biology, and do

not think that an optimized quantitative biological test is all what they need to carefully check by genetic perturbation (in this case unwanted) a functional read out.

3 - "Usage of other T cell lines instead of HEK cells for photo-crosslinking experiments".

For sure, HEK cells are a convenient system, but it has its conspicuous pitfalls. First and foremost, usage of at least surrogate T-cell lines instead of HEK cells.

James and Vale Nature paper reported that expression of full TCR-CD3 in HEK Cells results in considerable basal CD3 ITAM phosphorylation (likely due to the Fyn and Syk kinase expressed in relatively high amounts in HEK), if CD45 is not co-expressed. ITAM phosphorylation is likely to influence the global configuration of CD3 tails and, by ricochet it might alter the configuration of TMD helical bundles and of the CD3 subunits vis-a-vis TCRab. In this case, structural alterations may not be dramatic, yet enough to cause confounding effects when using cross-linking for structural studies. Using Jurkat lacking one CD3 subunit at a time is a superior experimental system and TCR-CD3 functional assays (see above) are, in the circumstance, the best bet for reporting on minor and major structural alteration.

The other disturbing drawback is an excessive expression of a recombinant protein, worse if it is a plasma protein. Unwanted, spurious non-physiological interactions favoured by mass action, may have significant effects on structure and function fidelity.

Response to the final comment:

"Nonetheless, based on our crosslinking experiments, we could not get clearcut evidence for changes in crosslinks between TCR and CD3 upon pMHC binding, suggesting an absence of conformational change involving these regions"

The authors should have also considered that such lack of "not..clearcut evidence" can be explained if Va is not stably bound to CD3d - i.e., their interaction is intermittent. VaVb appear to move together quite frequently up-and-down over CD3d and CD3g, respectively. This is suggested by an accurate MDS study by Weikl and co-workers (Elife 2021) and in Prakhaash et al. 2021, the latter also pointing out a couple of potential electrostatic interactions between V regions and CD3 ectodomain residues. The author might wish to cite these two papers, though it looks as they do not feel confident that (accurate and competent) MDS provides reliable predictive value and opportunities to test hypotheses. Importantly, such dynamic bending is visible only when TCR-CD3 is embedded in a lipid bilayer, better if made of a complex mixture of membrane lipids and values the "in-cell" approach of the work.

Referee #2:

This is a manuscript on a very hotly and controversially discussed topic, as e.g. witnessed by the report of referee #1. It provides an important puzzle piece to understand the functioning of the TCR.

My comments have been addressed well, the new BioRxiv of Notti has been discussed and new X-link experiments have been performed and integrated. I have also read to point-to-point reply addressing the other referees and to my opinion their comments have been well addressed as far as it was possible.

Thus, I recommend publication of this revised version.

Referee #3:

Several structural studies on the whole pictures of unliganded and even liganded TCR-CD3 cryo-EM structures at high resolutions have been reported recently. Among these structures, some are solved in detergents (Dong et al., 2019; Chen et al., 2022; Susac et al., 2022; Saotome et al., 2023) and some are done in nanodiscs (Notti et al., bioRxiv, 2023), a much more native-like bilayer environment. Furthermore, Notti et al. reported that their structure showed distinction from those in detergents, representing a resting state of an unstimulated TCR on T cells. Compared with these reported studies, an important contribution by the manuscript from Natarajan et al. is the characterization of the in situ TCR-CD3 conformation on the cell surface. Thus, it will be intriguing to ask which structure is more close to the native one, and how the conformation change model applies in TCR triggering.

Regarding point 1: Based on my suggestion and the information from the Notti et al. manuscript, Natarajan et al. performed further crosslinking analysis at several key sites, such as the interface between TCR α -CD3 δ , in the absence and presence of pMHC tetramer binding to get more insights regarding the conformational change model (Appendix Figure S16). They chose several residues for the crosslinking analysis. Unfortunately, some mutated residues were poorly expressed on the cell surface (e.g. δ N16 and δ K59), α R39 mutant did not reveal any noticeable crosslinks, and only δ E55 showed much weaker crosslink. Based upon these results, they addressed that they could not infer the occurrence of conformational change involving the TCR α -CD3 δ region because they observed no change in the crosslinks upon binding with IEk/MCC tetramer. It is good to see that Natarajan et al. additionally performed careful experiments to demonstrate this point, but at the same time, their conclusions

are not convincing enough based on the experimental data at the present stage. They may choose more residues nearby for further crosslinking analysis, or they should add more discussion in their manuscript about the experimental limitation and future perspective and expectation.

Regarding point 2: It is good to see that Natarajan et al. have included discussion and citation on the cross-saturation experiment and in the revised manuscript.

Regarding point 3: Natarajan et al. performed additional experiments to investigate the role of cholesterol molecules played in TCR-CD3 complex. It is good to see that they have come to a conclusion that cholesterol plays more of an allosteric role in maintaining TCR-CD3 extracellular conformation rather than just being critical for the TCR-CD3 complex stability, a new insight on the function of cholesterol molecules in TCR-CD3 complex.

We are pleased that our revised manuscript "*In situ* cell-surface conformation of the TCR-CD3 signaling complex" was reviewed positively by the reviewers and recommended for publication in EMBO Reports.

Here, we respond point-by-point to the comments for the revised manuscript from the reviewers:

Reviewer 1:

The authors have provided answers to all my concerns and suggestions. I am sufficiently satisfied with the way they made changes in the manuscript with new controls and sensible arguments and new citations. These changes make the work experimentally stronger and more reliable. Overall, the answers show the authors' professionalism and unquestionable competence in their area of expertise. I recommend publication of the manuscript without revision.

However, in the interest of clarity, rigor and fairness, I have added a few comments for the authors to consider. They are intended to mitigate potential misunderstandings and to provide information that may be useful in future studies by these highly competent workers.

Response:

We thank the reviewer for the detailed analysis of our manuscript and the valuable suggestions provided that made the manuscript undoubtedly better.

Comment 1 - Structural perturbations introduced by pAzpa substitutions:

"This analysis suggests that bias in the study due to perturbation of the protein by pAzpa mutations is unlikely".

However, local dynamics could be altered by pAzpa substitutions without necessarily result in major structural perturbations. Changes in local dynamics of side chains and/or main chain could diminish or increase TCR-CD3 functional response.

Response:

This is true and we witnessed T cell signal dampening while introducing stabilizing mutations into the TCR-CD3 interface (Fig 5B-D). We have discussed this in the Discussion section in the final version of the manuscript (Page 27).

Comment 2 - "Viability of the pAzpa-incorporated TCR-CD3 complex".

Viability checks by Ab probing should be carried out by Ab titration, a most accurate quantitative test for probing potential minor, major or no effects by the pAzpa substitution.

"The reviewer's assertion that introducing crosslinkers destabilizes the native conformation at every instance brings into question all previous structural studies".

Not necessarily. It depends on how potential destabilization is checked, that is, the technique used for this purpose and its stringency to inform about minor or major structural changes.

For instance, for a biological object whose function is known and measurable quantitatively in a cell, a simple, robust and accurate biological tests (e.g., a T-cell activation readout upon TCR-CD3 natural ligand titration) comparing in parallel WT vs. mutant (pAzpa substitutions, in this case) is by far the best and more powerful test - i.e., comparing *in vitro* the functional fitness of a T-cell (or an acceptable surrogate T-cell line).

At times, investigators applying their knowledge of fundamental physics and physical chemistry (not a single biologist would object the criticality of such "hard science" disciplines to explain Biology) forget simple and powerful notions in Biology, and do not think that an optimized quantitative biological test is all what they need to carefully check by genetic perturbation (in this case unwanted) a functional read out.

Response:

We thank the reviewer for the suggestions. We believe that tetramer staining of the assembled TCR-CD3 complex and correlating it to the surface expression of the complex through TCR β and CD3 ϵ antibody staining

is an equally good measure for viability of the mutant complex. Nonetheless in our future studies involving UAA substitutions we will add antibody titrations to the complex viability tests. Regarding functional readouts from the pAzpa-substituted complex, we could not test this in our current experimental setup. However, simple amino acid substitutions in the TCR-CD3 interface indeed led to altered T cell functionality (Fig 5B-D) which might also be the case with pAzpa substitutions. However, this does not make the complex unviable or unstable. We have discussed this in the Discussion section in the final version of the manuscript (Page 27).

Comment 3 - "Usage of other T cell lines instead of HEK cells for photo-crosslinking experiments".

For sure, HEK cells are a convenient system, but it has its conspicuous pitfalls. First and foremost, usage of at least surrogate T-cell lines instead of HEK cells.

James and Vale Nature paper reported that expression of full TCR-CD3 in HEK Cells results in considerable basal CD3 ITAM phosphorylation (likely due to the Fyn and Syk kinase expressed in relatively high amounts in HEK), if CD45 is not co-expressed. ITAM phosphorylation is likely to influence the global configuration of CD3 tails and, by ricochet it might alter the configuration of TMD helical bundles and of the CD3 subunits vis-a-vis TCR α . In this case, structural alterations may not be dramatic, yet enough to cause confounding effects when using cross-linking for structural studies. Using Jurkat lacking one CD3 subunit at a time is a superior experimental system and TCR-CD3 functional assays (see above) are, in the circumstance, the best bet for reporting on minor and major structural alteration.

The other disturbing drawback is an excessive expression of a recombinant protein, worse if it is a plasma protein. Unwanted, spurious non-physiological interactions favored by mass action, may have significant effects on structure and function fidelity.

Response:

We agree with many of the reviewer's concerns. The logical next step would be to perform crosslinking experiments in a Jurkat or T cell hybridoma system and we are working towards it. We have discussed this in the Discussion section in the final version of the manuscript (Page 29).

Final comment - "Nonetheless, based on our crosslinking experiments, we could not get clearcut evidence for changes in crosslinks between TCR α and CD3 δ upon pMHC binding, suggesting an absence of conformational change involving these regions"

The authors should have also considered that such lack of "not..clearcut evidence" can be explained if Va is not stably bound to CD3d - i.e., their interaction is intermittent. VaVb appear to move together quite frequently up-and down over CD3d and CD3g, respectively. This is suggested by an accurate MDS study by Weikl and co-workers (Elife 2021) and in Prakhaash et al. 2021, the latter also pointing out a couple of potential electrostatic interactions between V regions and CD3 ectodomain residues. The author might wish to cite these two papers, though it looks as they do not feel confident that (accurate and competent) MDS provides reliable predictive value and opportunities to test hypotheses. Importantly, such dynamic bending is visible only when TCR-CD3 is embedded in a lipid bilayer, better if made of a complex mixture of membrane lipids and values the "in-cell" approach of the work.

Response:

We believe that the lack of clearcut evidence is due to lack of sufficient crosslinks between TCR α -CD3 δ as we obtained only one weak crosslink (CD3 δ E55) and not because of the highly dynamical nature of Va domain movement. We have discussed this in the Discussion section in the final version of the manuscript (Page 28). Photo-crosslinking is well suited to study transient domain movements, bending and complex formations which we have detailed in the Introduction section (Page 5). The reviewer has pointed out that dynamic bending in the TCR-CD3 complex is only visible in a lipid bilayer. Our photo-crosslinking experiments were performed on the mammalian cell surface (HEK cells) and we could not obtain clearcut evidence for bending due to lack of sufficient crosslinks between TCR α -CD3 δ .

Reviewer 2:

This is a manuscript on a very hotly and controversially discussed topic, as e.g. witnessed by the report of referee #1. It provides an important puzzle piece to understand the functioning of the TCR.

My comments have been addressed well, the new BioRxiv of Notti has been discussed and new X-link experiments have been performed and integrated. I have also read to point-to-point reply addressing the other referees and to my opinion their comments have been well addressed as far as it was possible.

Thus, I recommend publication of this revised version.

Response:

We thank the reviewer for valuable suggestions to improve the manuscript, acknowledging the importance of our work and recommending the manuscript for publication.

Reviewer 3:

Several structural studies on the whole pictures of unliganded and even liganded TCR-CD3 cryo-EM structures at high resolutions have been reported recently. Among these structures, some are solved in detergents (Dong et al., 2019; Chen et al., 2022; Susac et al., 2022; Saotome et al., 2023) and some are done in nanodiscs (Notti et al., bioRxiv, 2023), a much more native-like bilayer environment. Furthermore, Notti et al. reported that their structure showed distinction from those in detergents, representing a resting state of an unstimulated TCR on T cells. Compared with these reported studies, an important contribution by the manuscript from Natarajan et al. is the characterization of the in situ TCR-CD3 conformation on the cell surface. Thus, it will be intriguing to ask which structure is closer to the native one, and how the conformation change model applies in TCR triggering.

Response:

We thank the reviewer for acknowledging the importance of our work and for the valuable ideas especially suggesting testing the conformational change model involving TCR α -CD3 δ region.

Comment 1 - Based on my suggestion and the information from the Notti et al. manuscript, Natarajan et al. performed further crosslinking analysis at several key sites, such as the interface between TCR α -CD3 δ , in the absence and presence of pMHC tetramer binding to get more insights regarding the conformational change model (Appendix Figure S16). They chose several residues for the crosslinking analysis. Unfortunately, some mutated residues were poorly expressed on the cell surface (e.g. δ N16 and δ K59), α R39 mutant did not reveal any noticeable crosslinks, and only δ E55 showed much weaker crosslink. Based upon these results, they addressed that they could not infer the occurrence of conformational change involving the TCR α -CD3 δ region because they observed no change in the crosslinks upon binding with IEk/MCC tetramer. It is good to see that Natarajan et al. additionally performed careful experiments to demonstrate this point, but at the same time, their conclusions are not convincing enough based on the experimental data at the present stage. They may choose more residues nearby for further crosslinking analysis, or they should add more discussion in their manuscript about the experimental limitation and future perspective and expectation.

Response:

Yes, we agree with the reviewer's statement that the evidence at this stage is not concrete against conformational change model. As suggested, we have elaborated on this in the Discussion section in the revised manuscript (Page 28).

Comment 2 - It is good to see that Natarajan et al. have included discussion and citation on the cross-saturation experiment and in the revised manuscript.

Response:

We thank the reviewer for this suggestion to include a discussion about cross-saturation NMR experiments.

Comment 3 - Natarajan et al. performed additional experiments to investigate the role of cholesterol molecules played in TCR-CD3 complex. It is good to see that they have come to a conclusion that cholesterol plays more

of an allosteric role in maintaining TCR-CD3 extracellular conformation rather than just being critical for the TCR-CD3 complex stability, a new insight on the function of cholesterol molecules in TCR-CD3 complex.

Response:

We thank the reviewer for this suggestion that allowed us to clarify the allosteric role of cholesterol as well as its importance in complex stability.

Dr. Michelle Krogsgaard
New York University Grossman School of Medicine
550 First Ave
Smilow Building Rm 710
New York, NY 10016
United States

Dear Dr. Krogsgaard,

Thank you for the submission of your further revised manuscript to our editorial offices. I now went through your final p-b-p-response and consider the comments of the referees as adequately addressed. I am thus very pleased to accept your manuscript for publication in the next available issue of EMBO reports. Thank you for your contribution to our journal.

Yours sincerely,
